# Efficient ammonia synthesis from the air using tandem non-thermal plasma and electrocatalysis at ambient conditions

Wei Liu[1], Mengyang Xia[1], Chao Zhao[1], Ben Chong[1], Jiahe Chen[1], He Li[1], Honghui Ou[1] & Guidong Yang [1] ✉

While electrochemical $N_2$ reduction presents a sustainable approach to $NH_3$ synthesis, addressing the emission- and energy-intensive limitations of the Haber-Bosch process, it grapples with challenges in $N_2$ activation and competing with pronounced hydrogen evolution reaction. Here we present a tandem air-$NO_x$-$NO_x^-$-$NH_3$ system that combines non-thermal plasma-enabled $N_2$ oxidation with $Ni(OH)_x$/Cu-catalyzed electrochemical $NO_x^-$ reduction. It delivers a high $NH_3$ yield rate of 3 mmol $h^{-1}$ $cm^{-2}$ and a corresponding Faradaic efficiency of 92% at −0.25 V versus reversible hydrogen electrode in batch experiments, outperforming previously reported ones. Furthermore, in a flow mode concurrently operating the non-thermal plasma and the $NO_x^-$ electrolyzer, a stable $NH_3$ yield rate of approximately 1.25 mmol $h^{-1}$ $cm^{-2}$ is sustained over 100 h using pure air as the intake. Mechanistic studies indicate that amorphous $Ni(OH)_x$ on Cu interacts with hydrated $K^+$ in the double layer through noncovalent interactions and accelerates the activation of water, enriching adsorbed hydrogen species that can readily react with N-containing intermediates. In situ spectroscopies and density functional theory (DFT) results reveal that $NO_x^-$ adsorption and their hydrogenation process are optimized over the $Ni(OH)_x$/Cu surface. This work provides new insights into electricity-driven distributed $NH_3$ production using natural air at ambient conditions.

With an annual production of more than 175 million tonnes, ammonia ($NH_3$) synthesized by the Haber-Bosch process at elevated temperatures and pressures has underpinned the population boom over the past century and is considered a promising carbon-neutral energy vector[1,2]. The huge energy consumption and carbon footprint of the Haber-Bosch process call for sustainable $NH_3$ production methods that can operate under ambient conditions and make use of renewable energy sources[3,4]. One such alternative−electrochemical $N_2$ reduction (eNRR)−has gained a surge of attention by its potential for decarbonizing $NH_3$ production[5–8]. While encouraging, eNRR in aqueous solution generally encounters poor $NH_3$ yield rate (-$10^{-1}$ nmol $s^{-1}$ $cm^{-2}$) as a result of high activation barrier of

stable N≡N bond and low solubility of $N_2$[9]. Additionally, severe hydrogen evolution reaction (HER) outcompetes eNRR, resulting in a Faradaic efficiency (FE) of less than 1%. Compared to eNRR, electrocatalytic $NO_x^-$ reduction reactions (e$NO_x^-$RR, x = 2, 3) are more likely to occur because of their high solubility and lower activation energy of N=O bond (204 kJ $mol^{-1}$) than that of N≡N bond (945 kJ $mol^{-1}$)[10–12]. Furthermore, e$NO_x^-$RR is thermodynamically more favorable than HER and thus can proceed in a wide potential window without HER interference[13]. Therefore, efficient electrocatalytic synthesis of $NH_3$ is expected to be achieved by converting stable $N_2$ into more reactive $NO_x^-$, followed by an electrochemical reduction process[14,15].

[1]A XJTU-Oxford International Joint Laboratory for Catalysis, School of Chemical Engineering and Technology, Xi'an Jiaotong University, Xi'an, Shaanxi 710049, China. ✉e-mail: guidongyang@xjtu.edu.cn

Inspired by natural lightning, non-thermal plasma (NTP)-enabled $N_2$ oxidation reaction (pNOR) has been recently studied in the field of electrocatalytic $NH_3$ synthesis[16–20]. For example, a spark discharge NTP was used to oxidize $N_2$ into $NO_x$, then the generated $NO_x$ species were captured to obtain an electrolyte containing $NO_x^-$, which underwent electrochemical reduction to synthesize $NH_3$[19]. Owing to its favorable $NH_3$ selectivity, Cu-based catalysts are frequently employed in $eNO_x^-RR$[19,20]. Although $eNO_x^-RR$ are not limited by reactant activation, the mismatch between fast electron transfer and sluggish proton supply over Cu restricts the $NH_3$ synthesis efficiency in this pNOR-$eNO_x^-RR$ tandem system. This arises due to the requirement of an alkaline solution for effective absorption of the $NO_x$ generated from pNOR. Within this alkaline environment, the $H_2O$ dissociation process on Cu, responsible for generating adsorbed hydrogen species ($H_{ad}$) involved in numerous deoxygenation and hydrogenation steps of $NO_x^-$, lags significantly behind the rapid $NO_x^-$ consumption process[21–24]. Given that 9 and 7 protons are involved in $eNO_3^-RR$ and $eNO_2^-RR$, respectively, a large overpotential is required to reach a dynamic equilibrium between the generation of $H_{ad}$ and its timely consumption. Under such potentials, the competitive adsorption of $H_2O$ with $NO_x^-$ on Cu will deteriorate the $NH_3$ selectivity[25]. In addition, large overpotentials will lead to unsatisfactory energy efficiency for $eNO_x^-RR$ due to more energy loss. In this context, improving the $H_{ad}$ supply efficiency on the Cu surface at low overpotentials to meet the demand of the hydrogenation process is crucial but remains challenging.

Herein, we demonstrate an efficient tandem $NH_3$ synthesis route of pNOR-$eNO_x^-RR$ by combining a spark discharge NTP for air-to-$NO_x$ conversion with $Ni(OH)_x/Cu$-catalyzed $NO_x^-$-to-$NH_3$ electroreduction. Kinetic isotopic effect (KIE) evaluation, electron paramagnetic resonance (EPR) measurement, molecular dynamics (MD) simulations, DFT, and in situ Raman spectra results reveal that the deposition of $Ni(OH)_x$ induces improved water activation and $NO_x^-$ adsorption at the interface between $Ni(OH)_x$ and Cu. The enriched $H_{ad}$ cater to the hydrogenation needs of nitrogenous intermediates on the Cu surface, enabling $Ni(OH)_x/Cu$ to achieve efficient $eNO_x^-RR$ toward $NH_3$ synthesis at low overpotentials. Further coupled with the optimized pNOR, the pNOR-$eNO_x^-RR$ tandem system delivers a record $NH_3$ yield rate of 3 mmol $h^{-1}$ $cm^{-2}$ with FE of 92% at −0.25 V vs. RHE in batch experiments, ranking among the highest performances reported to date. More significantly, in a flow mode concurrently operating the pNOR and the $eNO_x^-RR$ using pure air as the feeding gas, a stale $NH_3$ yield rate of ca. 1.25 mmol $h^{-1}$ $cm^{-2}$ is achieved over 100 h, culminating in the production of high-purity solid $NH_4Cl$ and liquid $NH_3$ solution through an air stripping method. This work not only offers a strategy to develop superior $eNO_x^-RR$ electrocatalysts but also provides insights for sustainable and distributed $NH_3$ synthesis leveraging atmospheric nitrogen and renewable electricity.

## Results and discussion
### Catalyst synthesis and characterization
The synthetic procedure of the $Ni(OH)_x/Cu$ electrode is illustrated in Fig. 1a. Firstly, Cu foam was chemically oxidized by $(NH_4)_2S_2O_8$ in an alkaline solution and then calcinated in the air to obtain CuO nanowire array (NWA)[26]. $Ni(OH)_x$ was deposited on the CuO NWA surface by rinsing CuO NWA into a $NiCl_2$ solution at open circuit potential (Supplementary Fig. 1)[27,28]. According to the Pourbaix diagrams of Cu and Ni, a cyclic voltammetry (CV) prereduction process within the potential range from −0.3 to 0.2 V in 1 M KOH with 0.1 M $NO_3^-$ was then performed to obtain $Ni(OH)_x/Cu$ NWA (Supplementary Fig. 2). Scanning electron microscopy (SEM) and transmission electron microscopy (TEM) images show that $Ni(OH)_x/Cu$ NWA well inherit the micrometer-long nanowire morphology from the CuO NWA and is evenly distributed on the Cu foam skeleton (Fig. 1b and Supplementary Figs. 3, 4). Element mapping images (Supplementary Figs. 5, 6)

indicate the homogeneous distribution of Cu, Ni, and O over porous $Ni(OH)_x/CuO$ and $Ni(OH)_x/Cu$ nanowires. In the high-resolution TEM image of $Ni(OH)_x/Cu$, a clear interface is observed between an amorphous layer and a crystalline phase (Fig. 1c). A lattice spacing of 0.208 nm is recognized in the crystalline zone and attributed to the (111) facet of Cu. High-angle annular dark-field scanning transmission electron microscopy (HAADF-STEM) combined with energy-dispersive X-ray spectroscopy further manifests the external coverage of nickel species on the Cu surface (Fig. 1d). As shown in Fig. 1e, the Raman signals appeared at 150, 218, 302, and 630 $cm^{-1}$ which can be assigned as Cu oxides vanish after the CV prereduction process, indicating the reduction of Cu oxides to a metallic state[29,30]. Cu oxides reduction can be further confirmed by X-ray diffraction (XRD) patterns (Supplementary Fig. 7), Cu 2p spectra, and Cu LMM Auger spectra over $Ni(OH)_x/CuO$ and $Ni(OH)_x/Cu$ electrodes (Supplementary Fig. 8a, b).

To detect the composition of the deposited nickel species over the Cu surface, we carried out various spectroscopic tests. In the high-resolution Ni 2p XPS spectra of $Ni(OH)_x/CuO$ and $Ni(OH)_x/Cu$, two prominent peaks at bind energy of 855.9 and 873.4 eV can be assigned to Ni $2p_{3/2}$ and Ni $2p_{1/2}$ of $Ni^{2+}$ spin-orbit doublets and those at 861.5 and 879.2 eV are ascribed to two accompanying satellites (Supplementary Fig. 8d)[31]. The O 1s spectra exhibit a clear peak at 531.5 eV, implying that nickel species exist as hydroxide (Supplementary Fig. 8c)[32]. Raman spectra of $Ni(OH)_x/CuO$ and $Ni(OH)_x/Cu$ nanowire were obtained by prolonging the $Ni(OH)_x$ deposition time to a detectable level (Supplementary Fig. 9). The bands at 457 and 500 $cm^{-1}$ are attributable to lattice stretching modes of Ni-OH and Ni-O, respectively. The symmetric O-H stretching mode of the free external -OH group is observed at 3690 $cm^{-1}$[33,34]. Additionally, the selected area electron diffraction of $Ni(OH)_x/Cu$ only shows the polycrystalline nature of Cu without any detection of crystalline $Ni(OH)_2$ (Supplementary Fig. 10), suggesting that the deposited nickel species exist in an amorphous structure. The fine structure of the deposited $Ni(OH)_x$ was investigated via X-ray adsorption spectroscopy. As shown in Fig. 1f and Supplementary Fig. 11, $Ni(OH)_x/Cu$ exhibits a very small edge shift in Ni K-edge X-ray absorption near edge structure (XANES) compared with the $Ni(OH)_2$ reference, indicating the valence state of Ni in $Ni(OH)_x/Cu$ is slightly smaller than +2. The Fourier transform extended X-ray absorption fine structure (FT-EXAFS) spectrum (Fig. 1g) and fitting results (Supplementary Fig. 12 and Supplementary Table 1) show the presence of two distinct peaks at -1.6 and -2.7 Å, attributable to Ni-O and Ni-Ni bonds of $Ni(OH)_2$[35,36]. The nearest neighbor Ni-O and Ni-Ni bond distances of $Ni(OH)_x/Cu$ are 2.045 ± 0.001 Å and 3.113 ± 0.001 Å, respectively. These values closely match those of Ni-O and Ni-Ni bond distances in the $Ni(OH)_2$ standard (2.050 ± 0.001 Å and 3.119 ± 0.007 Å). Yet, the coordination numbers (CN) of Ni-O (4.6 ± 0.3) and Ni-Ni (4.8 ± 0.7) are reduced compared to those of the crystalline $Ni(OH)_2$ (CN of 6 for both Ni-O and Ni-Ni), which further confirms its disordered structure. Taken together, we conclude that the deposited nickel species on the Cu surface is amorphous $Ni(OH)_x$, rather than Ni, NiO, or CuNi alloy. Other control samples, including Cu, $Ni(OH)_2$, and $Ni(OH)_x/Cu$ with different $Ni(OH)_x$ deposition time, were also prepared, characterized, and evaluated (Supplementary Figs. 7, 13, and 14).

### Evaluation of electrocatalytic performance in model electrolytes
Having established the structural properties of $Ni(OH)_x/Cu$ electrocatalyst, we then evaluated its $eNO_3^-RR$ performance in a model electrolyte of 1 M KOH with 0.1 M $NO_3^-$. Given the presence of $NO_3^-$ and $NO_2^-$ in the pNOR-$eNO_x^-RR$ system, and with $NO_2^-$ serving as an intermediate in the $eNO_3^-RR$ process, it is more representative to direct our focus onto $eNO_3^-RR$. The amounts of reactant ($NO_3^-$) and products ($NH_3$ and $NO_2^-$) were quantified using ion chromatography and colorimetric methods, respectively (Supplementary Figs. 15−17).

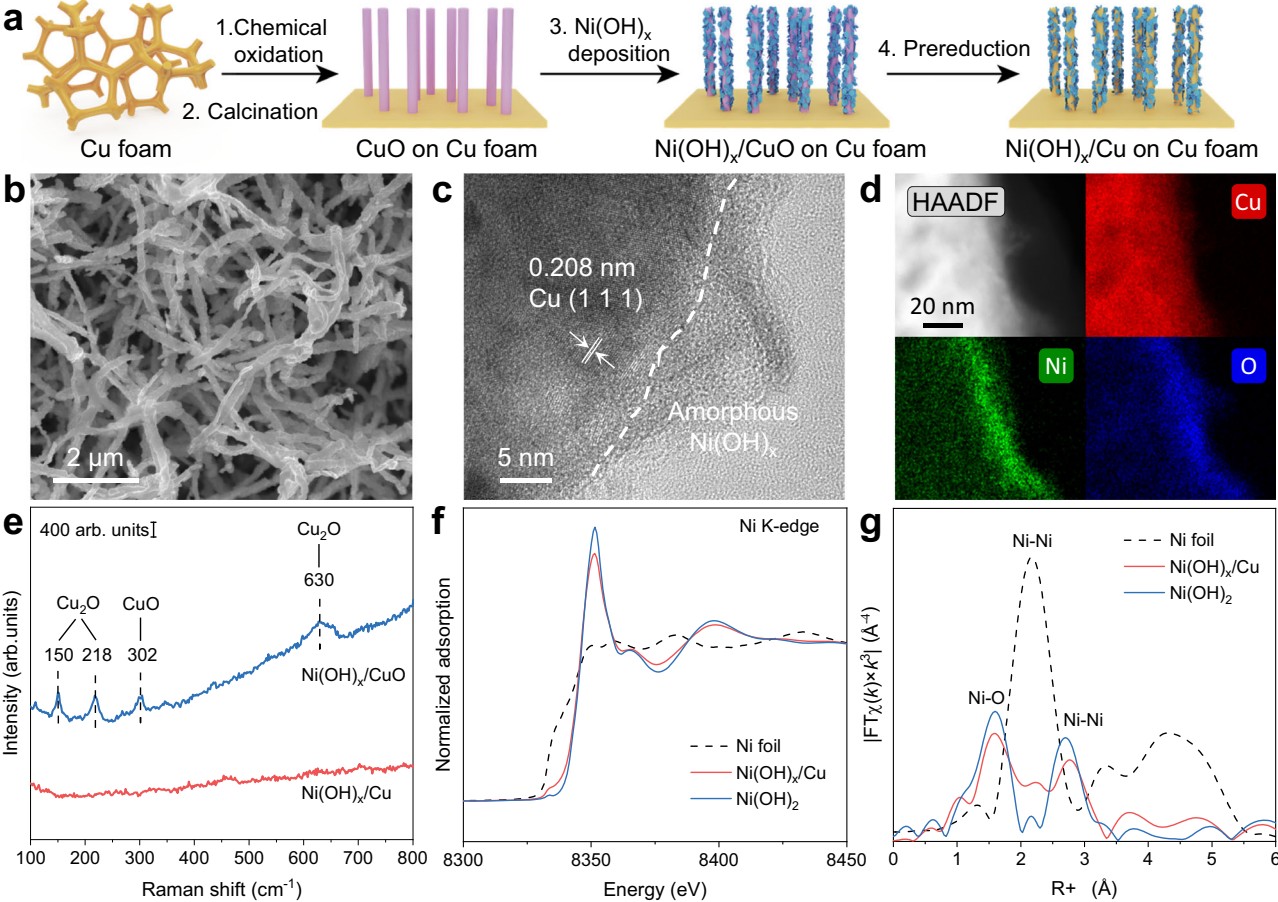

**Fig. 1 | Characterization of the structure and composition of Ni(OH)$_x$/Cu.**
**a** Schematic diagram of the synthetic route for Ni(OH)$_x$/Cu. **b** SEM image. **c** High-resolution TEM image of Ni(OH)$_x$/Cu. **d** HAADF STEM-EDS mapping of Ni(OH)$_x$/Cu. **e** Raman spectra of Ni(OH)$_x$/CuO and Ni(OH)$_x$/Cu. **f** Ni K-edge XANES and (**g**) Fourier-transformed EXAFS spectra of Ni foil, Ni(OH)$_x$/Cu, and Ni(OH)$_2$.

All the potentials were reported versus RHE unless otherwise stated (Supplementary Fig. 18).

The linear sweep voltammetry (LSV) polarization curves in Fig. 2a and Supplementary Fig. 19 reveal that, in the presence of NO$_3^-$, eNO$_3^-$RR exhibits greatly enhanced current density when compared to the HER LSV curve in the absence of NO$_3^-$ over Cu, Ni(OH)$_2$, and Ni(OH)$_x$/Cu. Ni(OH)$_x$/Cu shows both higher HER and eNO$_3^-$RR activities than those of the Cu electrode. At overpotentials less than −0.2 V, Cu shows no more than 60% of NH$_3$ FE, which is consistent with previous reports that Cu exclusively catalyzes the conversion of NO$_3^-$ to NO$_2^-$ at low overpotential (Fig. 2b and Supplementary Fig. 20)[12]. In contrast, with a maximum FE of 91.6%, Ni(OH)$_x$/Cu demonstrates superior NH$_3$ yield rates than Cu and Ni(OH)$_2$ counterparts after electrochemically active surface area (ECSA)-normalization, indicating the superior intrinsic activity of Ni(OH)$_x$/Cu (Fig. 2b, c and Supplementary Figs. 21, 22). Figure 2d shows that the partial current density for NH$_3$ production of Ni(OH)$_x$/Cu reaches 639.6 mA cm$^{-2}$ at −0.3 V, which is two times and three orders of magnitude higher than that of Cu and Ni(OH)$_2$, respectively. Interestingly, the eNO$_3^-$RR performances of Ni(OH)$_x$/Cu electrodes are varied with the coverage of deposited Ni(OH)$_x$ and exceed that of the Ni(OH)$_x$/CuO electrode (Supplementary Figs. 23−25 and Supplymentary Table 2). The isotope labeling experiment using K$^{15}$NO$_3$ as the reactant was conducted to eliminate potential ammonium contamination from interfering with the results. As shown in Fig. 2e, when using $^{15}$NO$_3^-$ as the nitrogen source, only doublet peaks of $^{15}$NH$_4^+$ are detected in the $^1$H nuclear magnetic resonance (NMR) spectra of the electrolyte without seeing any triple coupling peaks of

$^{14}$NH$_4^+$. The batch NO$_3^-$-to-NH$_3$ conversion capacity of Ni(OH)$_x$/Cu was evaluated by prolonging the electrolysis time. With 99% selectivity of NO$_3^-$-to-NH$_3$ and an overall FE of more than 98%, almost complete transformation can be achieved within 80 min (Supplementary Fig. 26). The appearance and disappearance of NO$_2^-$ during the reaction indicate that it is an intermediate product and can be further reduced to NH$_3$. Such an outstanding eNO$_3^-$RR activity of the Ni(OH)$_x$/Cu was also assessed in the electrolytes with varied NO$_3^-$ concentrations (Supplementary Fig. 27). As shown in Fig. 2f, Ni(OH)$_x$/Cu exhibits over 80% FE of NH$_3$ and over 0.75 of selectivity in the NO$_3^-$ concentration range from 5 mM to 1 M, indicating NH$_3$ synthesis is the main reaction. However, when the concentration diminishes to 1 mM, the FE drops to 70%, which could be attributed to the intensified HER at low NO$_3^-$ concentration. At an extremely high NO$_3^-$ concentration of 2 M, NH$_3$ production will be unfavorable due to the insufficient hydrogenation of NO$_3^-$. The results from 15 independent experiments and 10 cyclic tests demonstrated good repeatability and stability of the Ni(OH)$_x$/Cu catalyst (Supplementary Figs. 28, 29). In addition, the long-term stability of Ni(OH)$_x$/Cu was also evaluated by chronopotentiometry in an H-type flow cell with continuous electrolyte flow to replenish the constantly consumed NO$_3^-$ (Supplementary Fig. 30)[12]. The voltage for maintaining the current of 400 mA cm$^{-2}$ remains stable over 25 h, and the FE of NH$_3$ reaches a plateau of 90%, indicating the outstanding stability of the Ni(OH)$_x$/Cu. SEM, XRD, and TEM results reveal that the structure and composition of Ni(OH)$_x$/Cu remain intact after the long-term test (Supplementary Fig. 31). In addition, the eNO$_2^-$RR activity of Ni(OH)$_x$/Cu also surpasses those of Cu and Ni(OH)$_2$ (Supplementary Figs. 32, 33).

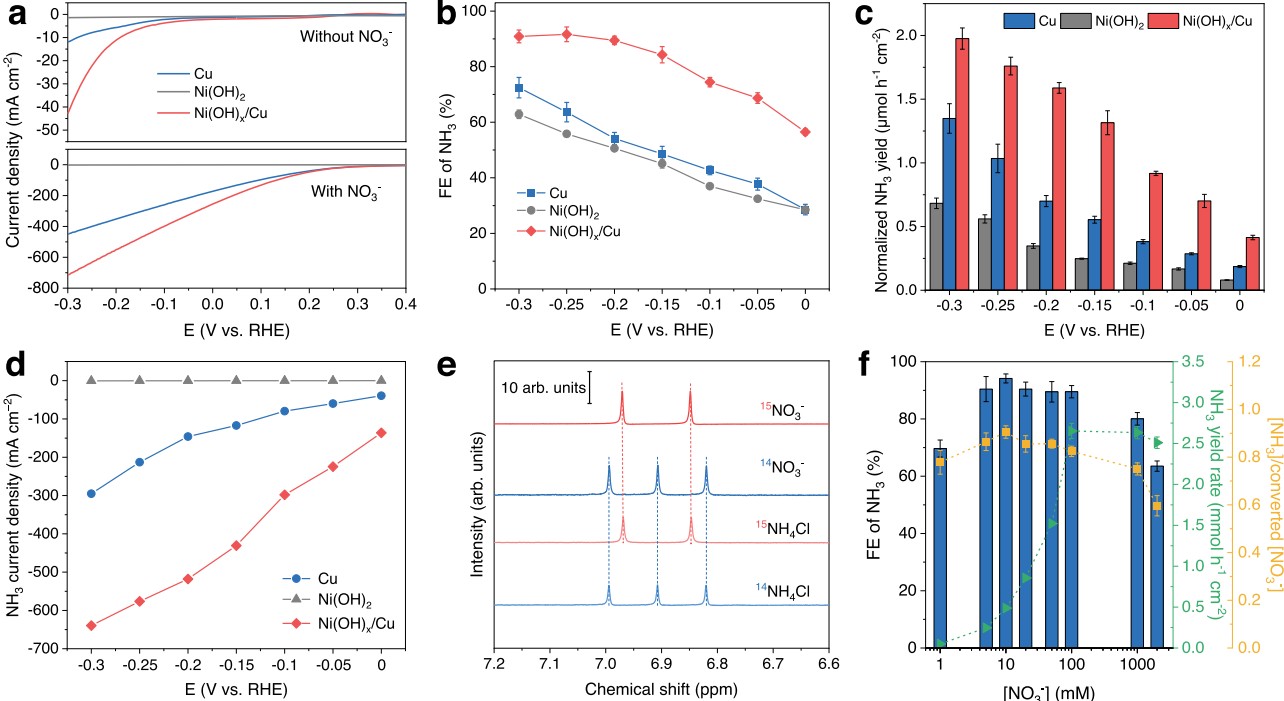

**Fig. 2 | Evaluation of the eNO₃⁻RR performance of electrodes in the model electrolyte. a** LSV curves of Cu, Ni(OH)₂, and Ni(OH)ₓ/Cu in 1 M KOH with and without 0.1 M NO₃⁻ under 400 r.p.m. without *iR* correction. Potential-dependent FEs of NH₃ (**b**), NH₃ yield rates (**c**), and NH₃ partial current densities (**d**) over Cu, Ni(OH)₂, and Ni(OH)ₓ/Cu. **e** NMR spectra of the ¹⁴NH₄⁺ and ¹⁵NH₄⁺ references and the electrolyte after eNO₃⁻RR over Ni(OH)ₓ/Cu at −0.25 V vs. RHE using ¹⁴NO₃⁻ and ¹⁵NO₃⁻ as the nitrogen source. **f** The FE of NH₃, NH₃ yield rate, and selectivity (the ratio of the generated NH₃ concentration [NH₃] to the consumed NO₃⁻ concentration [NO₃⁻]) over Ni(OH)ₓ/Cu at −0.25 V vs. RHE at NO₃⁻ concentration in the range from 1 mM to 2 M. The error bars represent the standard deviation from at least three independent measurements.

Taken together, the Ni(OH)ₓ/Cu catalyst can be used in the tandem pNOR-eNOₓ⁻RR system as an efficacious and stable catalyst for the eNOₓ⁻RR process.

**Mechanism insight**
In the alkaline media, the performance of eNO₃⁻RR on Cu strongly dependent on H$_{ad}$ that generated from water dissociation, which participate in both the deoxygenation and hydrogenation steps of adsorbed NO₃⁻ [22,23]. Due to endothermic water adsorption and sluggish kinetics of water dissociation on Cu, eNO₃⁻RR over Cu predominantly produces NO₂⁻ at low overpotentials (Supplementary Fig. 20)[12]. While at more negative overpotentials of HER region, high FE of NH₃ can be obtained on Cu (Supplementary Fig. 34). Interestingly, the eNO₃⁻RR performances of the Ni(OH)ₓ/Cu electrodes show a correlation with their HER activities (Supplementary Fig. 23). Based on these observations, we hypothesize that the improved eNO₃⁻RR performance on Ni(OH)ₓ/Cu is due to its enhanced water activation.

We first performed kinetic isotopic effect (KIE) evaluation, a well-established method for studying reactions involving protons by substituting hydrogen with deuterium[37–39]. Cathode shifts are observed over the LSV curves of HER and eNO₃⁻RR on Cu and Ni(OH)ₓ/Cu because of the more sluggish dissociation kinetics of D₂O than that of H₂O (Fig. 3a, b)[40]. The Tafel slope for HER over Ni(OH)ₓ/Cu is 175 mV/dec, which is smaller than that of Cu (275 mV/dec), suggesting that the Volmer step (H₂O dissociation) is the rate-determining step in alkaline HER and more efficient H₂O dissociation kinetics over Ni(OH)ₓ/Cu (Supplementary Fig. 35)[41]. Additionally, the values of cathode shift for LSV cureves of HER and eNO₃⁻RR over Ni(OH)ₓ/Cu are smaller than those over Cu, indicating that the accelerated H₂O dissociation kinetics makes the Ni(OH)ₓ/Cu electrode less restricted by the isotope substitution (Supplementary Fig. 36).

To gain further insights into the role of the activation of H₂O in NO₃⁻ reduction, we carried out studies on the KIE of H/D over Cu and Ni(OH)ₓ/Cu. The KIEs of H/D, defined as the ratio of NH₃ yield rate in H₂O and D₂O, are calculated to be 5.15 and 1.96 for Cu and Ni(OH)ₓ/Cu, respectively (Fig. 3c and Supplementary Fig. 37). These KIE values are characteristics of the primary kinetic isotope effect, implying that the H-OH bond breaking is involved in the rate-determining step for NO₃⁻ reduction to NH₃[38]. With a more sluggish proton supply, Cu only shows 23.4% of NH₃ FE in the D₂O media. By contrast, the smaller KIE value and the well-retained FE of NH₃ (72.2%) over Ni(OH)ₓ/Cu in D₂O indicates that the eNO₃⁻RR is less limited by H₂O dissociation with the presence of surficial Ni(OH)ₓ.

To compare the amount of H$_{ad}$ on the surface of Cu and Ni(OH)ₓ/Cu qualitatively, CV curves in the Ar-saturated 1 M KOH solution were recorded at a potential range of −0.3–0.05 V (Supplementary Fig. 38)[40]. The H$_{ad}$ desorption peak of Ni(OH)ₓ/Cu in the potential range of −0.2 - −0.1 V is more significant than that of Cu, suggesting the existence of more H$_{ad}$ on the surface of Ni(OH)ₓ/Cu. We further performed EPR measurement using 5,5-dimethyl-1-pyrroline-N-oxide (DMPO) as the H$_{ad}$ trapping reagent to verify the generation of H$_{ad}$ directly. A typical DMPO-H spin adduct signal pattern consisting of nine EPR peaks with an intensity ratio of 1:1:2:1:2:1:2:1:1 is observed in Fig. 3d, e[13]. The EPR signals of Ni(OH)ₓ/Cu are much more intense than that of Cu, implying Ni(OH)ₓ indeed improves H$_{ad}$ formation. However, no signals of DMPO-H adduct are observed in the presence of 0.1 M NO₃⁻, suggesting the generated H$_{ad}$ is fast consumed by the hydrogenation process of adjacent N-containing intermediates, which is consistent with the results of the H$_{ad}$ quenching test by adding DMPO (Supplementary Figs. 37, 39).

Note that anion-hydrated cation networks (X^{δ−}-M⁺(H₂O)ₙ, where n refers to the number of ionic hydrations) can be formed in the double layer through non-covalent Coulomb interactions between

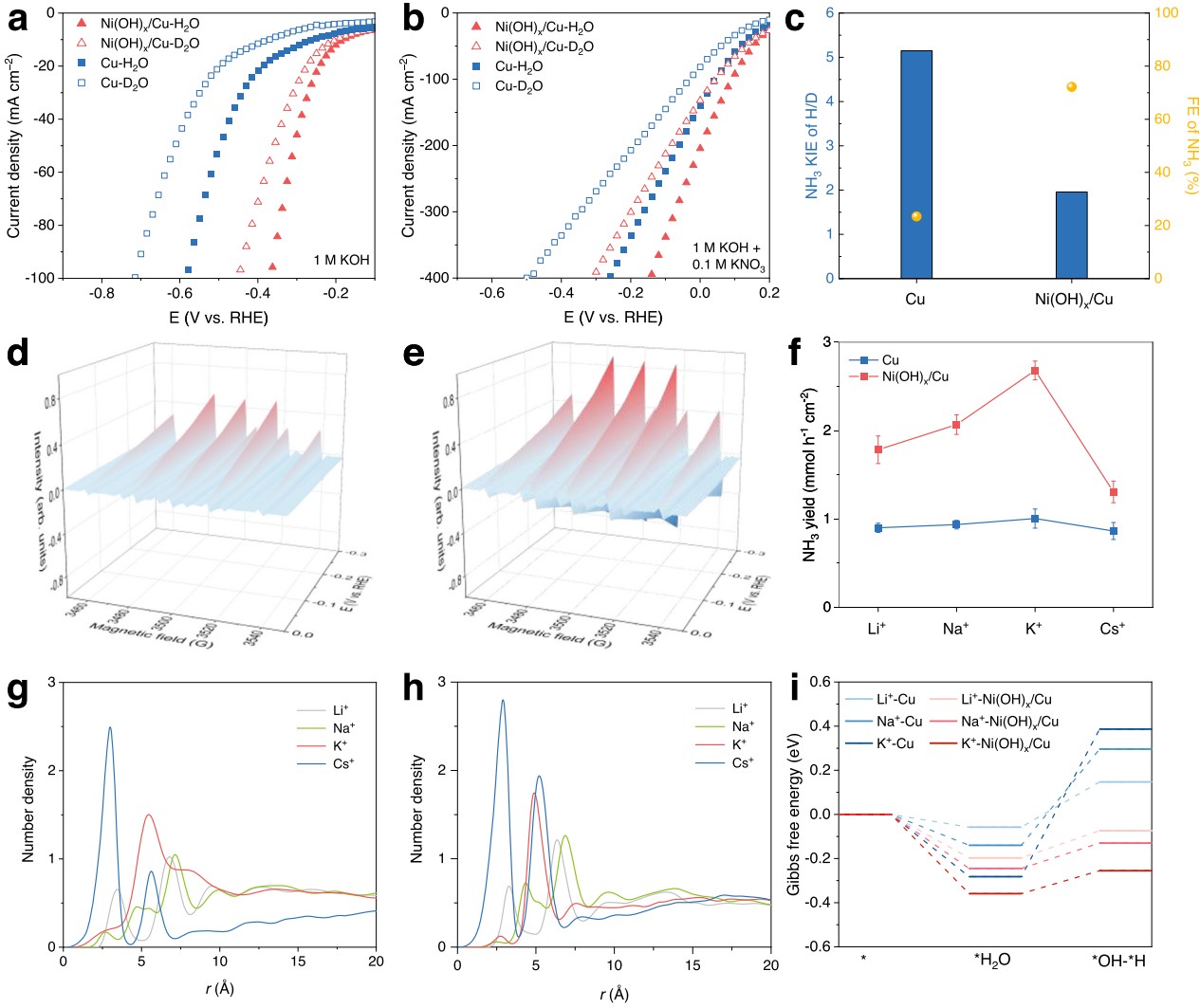

**Fig. 3 | Functioning mechanism of nickel species on Ni(OH)$_x$/Cu for eNO$_x^-$RR.** Comparison of the LSV polarization curves over Cu and Ni(OH)$_x$/Cu in electrolytes using H$_2$O or D$_2$O as the solvent without (**a**) and with (**b**) adding 0.1 M NO$_3^-$. **c** KIE of H/D for NH$_3$ synthesis over Cu and Ni(OH)$_x$/Cu and the corresponding FE of NH$_3$ in the presence of 0.1 M NO$_3^-$. DMPO-involved EPR spectra of Cu (**d**) and Ni(OH)$_x$/Cu (**e**) at different overpotentials in the absence of NO$_3^-$. **f** Effect of alkali metal cations on the eNO$_3^-$RR performance at −0.25 V vs. RHE over Cu and Ni(OH)$_x$/Cu. Density profiles of Li$^+$, Na$^+$, K$^+$, and Cs$^+$ over the Cu (**g**) and the Ni(OH)$_x$/Cu (**h**). **i** Gibbs free energy diagram of H$_2$O adsorption and dissociation on Cu and Ni(OH)$_x$/Cu surface in the presence of different alkali metal cation hydrates. The error bars represent the standard deviation from at least three independent measurements.

surface-adsorbed anionic species (e.g., F$^-$, S$^{\delta-}$) and hydrated cations in the electrolyte, thus enhancing H$_2$O activation to generate H$_{ad}$[38]. To validate whether a similar promotion effect is involved in our reaction, we investigated the impact of different alkali metal cations in MOH and MNO$_3$ (M = Li, Na, K, and Cs) electrolytes on eNO$_3^-$RR performance (Fig. 3f). Even though the impact of the alkali cation over Cu is very limited due to its weak interaction with hydrated cation, the NH$_3$ yield rate over Ni(OH)$_x$/Cu increases from 1.79 to 2.7 mmol h$^{-1}$ cm$^{-2}$ on changing the alkali cation from Li$^+$ to K$^+$, and decreases to 1.3 mmol h$^{-1}$ cm$^{-2}$ in the presence of Cs$^+$. Since the cation hydrate's structure and behavior can be varied with parameters like pH, potential on the electrode, and so on, classic molecular dynamics (MD) simulations were performed to gain a molecular perspective on this cation effect[42,43]. As displayed in Fig. 3g, h, the z-axial cation number density profiles from MD simulations revealed increasing peak intensity in the order of Li$^+$ < Na$^+$ < K$^+$ < Cs$^+$, indicating the greater willingness of large, weakly solvated cations to approach the electrode surface. The peak centers for Li$^+$, Na$^+$, K$^+$, and Cs$^+$ cations on the Ni(OH)$_x$/Cu (6.4 Å, 6.9 Å, 4.9 Å, and 2.9 Å) are closer to the surface than those on Cu (6.8 Å, 7.2 Å, 5.5 Å, and 3 Å). In addition, the number

densities corresponding to the highest peak for Li$^+$, Na$^+$, K$^+$, and Cs$^+$ on the Ni(OH)$_x$/Cu (1.2, 1.3, 1.7, and 2.8) are larger than those of on the Cu (1.0, 1.1, 1.5, and 2.5). These results suggest that Ni(OH)$_x$ species can attract cations close to the electrified interface. We then calculated the Gibbs free energy of H$_2$O adsorption and dissociation processes on Cu and Ni(OH)$_x$/Cu in the presence of different alkali metal cation hydrates (Fig. 3i and Supplementary Figs. 40, 41). The water coordination numbers were set as 4, 5, and 7 for Li$^+$, Na$^+$, and K$^+$ cations, respectively, based on the integration of cation−O (in H$_2$O) radial distribution functions in Supplementary Fig. 42. Cs$^+$ was excluded from the DFT calculations due to they mainly distribute at a distance of 3 Å from the surface, where they undergo physisorption or chemisorption on the catalyst surface and are expected to influence the eNO$_3^-$RR activity in a very different way (Supplementary Fig. 43). On the Cu surface, the Gibbs free energies for water adsorption (* → *H$_2$O) in the presence of Li$^+$(H$_2$O)$_4$, Na$^+$(H$_2$O)$_5$, and K$^+$(H$_2$O)$_7$ are −0.058 eV, −0.14 eV, and −0.282 eV, respectively, whereas the subsequent water dissociation process on Cu (*H$_2$O → *OH-*H) delivers Gibbs free energy uphill of 0.205 eV, 0.436 eV, and 0.668 eV. These results demonstrate that water adsorption on the Cu

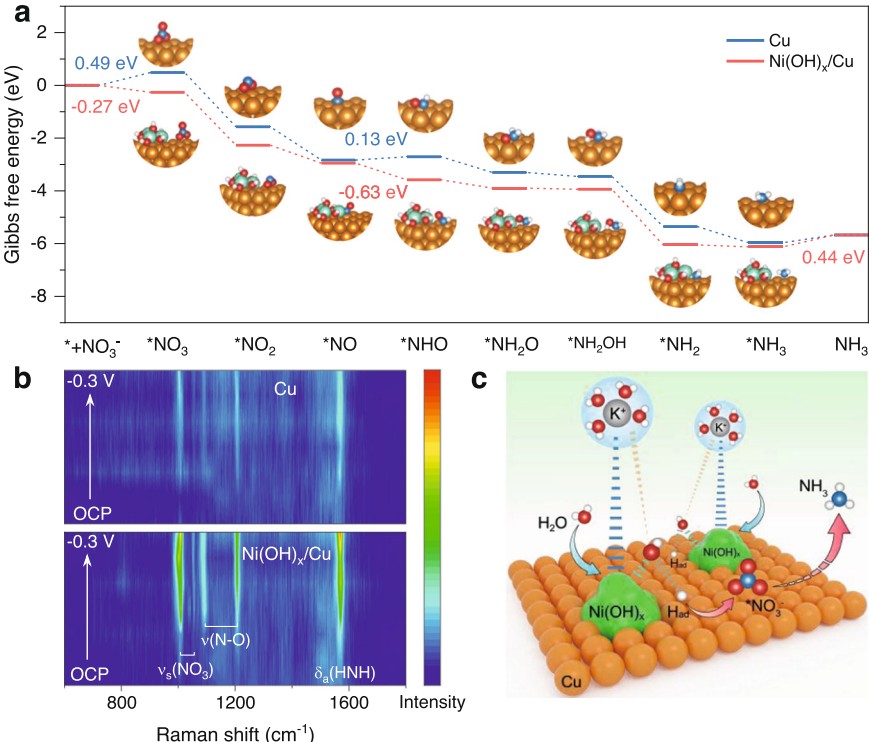

**Fig. 4 | Investigation of the eNOₓ⁻RR mechanism. a** Gibbs free energy diagram of eNO₃⁻RR on Cu and Ni(OH)ₓ/Cu. **b** Potential-dependent in situ Raman contour maps of Cu and Ni(OH)ₓ/Cu obtained during eNO₃⁻RR in the potential range from 0.2 to −0.3 V vs. RHE. **c** Schematic illustration for the role of Ni(OH)ₓ species in promoting water activation and N-containing intermediates hydrogenation on Cu.

surface becomes more favorable with the change of the cation from Li⁺ to K⁺, while the water dissociation process becomes more unfavorable. Thus, these two opposite influences eventually make the cation effect on the generation of $H_{ad}$ on the Cu surface very limited. On the Ni(OH)ₓ/Cu surface, the Gibbs free energies of water adsorption are −0.197 eV, −0.245 eV, and −0.359 eV in the presence of Li⁺(H₂O)₄, Na⁺(H₂O)₅, and K⁺(H₂O)₇, respectively. The subsequent water dissociation process on Ni(OH)ₓ/Cu delivers Gibbs free energy uphill of 0.124 eV, 0.115 eV, and 0.104 eV. Compared with Cu, Ni(OH)ₓ/Cu shows more Gibbs free energy downhill of the water adsorption process and less Gibbs free energy uphill of the water dissociation process in the presence of the same cation hydrate, indicating that the water adsorption and dissociation on Ni(OH)ₓ/Cu are more favorable. Notably, the Gibbs free energy for water adsorption and the energy barrier for water dissociation on Ni(OH)ₓ/Cu both decreases in the order of Li⁺ > Na⁺ > K⁺, indicating that the water adsorption and dissociation processes become more favorable with increasing cation size. This cation effect is well aligned with the experimentally observed cation-dependent HER activities (Supplementary Fig. 44).

To uncover the reaction pathway of eNO₃⁻RR to NH₃, the reaction intermediates were monitored via in situ electrochemical attenuated total reflection surface-enhanced infrared spectroscopy (ATR-SEIRAS) (Supplementary Fig. 45). The absorption peak that appeared at 1190 cm⁻¹ is assignable to -N-O- stretching vibration of hydroxylamine (NH₂OH)[44]. We then performed DFT calculations for eNO₃⁻RR toward NH₃ on Cu and Ni(OH)ₓ/Cu surface. The structural models in this work are shown in Supplementary Fig. 46. Figure 4a shows that all the Gibbs free energies of intermediates on the Ni(OH)ₓ/Cu surface are smaller than those on the Cu surface, which means that the eNO₃⁻RR activity on Ni(OH)ₓ/Cu is superior. On pure Cu, the first NO₃⁻ adsorption step is the potential-determining step (PDS), of which the maximum free energy uphill reaches 0.49 eV. In

contrast, Ni(OH)ₓ/Cu with a free energy downhill of −0.27 eV is more likely to adsorb NO₃⁻. The PDS is changed from the NO₃⁻ adsorption to the *NH₂ protonation step (0.44 eV) upon introducing Ni(OH)ₓ. Additionally, the hydrogenation process of *NO → *NHO is more likely to occur over Ni(OH)ₓ/Cu (−0.63 eV) than Cu (0.13 eV), thus the reaction is facilitated over Ni(OH)ₓ/Cu. The facilitated adsorption of intermediates on Ni(OH)ₓ/Cu can be evidenced by the in situ Raman spectra, where the intensity of signals designated as the symmetric stretch of *NO₃, N-O stretch with nitrito binding intermediates, and antisymmetric stretch of H-N-H in NH₃ are more significant over Ni(OH)ₓ/Cu (Fig. 4b and Supplementary Fig. 47)[45].

Based on the above results and analysis, we propose that the Ni(OH)ₓ species on Cu enrich hydrated cations (K⁺(H₂O)ₙ) near the catalyst surface in the double layer through noncovalent interactions. The near-surface H₂O molecular can be facilely adsorbed and dissociated at the Ni(OH)ₓ – Cu interface, forming $H_{ad}$ to participate in the following hydrogenation steps of N-containing intermediates adsorbed on Cu toward NH₃ (Fig. 4c).

## Evaluation of the performance of air-to-NH₃ conversion in the pNOR-eNOₓ⁻RR tandem system

In light of the above findings, we further evaluated the NH₃ production performance of the pNOR-eNOₓ⁻RR tandem system with Ni(OH)ₓ/Cu cathode using air as the source of nitrogen (Fig. 5a). In pNOR, we applied a 220 V source (AC) to initiate a neon-sign transformer to output a high voltage upon the spark discharge NTP (Supplementary Figs. 48, 49). After switching on the discharge, the color of the gas mixture turned brown, indicating the generation of NO₂ (Supplementary Fig. 49c)[19]. The mass spectral signals of NO and NO₂ in the air are almost undetectable yet become stronger after the discharge, indicating that NO and NO₂ are indeed produced from the discharge process rather than from the feeding gas (Supplementary Fig. 50). Then, these produced NO and NO₂ can evolve into NO₂⁻ and NO₃⁻ in

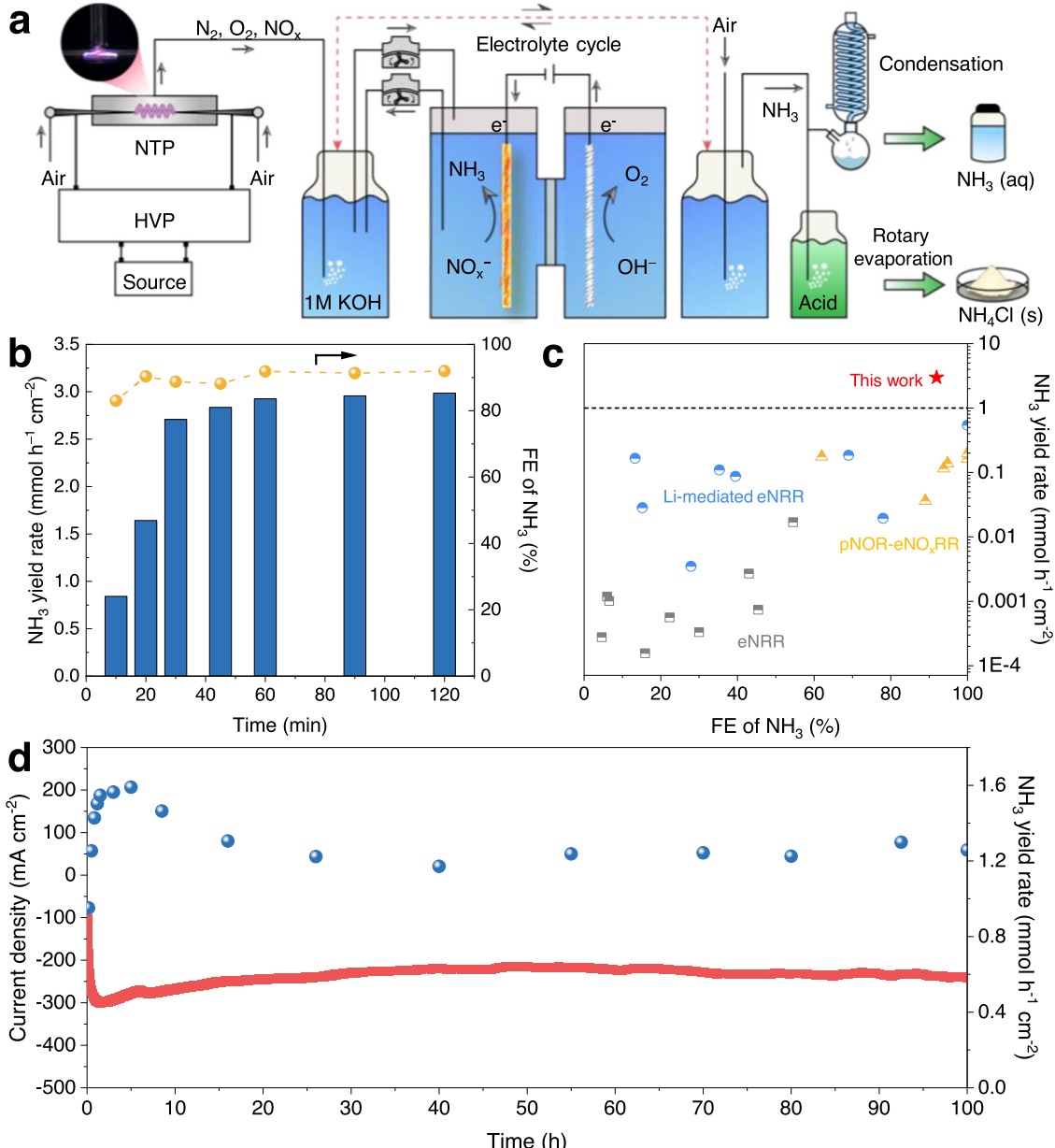

**Fig. 5 | Performance of pNOR-eNO$_x^-$RR system for air-to-NH$_3$ conversion.**
**a** Schematic of pNOR-eNO$_x^-$RR system and NH$_3$ products collection. **b** Discharge time-dependent NH$_3$ yield rate and FE over Ni(OH)$_x$/Cu at −0.25 V for 30 min of electrolysis. **c** Comparison of the NH$_3$ yield rate and FE of this work (red star) with the previous reports including eNRR (gray square), Li-mediated eNRR (blue sphere), and pNOR-eNO$_x^-$RR (yellow triangle). **d** A continuous run of the pNOR-eNO$_x^-$RR system at −0.25 V in a flow mode using pure air as the feeding gas.

the alkaline electrolyte through reactions as Eqs. (1, 2).

$$NO + NO_2 + 2OH^- \rightarrow 2NO_2^- + H_2O \qquad (1)$$

$$2NO_2 + 2OH^- \rightarrow NO_2^- + NO_3^- + H_2O \qquad (2)$$

The spark discharge conditions of pNOR, such as the ratio of air to O$_2$, discharge distance, and gas flow rate, were then optimized (Supplementary Fig. 51). In addition, the pNOR process is stable for linearly producing NO$_x^-$ (Supplementary Fig. 52). The failure of NH$_3$ production by running either pNOR or eNO$_x^-$RR alone highlights the necessity of simultaneous operation of these two parts for NH$_3$ synthesis (Supplementary Fig. 53). Moreover, using Ar as the feeding gas leads to no NH$_3$ generation,

confirming that N$_2$ in the air is the N source. We then performed batch experiments to investigate the impact of the spark discharge time on the NH$_3$ yield rate and corresponding FE (Supplementary Fig. 54). The solutions with different spark discharge time were applied as the afterward electrolyte. As shown in Fig. 5b, the NH$_3$ yield increases rapidly as the discharge time increases from 10 to 30 min. Further expanding the discharge time to 120 min only leads to a small increment of NH$_3$ yield rate from 2.7 to 3.0 mmol h$^{-1}$ cm$^{-2}$. The FEs of NH$_3$ scatter around 90% when the discharge time exceeds 20 min, and a smaller NH$_3$ FE of 83% is obtained under 10 min, maybe due to the competing HER reaction under low NO$_x^-$ concentration. The NH$_3$ production efficiency of our pNOR-eNO$_x^-$RR system surpasses other electrochemical NH$_3$ synthesis alternatives such as eNRR, Li-mediated eNRR, and other reported pNOR-eNO$_x^-$RR works (Fig. 5c and Supplementary Table 3).

Even with pure air for pNOR, the $NO_x^-$ generation rate (4.05 mmol h$^{-1}$) was still greater than the maximum $NO_x^-$ consumption rate (3 mmol h$^{-1}$) achievable with a 1-cm$^2$ Ni(OH)$_x$/Cu, so we conducted a 100-h test in a flow mode with simultaneous initiation of NTP and electrolysis using a pure air inlet (Supplementary Fig. 55). Figure 5d shows that in the initial stage, the current density and NH$_3$ yield increases with the constant accumulation of $NO_x^-$ (Supplementary Fig. 56). Although the charge-transfer resistance of the electrode in electrochemical impedance spectroscopies decreased gradually due to the increasing concentration of $NO_x^-$, the increasing solution resistance after 5 h of operation limited the increase in current density (Supplementary Fig. 57). Considering that the pH barely changes during the 100-h operation, the increasing solution resistance mainly be caused by the accumulation of $NO_x^-$ with lower mole conductivities ($NO_3^-$: 71.44 S cm$^2$ mol$^{-1}$, $NO_2^-$: 71.7 S cm$^2$ mol$^{-1}$) than that of OH$^-$ (198.0 S cm$^2$ mol$^{-1}$) (Supplementary Figs. 58, 59). Eventually, the whole system delivers a steady current density of ca. 240 mA cm$^{-2}$ and NH$_3$ production rate of ca. 1.25 mmol h$^{-1}$ cm$^{-2}$, outperforming the results obtained over the Cu electrode (Supplementary Fig. 60). Note that the FE for NH$_3$ production cannot be determined because the amounts of $NO_2^-$ and $NO_3^-$ under continuous reaction are imponderable. Our pNOR-eNO$_x^-$RR tandem system delivers an overall energy consumption of 18.36 MJ mol$^{-1}$ NH$_3$ and energy efficiency of 2.08%, which is comparable to a previous work by combining a non-thermal plasma bubble column reactor and Cu nanowire-catalyzed electrochemical conversion and much lower than a plasma-activated proton conducting solid oxide electrolyzer for NH$_3$ synthesis (605 MJ mol$^{-1}$ NH$_3$) (Supplementary Note 1)[20,46]. Although this energy consumption value is still larger than that of the Haber-Bosch process (0.52−0.81 MJ mol$^{-1}$ NH$_3$), the application of the pNOR-eNO$_x^-$RR system can provide a new option for the development of decentralized NH$_3$ synthesis, considering that the traditional Haber-Bosch process is only economically viable in large-scale production[47].

We then applied an air stripping method to collect the NH$_3$ product from the electrolyte and absorption solution due to the high NH$_3$ vaper pressure in the alkaline media[48]. The stripped-out NH$_3$ was either condensed into a concentrated NH$_3$ aqueous solution or trapped by the acid solution to collect solid NH$_4$Cl after performing rotary evaporation. Eventually, 69.8% and 23% of the produced NH$_3$ was collected as NH$_4$Cl (4.6956 g) and NH$_3$ aqueous solution (63.8 mL, 0.77 wt%), respectively (Supplementary Fig. 61). The purity of liquid and solid products is confirmed by $^1$H NMR and XRD, respectively (Supplementary Fig. 62).

In summary, we present a tandem NH$_3$ synthesis system that combines a spark discharge NTP-enabled N$_2$-to-NO$_x$ conversion with electrocatalytic NO$_x^-$-to-NH$_3$ transformation. The pNOR-eNO$_x^-$RR system with the Ni(OH)$_x$/Cu cathode performs a high NH$_3$ yield of 3 mmol h$^{-1}$ cm$^{-2}$ and corresponding FE of 92% using air and H$_2$O as the sources of nitrogen and proton, respectively. It also exhibits excellent stability for long-term continuous operation. Experimental and theoretical results reveal that Ni(OH)$_x$ species on the Cu surface interact with hydrated K$^+$ in the double layer, contributing to enhancing H$_2$O adsorption and dissociation to form active H$_{ad}$ at low overpotentials for the hydrogenation of nitrogenous intermediates on Cu. Meanwhile, Ni(OH)$_x$/Cu shows optimized intermediates adsorption behavior, making the reaction proceed in the direction of NH$_3$ formation. This work provides a perspective for the rational design of an electrocatalyst for eNO$_x^-$RR in alkaline conditions and offers one renewable-electricity-powered avenue for decentralized NH$_3$ mass production using air at ambient conditions.

## Methods
### Preparation of Ni(OH)$_x$/Cu NWA, Cu NWA, and Ni(OH)$_2$ catalysts
In a typical procedure, a piece of 1 × 3 cm$^2$ Cu foam was washed respectively with 1 M HCl, ethanol, and DI water for 10 min each to clean its surface. Then, the Cu foam was immersed in 30 mL aqueous solution containing 3 g NaOH and 0.68 g (NH$_4$)$_2$S$_2$O$_8$ for 20 min to obtain Cu(OH)$_2$ NWA on the Cu skeleton. The synthesized Cu(OH)$_2$ NWA was annealed in a muffle furnace at 200 °C for 3 h to convert it into a CuO NWA. Next, the CuO NWA was soaked in 0.1 M NiCl$_2$·6H$_2$O for 15, 30, and 45 min at open circuit potential to achieve varied surface coverage of hydroxide. After the deposition step, the electrode was rinsed with DI water thoroughly and dried at 60 °C in an oven. Finally, a cyclic voltammetry prereduction process was performed in the electrolyte from 0.2 V to −0.3 V for 10 cycles with a scan rate of 5 mV s$^{-1}$ to fully reduce CuO to Cu. Cu NWA electrode was synthesized by performing the same procedure except for the deposition of Ni(OH)$_x$. Ni(OH)$_2$ was synthesized via a solvothermal method according to the reference[49]. Typically, 0.7 g of Ni(NO$_3$)$_2$·6H$_2$O was dissolved into 48 mL of ethanol under magnetic stirring, followed by adding 4.8 mL of oleylamine and 24 mL of ethanol in sequence. After being stirred for 30 min, the mixture was transferred to a 120 mL Teflon-lined stainless autoclave and heated to 190 °C for 16 h. Finally, the precipitates were washed with cyclohexane, DI water, and ethanol thoroughly and dried at 60 °C under vacuum.

### Materials characterizations
The morphology was analyzed by field emission SEM (TESCAN MAIA3LMH) and TEM (Talos F200X). XRD patterns were recorded on a Shimadzu XRD-6100 with Cu Kα radiation. XPS spectra were collected on a Thermo Fisher ESCALAB Xi+ X-ray photoelectron spectrometer. All the XPS data were calibrated by shifting the C 1$s$ peaks to 284.8 eV. The Raman spectra were measured on a Renishaw inVia Qontor Ramna microscope using laser excitation wavelength of 633 nm for copper and copper oxide detection and 532 nm for Ni(OH)$_x$ species detection. The concentration of nitrate in the electrolyte was quantified on a Thermo Scientific Dionex Integrion. The EPR measurements were performed on a JEOL JES-FA200 spectrometer. The $^1$H NMR spectra were measured on a AVANCE III HD 600 MHz NMR spectrometer. The mass spectra were collected on a GCMS-QP2020NX Shimadzu instrument. The contents of Ni and Cu elements in the Ni(OH)$_x$/Cu samples were measured on a NexION 350D inductively coupled plasma mass spectrometer (ICP-MS).

### XAFS measurement
The Ni K-edge XAFS spectra were collected at the 1W1B beam line of Beijing Synchrotron Radiation Facility, employing transmission mode for Ni foil and Ni(OH)$_2$, while fluorescence mode was utilized for Ni(OH)$_x$/Cu. Data processing and analysis were conducted using the Athena and Artemis software from the Demeter data analysis packages, incorporating the FEFF6 program for EXAFS fitting[50,51]. Energy calibration was performed referencing both standard and Ni foil measurements concurrently. To prepare the data, a linear function was subtracted from the pre-edge region, followed by normalization of the edge jump with Athena software. Subsequently, $\chi(k)$ data isolation involved subtraction of a smooth, third-order polynomial approximating the absorption background of an isolated atom. Fourier transformation of the $k^3$-weighted $\chi(k)$ data followed, utilizing a Han-Feng window function ($\Delta k = 1.0$). For EXAFS modeling, global amplitude EXAFS parameters ($CN$, $R$, $\sigma^2$, and $\Delta E_0$) were derived via nonlinear fitting in Artemis software, incorporating least-squares refinement[52]. In the analysis of Ni(OH)$_2$ EXAFS, the obtained amplitude reduction factor $S_0^2$ value (0.954) was utilized to determine coordination numbers ($CNs$) in the Ni-O, Ni-Ni, and Ni-Cu scattering paths within the sample.

### Electrochemical measurements
All the electrochemical measurements in this study were conducted using a CH Instruments 660E Potentiostat at room temperature. Nitrate reduction was performed in a commercial gas-tight H-type cell separated by a Nafion 117 cation exchange membrane (Dupont). In a

typical three-electrode system, the synthesized Cu, Ni(OH)$_x$/Cu, and Ni(OH)$_2$ electrodes were used as the working electrode, while a graphite rod electrode and a Hg/HgO electrode were used as the counter and reference electrodes, respectively. For the Ni(OH)$_2$ electrode, a catalyst ink was prepared by mixing 2 mg of Ni(OH)$_2$ with 750 mL of isopropanol, 200 mL of distilled water, and 50 µL of Nafion solution (5 wt%). The mixture was sonicated for 30 min to form a homogeneous ink. Then, 100 µL of the catalyst ink was drop-cast onto a carbon paper electrode with an active area of $1 \times 1\,cm^2$. The synthesized Cu and Ni(OH)$_x$/Cu were directly applied as the working electrode. The copper foam electrodes were sealed with silicone rubber to ensure a geometric area of $1 \times 1\,cm^2$. In this work, the recorded potentials against Hg/HgO were converted to the RHE scale. It was calibrated by performing CV in the high purity hydrogen saturated 1 M KOH electrolyte with two Pt electrodes as the working and counter electrodes. CVs were run at a scan rate of $1\,mV\,s^{-1}$. The average value of the two potentials corresponding to zero current is taken as the thermodynamic potential of the hydrogen electrode reaction. 25 mL aqueous solution of 1 M KOH containing 0.1 M NO$_3^-$ was used as the electrolyte in both the cathode and anode chambers. Chronoamperometry tests were performed under different biases at a stirring rate of 400 revolutions per minute. Ar was injected into the electrolyte for 10 min before testing to expel the oxygen and continuously pumped into the electrochemical cell during the test. A 0.01 M H$_2$SO$_4$ absorption solution was needed to collect NH$_3$ in the exhaust gas since NH$_3$ is volatile in an alkaline solution. All the batch experiments were tested for 30 min, and the electrolyte and absorption solution were taken out for quantitative analysis. The linear scanning voltammetry (LSV) curves were collected at a scan rate of $5\,mV\,s^{-1}$. The long-term stability test was performed using the chronopotentiometry method at a current density of $400\,mA\,cm^{-2}$ for 25 h in a homemade flow cell system at an electrolyte flow rate of $5\,mL\,min^{-1}$. The electrochemical impedance spectroscopy was recorded with a frequency range of 0.1 Hz to 100 KHz and an amplitude of 5 mV.

The ECSA was calculated by Eq. (3).

$$ECSA = C_{dl}/C_s \tag{3}$$

where $C_{dl}$ is the double-layer capacitance, and $C_s$ is the specific capacitance. $C_{dl}$ was determined by the CV scanning at the non-Faradaic potential range at different scan rates. The slope of the plot of half the difference between anodic and cathodic currents versus scan rate was taken as $C_{dl}$. The $C_s$ value of 40 µF cm$^{-2}$ was used in this study.

## Ion concentration detection

**NH$_4^+$ quantification.** The yield of ammonia in the electrolyte was detected by the indophenol blue method. First, 2 mL of the diluted post-electrolysis electrolyte was mixed with 2 mL of 1 M NaOH solution containing 5 wt% salicylic acid and 5 wt% sodium citrate. Then, 1 mL of 0.05 M NaClO and 0.2 mL of 1 wt% sodium nitroferricyanide (C$_5$FeN$_6$Na$_2$O) were added. The mixture was shaken for 30 s and then placed in the dark for 2 h to complete the color reaction. The ammonia concentration was measured using the UV-vis spectrometer to determine the absorbance at 650 nm. To quantify the amount of NH$_3$, quantified NH$_4$Cl was dissolved in 1 M KOH to obtain a series of standard solutions. The calibration curve was obtained by linear fitting.

**NO$_2^-$ quantification.** A color reagent was made by adding 4.0 g of p-aminobenzenesulfonamide, 0.2 g of N-(1-naphthyl) ethylenediamine dihydrochloride, and 10 mL of phosphoric acid (85 wt% in water) into 50 mL of DI water. The electrolyte was first diluted to the detection range. After adding 0.1 mL of the color reagent, the mixture was rested for 20 min. The absorbance at 540 nm was recorded using the UV-vis spectrometer. A concentration-absorbance curve was calibrated using standard NaNO$_2$ solutions with a series of concentrations. The nitrite concentration was calculated based on the recorded absorbance and the calibration curve.

**NO$_3^-$ quantification.** The concentration of nitrate was quantitatively detected by ion chromatography. A concentration-peak area curve was calibrated using standard NaNO$_3$ solutions with a series of engagements. The nitrate concentration was calculated based on the recorded peak area and the calibration curve.

**NMR determination of ammonium.** The produced ammonium in isotope-labeling experiments was detected via $^1$H nuclear magnetic resonance (NMR, 600 MHz) using DMSO-$d_6$ as a solvent. Typically, the pH of the as-obtained solution was adjusted to 2 using 3 M HCl. Then, 0.4 mL of the solution was mixed with 50 µL of DMSO-$d_6$ and 50 µL of 10 mM maleic acid.

## EPR spectroscopy experiments

EPR spectra were obtained using 5,5-dimethyl-1-pyrroline-N-oxide (DMPO) as the H$_{ad}$ trapping reagent. Before testing, the electrolyte was saturated with Ar to avoid the oxidation of DMPO. Then, DMPO was added to the cathode electrolyte to reach a concentration of 30 mmol L$^{-1}$. Each EPR spectra was collected after 10 min electrolysis at −0.25 V vs. RHE under Ar-bubbling.

## Calculation of FE, partial current density, NH$_3$ yield rate, and selectivity

The FE of production of NH$_3$ or NO$_2^-$ was calculated with the Eq. (4).

$$FE = (C \times V \times n \times F)/Q \tag{4}$$

where C represents the NH$_3$ concentration in the electrolyte (mol L$^{-1}$), V is the electrolyte volume in the cathode chamber (L), F is the Faraday constant (96485 C mol$^{-1}$), Q is the total charge consumed on the electrode during electrolysis (C), n is the consumed electron number, which is 8 for producing one NH$_3$ and 2 for NO$_2^-$.

Considering that both NO$_2^-$ and NO$_3^-$ contributed to the NH$_3$ production in the pNOR-eNO$_x^-$RR system, the FE for NH$_3$ production was calculated by Eq. (5).

$$FE = (6F \times C_1 + 8F \times C_2) \times V/Q \tag{5}$$

where $C_1$ and $C_2$ were obtained by Eqs. (6, 7).

$$C_1 = C \times \Delta C_{NO_2^-}/(\Delta C_{NO_2^-} + \Delta C_{NO_3^-}) \tag{6}$$

$$C_2 = C \times \Delta C_{NO_3^-}/(\Delta C_{NO_2^-} + \Delta C_{NO_3^-}) \tag{7}$$

The partial current density for producing NH$_3$ was calculated by Eq. (8).

$$j_{NH_3} = (Q \times FE_{NH_3})/(A \times t) \tag{8}$$

where A is the electrode geometric area, and t is the electrolysis time.

The ammonia yield rate was calculated by Eq. (9).

$$Y_{NH_3} = (C \times V)/(A \times t) \tag{9}$$

The selectivity of consumed NO$_3^-$ to NH$_3$ was calculated by Eq. (10).

$$S_{NH_3} = C_{ammonia}/\Delta C_{nitrate} \tag{10}$$

where $C_{ammonia}$ is the concentration of generated NH$_3$ and $\Delta C_{nitrate}$ is the concentration change of NO$_3^-$.

## In situ Raman and in situ ATR-SEIRAS

In situ Raman spectra were conducted on a Renishaw inVia Qontor Ramna microscope using a 633 nm solid laser as an excitation source. The measurement was carried out in a homemade reactor with the glassy carbon electrode, Hg/HgO, and Pt wire as the working electrode, reference electrode, and counter electrode, respectively. The catalyst ink was obtained by mixing isopropanol, water, and Nafion solution, and the catalyst stripped by ultrasonic on a copper foam electrode. The working electrode was prepared by drop-casting the catalyst ink onto the glassy carbon electrode. The electrolyte of 1 M KOH and 0.1 M KNO$_3$ was pumped with Ar for 30 min before testing. Raman spectra were collected under different potentials. Each potential was applied for at least 5 min before spectra collection to ensure a steady-state condition of the catalyst surface. During the test, the objective was protected by a Teflon film and immersed in the electrolyte. Water was filled into the gap between the objective and the Teflon protection film to avoid the interference of the air bubble.

In situ ATR-SEIRAS was performed on Thermo-Fisher Nicolet iS50 equipped with a liquid nitrogen-cooled HgCdTe (MCT) detector and VeeMax III ATR accessory. A silicon semi-cylindrical prism coated with gold and catalyst was used as the working electrode. The Hg/HgO electrode and Pt wire were used as the reference and counter electrodes, respectively. ATR-SEIRAS was recorded by stepwise switching the potential from 0.2 V to −0.4 V (vs. RHE). The spectra collected at open circuit voltage were used for the background subtraction. All ATR-SEIRAS measurements were acquired by averaging 64 scans at a spectral resolution of 4 cm$^{-1}$.

## pNOR-eNO$_x^-$RR continuous test and NH$_3$ products collection

The spark discharge NTP was generated by a high voltage power supply (HB-C06, Foshan Hongba Electronics Co., Ltd, China), enabled by a manual contacting voltage regulator (TDGC2-0.5KVA, Zhejiang Chengqiang Electric Co., Ltd, China). The spark discharge NTP reactor consisted of a "T"-shaped glass tube with an inner diameter of 2 mm and two stainless-steel tubes with an inner diameter of 0.5 mm for gas inlet and plasma generation. Pure air was used as the feeding gas. 300 mL and 1000 mL of 1 M KOH were used as electrolytes in the cathode and anode chambers, respectively. The produced NO$_x$ was adsorbed in an external electrolyte storage tank, in which the electrolyte was circulated through a peristaltic pump. Two adsorption tanks each filled with 100 mL of 0.2 M HCl were applied at the gas outlet to collect volatile NH$_3$. The pNOR-eNO$_x^-$RR continuous test was initiated by running the NTP and electrocatalysis simultaneously. After the 100-h continuous test, 0.5 mol KOH was added in the mixture of the electrolyte and the adsorption solution to increase its alkalinity. The NH$_3$ was separated by air-stripping at a gas flow rate of 30 sccm for 12 h in an oil bath of 70 °C. To obtain solid NH$_4$Cl product, the gas outlet stream was purged into 200 mL of 1 M HCl, which was then treated with a rotary evaporator to collect the NH$_4$Cl powder. To obtain a concentrated NH$_3$ aqueous solution, the NH$_3$ vapor in the gas outlet stream was condensed into NH$_3$ (aq) through a graham condenser.

## DFT calculation

The first-principles calculations were carried out with DFT implemented in the Vienna ab-initio simulation package (VASP). The generalized gradient approximation (GGA) with the Perdew-Burke-Ernzerhof (PBE) exchange-correlation functional, the projector augmented wave (PAW) pseudopotential for the core electrons, and a plane-wave basis with the kinetic cut-off energy of 500 eV was applied in all calculations. We performed Brillouin-zone integrations using Monkhorst-Pack grids with (4 × 6 × 1) mesh for the structure optimization. The energy (converged to $1.0 \times 10^{-5}$ eV atom$^{-1}$) and force (converged to −0.05 eV Å$^{-1}$) were set as the convergence criterion for geometry optimization. A vacuum space of 20 Å along the Z direction

was used to separate the interaction between the neighboring slabs. Cu was represented by (1 1 1) terminated slab because it represents the low-energy surface[53]. Three molecular units of Ni(OH)$_2$ were deposited on the Cu (1 1 1) surface and optimized. We used the Hubbard U correction in the implementation of Dudarev et al. with a U value of 6.2 for Ni[54]. To assess the role of alkali cations, we introduce in the simulation cell with solvated alkali cations (with 4, 5, and 7 H$_2$O in Li$^+$, Na$^+$, and K$^+$ coordination shell, respectively), fixing their z-coordinate to distances of 6.8, 7.2, and 5.5 Å from the Cu surface and 6.4, 6.9, and 4.9 Å from the Ni(OH)$_x$/Cu surface[55,56]. The adsorption energy of reaction intermediates was defined as:

$$E_{ads} = E_{M-Sub} - E_{Sub} - E_M \qquad (11)$$

where E$_{M-Sub}$, E$_{Sub}$, and E$_M$ represent the energies of an adsorbed system, a clean substrate, and an adsorbate, respectively. The Gibbs free energies variations were calculated by:

$$\Delta G = E_{ads} + \Delta ZPE - T\Delta S \qquad (12)$$

where ΔZPE and ΔS represent the change of zero point energy and entropy, respectively.

The H$_2$O dissociation energy was calculated by:

$$E_{dissociation} = E_{H_2O^*} - E_{H^* + OH^*} \qquad (13)$$

where * designates a surface adsorbed specie.

Given that it is difficult to directly calculate the energy of charged NO$_3^-$, the adsorption free energy of NO$_3^-$ (ΔG(*NO$_3$)) was calculated with the assistance of the gaseous HNO$_3$ (Supplementary Fig. 63 and Supplementary Note 2)[57,58].

## Molecular dynamics (MD) simulation

MD simulation boxes with dimensions of around 40 × 40 × 100 Å$^3$ are created, in which the solid substrate is the Cu or the Ni(OH)$_x$/Cu, while 1.1 M Li$^+$, 1.1 M Na$^+$, 1.1 M K$^+$, or 1.1 M Cs$^+$ aqueous solution is placed on the solid crystal for comparison. Periodic boundary conditions (PBCs) are imposed in the two orthogonal (x and y) directions to mimic infinite planar Cu crystalline substrate, while a wall-boundary condition is applied in the out-of-plane (z) direction of the substrate. The forcefield parameters of as-investigated systems are taken from the literature[59]. For the non-bonded atomic interactions in the system, the 12-6 Lennard-Jones potential with a cutoff distance of 10.0 Å is applied to describe the van der Waals (vdW) forces between atoms, while the standard Coulomb potential is utilized to mimic the electrostatic interactions that is evaluated by the particle–particle particle–mesh (PPPM) algorithm. To satisfy an imposed voltage of −0.25 V vs. RHE across the systems along the z-direction, the charges of each Cu atom are computed at each timestep using the constant potential fix in the Large-scale Atomic/Molecular Massively Parallel Simulator (LAMMPS)[60]. Prior to MD simulations, energy minimizations are firstly performed to relax the configuration of as-investigated systems with energy and force tolerances of 0.0001 Kcal/mol and 0.0001 Kcal/(mol·Å), respectively. Then, MD simulations with 1,000,000 timesteps are carried out to further relax the systems at temperature of 300 K under canonical (NVT) ensemble, in which the temperature is maintained by the Nose-hoover thermostat. Finally, production MD simulations with 1,000,000 timesteps are performed to capture the structural behaviors of ions and water in the vicinity of the surface of Cu-based substrates. During the whole MD simulations, the Cu-based substrates is frozen. The dynamics of atoms in the solution are based on the classical Newton's motion, in which the velocity-Verlet algorithm with a timestep of 1.0 fs is applied to integrate the classic Newton's equation. All the MD simulations are implemented using the LAMMPS package.

## Data availability

The data supporting the plots within this paper and other study findings are available from the corresponding author upon request. Source data are provided with this paper.

## Code availability

The codes that support the findings of this study are available from the corresponding author on request.

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

## Acknowledgements

This work was supported by the National Key R&D Program of China (2020YFA0710000, G.Y.), Joint Funds of the National Natural Science Foundation of China (U22A20391, G.Y.), National Natural Science Foundation of China (Grant No. 22078256, G.Y.), Innovation Capability Support Program of Shaanxi (NO. 2023-CX-TD-26, G.Y.), and the Programme of Introducing Talents of Discipline to Universities (B23025, G.Y.).

## Author contributions

W.L. designed and carried out the synthesis, characterizations and catalytic reactions, analyzed the data, and wrote the original draft. M.X. performed the DFT calculations. C.Z. carried out catalytic reactions. B.C. designed and drew the schematic diagrams. J.C. optimized the non-thermal plasma working conditions. H.L. and H.O. reviewed and revised the manuscript. G.Y. contributed significantly to the analysis of the data, supervision of the project, manuscript preparation, and funding acquisition. All the authors commented on the manuscript and have given approval for the final version of the manuscript.

## Competing interests

The authors declare no competing interests.
