## [Peer Review File · Nature Communications]

REVIEWER COMMENTS

Reviewer #1 (Remarks to the Author):

Authors thoroughly investigated coupling non-thermal plasma enabled N₂ oxidation with Ni(OH)_x/Cu-catalyzed electrochemical NO_x- reduction. They tested that Ni(OH)_x/Cu can intrinsically convert NO₃⁻ towards NH₃ and also the impact from cations on the FE of NH₃ on Ni(OH)_x/Cu catalysts. They also did evaluation of the performance of air-to-NH₃ conversion in proposed tandem pNOR-eNO_x-RR system. All there are very nice. However, there are some critical questions or suggestions for this manuscript from my aspect, as follows.

Major comments:

(1) Even though many aspects related to conversion of NO₃⁻ towards NH₃ has been investigated on Ni(OH)_x/Cu, still some insights are missing.

i) Is Ni(OH)_x really on top of Cu contributes to the NO₃⁻ reduction, what about NiO or even NiCu alloy (oxides) contribute to the increase of NH₃ formation? How does the catalytic active site look like really needs more attention.

ii) As for cation effect, it is very interesting. Independently if more detailed investigation is included, it can be full another story. That would be great if more insightful explanation, like DFT is provided, like this has been done on CO₂/CO reduction on transition metals, of course, in order to do this, detailed active sites is demanded.

Overall, it feels authors wanna do all the tests/ examinations they can do but for some aspects they lack of systematical insight investigation, like the real active center: Ni(OH)_x/Cu, or NiO/Cu or even NiCu, like also the cations effects on NH₃ FE.

(2) In the study, for the impact from cations (Li⁺, Na⁺, K⁺, Cs⁺), authors claimed 'The M⁺(H₂O)_n (where n = 22, 13, 7, and 6 for Li⁺, Na⁺, K⁺, and Cs⁺, respectively) with smaller n and radii is believed to have a stronger interaction with anionic species and thus a greater ability to activate H₂O'. From this, Cs⁺ and K⁺ should have similar effect on the NO₃⁻ reduction. however, it seems for Cs⁺, it doesn't follow the trend and actually deactivates the NH₃ formation, what is the possible explanation for this?

(3) In DFT section, the construction of Ni(OH)_x/Cu model is a bit arbitrary. Whether it is the real active center is questionable. For now, the DFT results presented in the paper can't sufficiently provide reliable information to support experimental observations.

(4) It has been reported that Cu is not a good catalyst to catalyze NO₃⁻ towards NH₃. At certain potential, it prefers NO₂⁻ production and then contributes low FE for NH₃ formation. The whole paper addresses the enhanced H adsorption on Ni(OH)_x/Cu compared to Cu but little attention has been paid to the presence of Ni metal or oxides or hydroxides can promote NO₃⁻ towards NH₃ via NO₂⁻ efficiently which Cu can not.

Minor comments

(1) In Figure 2f, sentence: 'The FEs of NH₃ exceed 90% within a NO₃⁻ concentration range from 5 mM to 100 mM while drop to 80% at 1 mM because the competing HER may become more significant at low NO₃⁻ concentration.' may not be appropriate since with NO₃⁻ at 50 mM, the FE is also around 80%. This

makes the concentration influence on FE of NH₃ questionable. More consideration may be required to explain why at 50 mM NO₃⁻, the FE gets lower compared to other concentrations.

(2) In supporting information, Figure S27 what is the NO₃⁻ concentration? Like 100mM? Besides what is the potential, -0.30 V vs. RHE? If the concentration is 0.1M and the applied potential is -0.30 V vs. RHE? Is the figure 2b is one of the 15 tests in Figure S27? However, at figure S27 there is no NH₃ yield rate around 3 mmol/h/cm². This needs some explanation. The average NH₃ yield looks like around 2.5 mmol/h/cm². Then back to supplementary figure 23f, it would be more appropriate to provide error bar for the normalized NH₃ yield rate since you have 15 independent tests. With this, if we consider the average normalized NH₃ yield rate and compare it with that on Cu, the difference between their intrinsic catalytic might be much closer than what you presented in supplementary figure 23f.

I) As a result, I would suggest the author to include the error bar for normalized NH₃ yield rate both for Cu and Ni(OH)_x/Cu. This should give a better understanding about the intrinsic catalytic activity difference between Cu and Ni(OH)_x/Cu.

II) I would also suggest replacing figure 2b with supplementary 23f since the latter provides 'real' insights for the catalytic activity difference.

(3) Sentences: "However, the impact of the alkali cation over Cu is very limited due to its weak interaction with hydrated cation. The results illustrate that the Ni(OH)_x species boosts the NH₃ formation on the Cu surface via interaction with hydrated K⁺ and accelerated H₂O dissociation." is a bit confusing, Please rewrite it like: even though the impact of the alkali cation over Cu is very limited due to its weak interaction with hydrated cation, The results illustrate that the Ni(OH)_x species boosts the NH₃ formation on the Cu surface via interaction with hydrated K⁺

(4) If it is possible to calculate, what is the overall energy efficiency on the tandem non-thermal plasma and electrocatalysis setup? For application, it deserves to be considered especially, authors here compared their system to the Li-mediated the N₂ conversion in Figure 5c

Reviewer #2 (Remarks to the Author):

This manuscript presented a tandem non-thermal plasma and electrocatalysis (pNOR-eNO_x-RR) NH₃ synthesis system over Ni(OH)_x/Cu foam catalyst. And a NH₃ yield rate of 3 mmol h⁻¹ cm⁻² and Faradaic efficiency of 92% at -0.25 V versus reversible hydrogen electrode were obtained. Besides, a 100-hour continuous test was carried out with a steady 1.3 mmol h⁻¹ cm⁻² NH₃ yield rate. The author illustrated the catalytic mechanism of Ni(OH)_x/Cu via in situ spectroscopies and density functional theory. But some of the data analysis and description in this manuscript are inaccurate or paradoxical. Based of all, I cannot recommend this manuscript for the publication in this journal.

1. XRD patterns demonstrated that the presence of Cu₂O in the Ni(OH)_x/Cu before CV prerreduction (Supplementary Figure 6), but Cu LMM Auger spectra did not detect Cu⁺ before CV rereduction (Supplementary Figure 7), why?

2. In Supplementary Figure 22, Ni(OH)_x/Cu electrode with 30 min deposition time showed optimum eNO₃-RR performance, why?

3. In Page 4, Right column, Lines 5-7, the author said that there is a strong synergy between Ni(OH)_x and

Cu upon Ni(OH)_x/Cu toward NH₃ production, some detail description should be provided to explain this synergy effect.

4. The feeding gas of pNOR is not pure air, but the mixture of N₂ and O₂ (Supplementary Figure 38), then the description of "ammonia synthesis from air" is inappropriate.

5. What is the energy consumption through pNOR-eNO_x-RR towards NH₃ synthesis?

6. The I-t curves of Cu and Ni(OH)_x/Cu under different applied potentials are suggested to be added in this manuscript.

7. In Supplementary Figure 44, the ¹H NMR spectra peak position of the concentrated NH₃ aqueous product is different from ¹⁴NH₄⁺ reference sample, why?

8. Supplementary Figure 39 showed that the NO₂⁻ is the main species during the spark discharge NTP, while the author investigated electrocatalytic NO₃⁻ reduction for NH₃ synthesis, why?

9. In Page 9, Left column, Lines 43-44, the whole system reached balance between consumption and production of NO_x after 20 h operation, then what is the concentration of NO_x⁻ after that?

10. Detailed test methods for the EPR measurement should be provided.

Reviewer #3 (Remarks to the Author):

The authors reported a tandem ammonia synthesis system, which integrated a pNOR device to activate N₂ in the form of NO_x⁻ followed by effective eNO_x-RR with Ni(OH)_x/Cu catalyst. The structural and catalytic function of the Ni(OH)_x/Cu catalyst was investigated in detail, and the tandem pNOR- eNO_x-RR device could run continuously and achieve ammonia synthesis. Some key details regarding the composition of the catalyst, electrochemical tests and the tandem pNOR- eNO_x-RR device worth more explanations, especially in the following aspects:

1. My biggest question is that eNO_x-RR catalysts based on a combination of Ni and Cu centers have been published for many times. For instance, Sargent et al. made a pioneer study on NiCu alloys to enhance the efficiency of eNO_x-RR for ammonia synthesis (J. Am. Chem. Soc. 2020, 142, 5702-5708), and recently, CuNi alloy nanoparticles on a Cu foil was synthesized by Zhou et al. via laser irradiation to achieve efficient eNO_x-RR (Energy Environ. Sci., 2023,16, 2991-3001). In this situation, I encourage the authors to state clear the advantage of the Ni(OH)_x/Cu catalyst in this work compared with reported eNO_x-RR systems with a combination of Ni and Cu catalytic centers.

2. According to the preparation of the catalyst, "Ni(OH)_x was deposited on the CuO NWA surface by rinsing CuO NWA into a NiCl₂ solution at open circuit potential" (the first paragraph in results and discussions). Here, the term "open circuit potential" is ambiguous, or at least not quantitative and precise enough like "+0.24 V vs. RHE", particularly given that the open circuit potential of an electrode can be variable, and the chemical state of Cu is highly susceptible to applied potentials (Supplementary Figure 1). Another issue related with catalyst preparation is that the relative amounts of Ni on Cu seem unclear. Especially, given that different rinsing times were used, the Ni loading on CuO NWA would vary significantly.

3. "The linear sweep voltammetry (LSV) polarization curves in Fig. 2a reveal that, in the presence of NO₃⁻, eNO₃-RR exhibits an earlier onset potential when compared to..." It looks obvious, but I think the term "onset potential" warrants a better definition, as different papers evaluate "onset potential" in different ways.

4. In Figure 3a and 3b, ΔE at certain potentials were used to visualize the difference of catalytic kinetics in H₂O and D₂O. Here, the question is that ΔE seems highly variable under different current densities, and why the authors chose 40 mA cm⁻² in HER and 200 mA cm⁻² in eNO₃-RR to evaluate ΔE brought by H/D difference? Why a larger ΔE was yielded in eNO₃-RR than HER?
5. The pNOR device in this work could run continuously for days at a stable current density of hundreds of mA cm⁻². I am very interested how many grams of NH₃ or NH₄Cl was eventually obtained? Based on Figure 5d, one would expect ~100 mmol cm⁻² ammonia after the 100-h test, however, only 100 mL 0.2 M HCl was used to absorb it in the gas outlet stream, which could only absorb 20 mmol ammonia. In this situation, will the inadequacy of acid sorbent cause the loss of ammonia obtained?
6. Whether Ni(OH)_x/CuO can be used as electrocatalytic catalyst for nitrate reduction of ammonia. If so, is the performance of Ni(OH)_x/CuO better or worse than that of Ni(OH)_x/Cu?
7. The authors claim that "The FEs of NH₃ exceed 90% within a NO₃⁻ concentration range from 5 mM to 100 mM ". However, according to Figure 2, the FEs of NH₃ at about 70 mM is similar to that at 1 mM, both of which are about 80%. The explanation given by the authors is that the competition HER is more significant due to the lower concentration of NO₃⁻ (1mM). So, what causes the FEs drop at 70 mM?
8. According to Figure S26b, no matter what NO₃⁻ concentration is, the current on the Ni(OH)_x/Cu electrode is not stable during half an hour of electrolysis, and the current density fluctuates by about 100 mA/cm². It seems that the Ni(OH)_x/Cu electrode is not very stable.
9. According to Figure 5d, during the 100-hour test, the ammonia yield and current density in the first 20 hours are not stable. The author should give the corresponding explanation.
10. Figure 5d and S43, compared with Ni(OH)_x/Cu electrode, the current density of the pNOR-eNO_x-RR system is more stable when the Cu electrode is used for long-term continuous testing.
11. High-resolution Ni 2p XPS spectra of Ni(OH)_x/CuO and Ni(OH)_x/Cu needs to be fitted.
12. Why did the NH₃ yield rate decrease from 3.0 to 1.3 mmol h⁻¹ cm⁻² when using the pNOR-eNO_x-RR system instead of the solutions with different spark discharge time as electrolyte.

Response to reviewers' comments:

We are grateful for the critical feedback and constructive suggestions provided by the reviewers. Their input is invaluable in improving the quality of our work. We have carefully revised our manuscript in response to their comments and highlighted the changes in yellow. Below, we provide point-to-point responses to the reviewers' comments. The original comments are presented in blue italics, while our responses are in black.

Reviewer #1

Authors thoroughly investigated coupling non-thermal plasma enabled N_2 oxidation with $Ni(OH)_x/Cu$ -catalyzed electrochemical NO_x^- reduction. They tested that $Ni(OH)_x/Cu$ can intrinsically convert NO_3^- towards NH_3 and also the impact from cations on the FE of NH_3 on $Ni(OH)_x/Cu$ catalysts. They also did evaluation of the performance of air-to- NH_3 conversion in proposed tandem pNOR-e NO_x^- RR system. All there are very nice. However, there are some critical questions or suggestions for this manuscript from my aspect, as follows.

Response: We appreciate the reviewer's praise of our work, as well as important suggestions which have substantially improved the quality of our manuscript. We have provided a detailed point-to-point response to your comments below.

Comment 1. *Even though many aspects related to conversion of NO_3^- towards NH_3 has been investigated on $Ni(OH)_x/Cu$, still some insights are missing. Is $Ni(OH)_x$ really on top of Cu contributes to the NO_3^- reduction, what about NiO or even NiCu alloy (oxides) contribute to the increase of NH_3 formation? How does the catalytic active site look like really needs more attention.*

Response: We employed various characterization techniques, including Raman spectroscopy, XPS, TEM, and XAS to confirm that the nickel species on the Cu surface in $Ni(OH)_x/Cu$ are amorphous $Ni(OH)_x$. Initially, we reduced the $Ni(OH)_x/CuO$ in a 1 M KOH electrolyte containing 0.1 M KNO_3 , which mirrors the conditions under which our e NO_3^- RR performance tests were conducted. Meanwhile, we controlled the range of CV for pre-reduction between -0.3 and 0.2 V vs. RHE, based on the Pourbaix diagrams, to ensure that $Ni(OH)_x$ would not be reduced to Ni. Consequently, the composition of $Ni(OH)_x/Cu$ catalysts obtained after pre-reduction should remain constant within the testing potential range from 0 to -0.3 V vs. RHE.

In the XPS spectra of both Ni(OH)_x/CuO and Ni(OH)_x/Cu, no zero-valent Ni was detected, indicating the absence of Ni or NiCu alloy. The Ni XANES edge position showed that the oxidation state of the Ni is approximately 2+, establishing NiO or Ni(OH)₂ as the most likely candidates. Additionally, a distinct peak appeared at a binding energy of 531.5 eV in the O 1s spectra, signifying the presence of nickel hydroxide species. Furthermore, in the Raman spectra of Ni(OH)_x/Cu, a characteristic signal of the symmetric O-H stretching mode of the free external -OH group was observed at a Raman shift of 3690 cm⁻¹, providing further evidence for the presence of nickel hydroxide species. EXAFS analysis revealed that the nearest neighbour Ni-O and Ni-Ni bond distances are 2.045±0.001 Å and 3.113±0.001 Å, respectively. These values closely match those of Ni-O and Ni-Ni bond distances in the Ni(OH)₂ standard (2.050±0.001 Å and 3.119±0.007 Å). Yet, the coordination numbers of Ni-O and Ni-Ni are smaller than in crystalline Ni(OH)₂, indicating that the Ni species in Ni(OH)_x/Cu electrodes exist in the form of amorphous Ni(OH)_x. These collectively affirmed that the nickel species on the surface of Cu are amorphous Ni(OH)_x, rather than metallic Ni, nickel oxide, or CuNi alloys.

The eNO₃⁻RR performance over the Cu surface is closely related to the supply of adsorbed hydrogen species, which are generated from the dissociation of water in an alkaline environment. Previous studies have shown Ni has slow kinetics for water dissociation, but favourable H adsorption, indicating that the adsorbed hydrogen species on its surface are more likely to combine and produce hydrogen gas. In addition, although the water dissociation on the surface of NiO is relatively fast, the O atoms on NiO have strong adsorption of H, which hinders the transfer of *H to the Cu surface [ACS Energy Lett. **2019**, 4, 3002–3010]. Compared with Ni and NiO, Ni(OH)_x species have specific advantages. Previous studies have shown that Ni(OH)₂ could be used as a universal promoter of alkaline HER on various metal surfaces. In a bifunctional mechanism formed in Ni(OH)₂ deposited catalysts, the edges of Ni(OH)₂ promote the dissociation of water and the production of hydrogen intermediates that then adsorb on nearby metal sites [Angew. Chem. Int. Ed. **2012**, 51, 12495–12498]. Hence, Ni(OH)_x/Cu has rapid dissociation kinetics for water molecules, and the adsorbed hydrogen species produced can effectively participate in the hydrogenation of N-containing intermediates on the Cu surface, leading to a superior eNO₃⁻RR performance.

Comment 2. *As for cation effect, it is very interesting. Independently if more detailed investigation is included, it can be full another story. That would be great if more insightful explanation, like DFT is provided, like this has been done on CO₂/CO*

reduction on transition metals, of course, in order to do this, detailed active sites is demanded.

Response: We are grateful to the reviewer for their constructive comment. We have conducted additional experiments and DFT calculations to provide more insightful explanations for this cation effect. We found that the solvated cations have a significant influence on water adsorption behaviour over the Ni(OH)_x/Cu.

Considering that there is a strong correlation between HER and eNO₃⁻RR performance over the Cu and Ni(OH)_x/Cu electrodes. We first evaluated HER performance of Cu and Ni(OH)_x/Cu in Ar-saturated 1 M MOH (M = Li, Na, K, and Cs). As shown in Figure R1-1, it can be observed that the HER performance Ni(OH)_x/Cu is highly dependent on the type of alkali metal cation, with their activities following the sequence of K⁺ > Na⁺ > Li⁺ > Cs⁺. However, the cation effect on the HER performance of Cu is insignificant. These trends are consistent with the effects of alkali metal cations on their corresponding eNO₃⁻RR performances.

Figure R1-1. HER polarization curves collected at room temperature on Cu (a) and Ni(OH)_x/Cu (b) in Ar-saturated 1 M MOH (M = Li, Na, K, and Cs) at a scan rate of 5 mV s⁻¹.

DFT simulations were then conducted to reveal a deeper insight. We referred to the computational methods used by Koper *et al.* in the studies of cation effects on HER and CO₂/CO reduction reactions [*J. Am. Chem. Soc.* **2023**, *145*, 19601-19610; *J. Am. Chem. Soc.* **2022**, *144*, 1589-1602; *ACS Energy Lett.* **2023**, *8*, 657-665]. First, we optimized models of various alkali metal cation hydrates with only the first hydration shell considered, wherein the coordination numbers of Li⁺, Na⁺, K⁺, and Cs⁺ were set as 4, 4, 7, and 6, respectively. The selection of these coordination numbers was based on *ab initio* molecular dynamics simulation results that the average number of water molecules around Li⁺, Na⁺, K⁺, and Cs⁺ is 3.9, 4.4, 6.6, and 6.2, respectively [*J. Mater. Chem. A*, 2020, *8*, 24428-24437]. Only the first hydration shell was calculated because

previous research found that cations adsorbed at active sites show a small hydration shell consisting of a single hydration layer [*Chem. Sci.*, **2022**, 13,7634–7643]. The optimized hydrate models of alkali metal cations of $\text{Li}^+(\text{H}_2\text{O})_4$, $\text{Na}^+(\text{H}_2\text{O})_4$, $\text{K}^+(\text{H}_2\text{O})_7$, and $\text{Cs}^+(\text{H}_2\text{O})_6$ are demonstrated in Figure R1-2. Given that hydrate cations predominantly accumulate in the outer Helmholtz plane (OHP) in the double layer [*Nat. Catal.* **2022**, 5, 923–933], we fixed cation hydrates 7 Å from the first layer of Cu atoms (Figure R1-3).

Figure R1-2. Hydrate models of different alkali metal cations: $\text{Li}^+(\text{H}_2\text{O})_4$ (a), $\text{Na}^+(\text{H}_2\text{O})_4$ (b), $\text{K}^+(\text{H}_2\text{O})_7$ (c), $\text{Cs}^+(\text{H}_2\text{O})_6$ (d).

Figure R1-3. Atomic configurations with different alkali metal cation hydrate fixed on Cu surface (a-d) and $\text{Ni}(\text{OH})_x/\text{Cu}$ surface (e-h).

Subsequently, we computed the Gibbs free energies for water adsorption ($\Delta G^*_{\text{H}_2\text{O}}$) on the Cu and $\text{Ni}(\text{OH})_x/\text{Cu}$ surfaces in the presence of different cation hydrates. As depicted in Figures R1-4 and R1-5, the $\Delta G^*_{\text{H}_2\text{O}}$ for $\text{Ni}(\text{OH})_x/\text{Cu}$ in the presence of hydrates of Li^+ , Na^+ , K^+ , and Cs^+ are -0.082 , -0.204 , -0.3 , and -0.04 eV, respectively. The tendency for water adsorption on $\text{Ni}(\text{OH})_x/\text{Cu}$ surface follows the trend $\text{K}^+ > \text{Na}^+ >$

$\text{Li}^+ > \text{Cs}^+$, which aligns well with its HER and eNO_3^- RR activities. The presence of K^+ hydrate is most conducive to the adsorption of water on $\text{Ni(OH)}_x/\text{Cu}$, thus can provide more adsorbed hydrogen species to hydrogenate N-containing intermediates on Cu, leading to the highest NH_3 yield rate. Li *et al.* also obtained the highest and the worst NH_3 yield rates in electrolytes containing K^+ and Cs^+ , respectively, in their proposed hydrogen radicals expedite nitrate-to-ammonia conversion [*J. Am. Chem. Soc.* **2020**, 142, 7036–7046]. Nevertheless, $\Delta G^*_{\text{H}_2\text{O}}$ for Cu with hydrates of Li^+ , Na^+ , K^+ , and Cs^+ are 0.086, -0.032 , -0.048 , and -0.097 eV, respectively. The variations of $\Delta G^*_{\text{H}_2\text{O}}$ upon Cu are smaller compared with those upon $\text{Ni(OH)}_x/\text{Cu}$. Correspondingly, the change in the NH_3 yield rate was very limited over the Cu catalyst by changing the alkali metal cation. These results indicate $\Delta G^*_{\text{H}_2\text{O}}$ can be considered a crucial indicator for the observed cation-dependency of the eNO_3^- RR performance over $\text{Ni(OH)}_x/\text{Cu}$.

Using a combination of theoretical and experimental investigations, we show the cation hydration and its resultant impact on the water adsorption behaviour to be the critical factor behind the ion specificity of eNO_3^- RR on $\text{Ni(OH)}_x/\text{Cu}$ electrode. We have supplemented Figures R1-1, R1-2, and R1-4 in the revised Supporting Information and Figure R1-5 in the revised Manuscript with a further discussion.

Figure R1-4. Atomic configurations of H_2O adsorption on Cu (a-d) and $\text{Ni(OH)}_x/\text{Cu}$ (e-h) with the presence of different alkali metal cation hydrates.

Figure R1-5. Gibbs free energy diagram of H₂O adsorption on Cu and Ni(OH)_x/Cu surface in the presence of different alkali metal cation hydrates.

Comment 3. *In the study, for the impact from cations (Li⁺, Na⁺, K⁺, Cs⁺), authors claimed ‘The M⁺(H₂O)_n (where n = 22, 13, 7, and 6 for Li⁺, Na⁺, K⁺, and Cs⁺, respectively) with smaller n and radii is believed to have a stronger interaction with anionic species and thus a greater ability to activate H₂O’. From this, Cs⁺ and K⁺ should have similar effect on the NO₃⁻ reduction. however, it seems for Cs⁺, it doesn’t follow the trend and actually deactivates the NH₃ formation, what is the possible explanation for this?*

Response: Thank you very much for your valuable feedback. The text in our previous manuscript mentions that Li⁺ and Na⁺ possess 22 and 13 hydration numbers, respectively, which represent the total water molecules in all of their bulk hydration shells. While cations adsorbed at active sites show an opposing cation-dependent solvation trend and indicate an overall smaller hydration shell consisting of a single hydration layer [*Chem. Sci.*, **2022**, 13,7634–7643]. Therefore, we should consider the first hydration shell more in the actual reaction.

On the one hand, our theoretical calculations indicate that Cs⁺ hydrates are detrimental to the adsorption of water molecules on the Ni(OH)_x/Cu surface (Figure R1-5). Consequently, in the presence of Cs⁺, eNO₃⁻RR activity will be constrained by a shortage of adsorbed active hydrogen species. On the other hand, Koper *et al.* found that at high pH and consequently higher near-surface cation concentrations, the accumulation of Cs⁺ at the OHP can inhibit HER [*ACS Catal.* **2021**, 11, 14328–14335]. This is because Cs⁺ may show a stronger interaction and direct adsorption on the electrode surface due to its less tightly bound solvation shell [*JACS Au* **2021**, 1, 1674–1687]. Experimentally, Duan *et al.* observed the direct adsorption of Cs⁺ on the electrode surface in their preliminary electrical transport spectroscopy studies [*Nat. Catal.* **2022**, 5, 923–933]. In our experimental conditions of eNO₃⁻RR, the

concentration of alkali metal cations is very high (1 M CsOH plus 0.1 M CsNO₃), potentially leading to direct adsorption of Cs⁺ on the catalyst surface, thereby poisoning active sites, blocking the access of water molecules and NO₃⁻ to the catalyst surface, thus leading to a decreased eNO₃⁻RR performance. We have added a new discussion in the revised manuscript.

Comment 4. *In DFT section, the construction of Ni(OH)_x/Cu model is a bit arbitrary. Whether it is the real active center is questionable. For now, the DFT results presented in the paper can't sufficiently provide reliable information to support experimental observations.*

Response: The construction of Ni(OH)_x/Cu model is mainly based on our experimental observations. Firstly, for the construction of the Cu surface, we selected the Cu (1 1 1) crystal facet. This choice is supported by a strong diffraction peak at 43.3° in the XRD spectrum of the Ni(OH)_x/Cu electrode, corresponding to the Cu (1 1 1) crystal facet. Additionally, in the HRTEM image, a lattice spacing of 0.208 nm was observed, confirming the exposure of the Cu (1 1 1) facet. Given that numerous studies have demonstrated Cu (1 1 1) as the energetically favoured surface, we opted for the (1 1 1) terminated slab to represent Cu in our Ni(OH)_x/Cu model. Furthermore, for the construction of the Ni(OH)_x structure, we placed three molecular units of Ni(OH)₂ on the Cu (1 1 1) surface. Considering the amorphous nature of Ni(OH)_x in Ni(OH)_x/Cu, utilizing a cluster model enables the simulation of the material's characteristic of long-range disorder. Moreover, the coordination numbers of Ni-O, Ni-Cu, and Ni-Ni in the Ni(OH)_x/Cu model are approximately 3.7, 1, and 2, respectively. These values closely align with the coordination numbers of 4.6±0.3, 2.2±0.3, and 4.8±0.7 obtained from EXAFS analysis. Importantly, this model effectively captures the feature of unsaturated coordination in the Ni(OH)_x structure. It is worth noting that as for the construction of amorphous structures in computational models, similar strategies have been employed in prior studies. For instance, Sargent *et al.* constructed a CeO₂ model with two molecular units on a Cu (1 1 1) surface for DFT calculations to represent experimentally obtained 30 nm amorphous Ce(OH)_x nanoparticles loaded on Cu (Figure R1-6a, b) [*Nat. Commun.* **2019**, 10, 5814]. Similarly, in their another work, they placed two molecular units of Cr₂O₃ on a Cu (1 1 1) surface to represent deposited CrO_x with a size of 10 nm (Figure R1-6c, d) [*Nat. Energy* **2019**, 4, 107–114]. Thus, the model construction for Ni(OH)_x/Cu effectively represents both the Cu surface and the amorphous Ni(OH)_x structure.

Figure R1-6. TEM images and the corresponding atomic configurations for DFT calculation of CeO_2/Cu (a, b) and CrO_x/Cu (c, d). Adopted from *Nat. Commun.* **2019**, 10, 5814 and *Nat. Energy* **2019**, 4, 107–114, respectively.

Table R1-1. ICP-MS data for the $\text{Ni}(\text{OH})_x/\text{Cu}$ nanowires that ultrasonic peeling from the Cu skeleton and $\text{Ni}(\text{OH})_x/\text{Cu}$ bulk electrodes with different deposition times.

Samples	Deposition time / min	Measured mass of Ni / mg	Measured mass of Cu / mg	$\text{Ni}(\text{OH})_x / \text{Cu}$ mass ratio / % ^a
$\text{Ni}(\text{OH})_x/\text{Cu}$ nanowire	15	0.0351	3.99	1.27
	30	0.0902	5.67	2.30
	45	0.1430	6.23	3.31
$\text{Ni}(\text{OH})_x/\text{Cu}$ bulk electrode	15	0.0378	14.00	0.39
	30	0.0474	10.70	0.64
	45	0.0578	11.00	0.76

^aThe x value is estimated to be 1.53 based on the EXAFS analysis.

From inductively coupled plasma mass spectrometer (ICP-MS) results (Table R1-1), it is evident that the relative content of $\text{Ni}(\text{OH})_x$ in $\text{Ni}(\text{OH})_x/\text{Cu}$ increases gradually with deposition time. Simultaneously, the HER and eNO_3^- RR performance of the $\text{Ni}(\text{OH})_x/\text{Cu}$ electrode initially improves and then decreases with the deposition amount of $\text{Ni}(\text{OH})_x$. This suggests a significant role played by the $\text{Ni}(\text{OH})_x$ -Cu interface in determining HER and eNO_3^- RR activities. Additionally, after ECSA normalization, $\text{Ni}(\text{OH})_x/\text{Cu}$ exhibits higher intrinsic activity than Cu and $\text{Ni}(\text{OH})_2$. This further confirms the active sites are at the interface of $\text{Ni}(\text{OH})_x$ and Cu.

Our DFT results indicate that the adsorption and dissociation of water molecules on the $\text{Ni}(\text{OH})_x/\text{Cu}$ surface are more favorable than on the Cu surface, aligning well with

the experimentally observed enhanced HER performance on Ni(OH)_x/Cu. Additionally, the DFT calculations also show that the introduction of Ni(OH)_x clusters enhances the adsorption of NO₃⁻ on the Cu surface. Correspondingly, *in situ* Raman spectra of Cu and Ni(OH)_x/Cu in eNO₃⁻RR also reveal a stronger *NO₃ signal on the Ni(OH)_x/Cu surface. Therefore, the results of DFT calculations coincide well with the experimental phenomena observed in our study.

Comment 5. *It has been reported that Cu is not a good catalyst to catalyze NO₃⁻ towards NH₃. At certain potential, it prefers NO₂⁻ production and then contributes low FE for NH₃ formation. The whole paper addresses the enhanced H adsorption on Ni(OH)_x/Cu compared to Cu but little attention has been paid to the presence of Ni metal or oxides or hydroxides can promote NO₃⁻ towards NH₃ via NO₂⁻ efficiently which Cu can not.*

Response: We acknowledge the valuable insights provided by the reviewer and have taken their concern into full consideration. During the reduction of NO₃⁻, it is indeed found that Cu facilitates the formation of NO₂⁻ more readily at a lower overpotential (prior to the hydrogen evolution potential). Our experimental results also revealed that within the potential range from -0.3 to 0 V (vs. RHE), FEs of NH₃ on Cu are no more than 60%. In order to address this challenge, a number of researchers have pursued the integration of alternative constituents displaying superior intrinsic activity for NO₂⁻ reduction. This strategy is designed to establish tandem catalysis, thereby augmenting the efficiency of NH₃ production [*Nat. Commun.* **2022**, 13, 1129]. It can be confirmed that there is no metallic Ni and nickel oxide present in Ni(OH)_x/Cu, but rather amorphous Ni(OH)_x species. Therefore, in this regard, we can determine whether Ni(OH)_x/Cu can form tandem catalysis by comparing the intrinsic activity of Ni(OH)_x and Cu towards eNO₂⁻RR.

In the endeavor to scrutinize the intrinsic activity of Ni(OH)_x and Cu in NO₂⁻ reduction towards NH₃, we conducted potential-dependent assessments of NO₂⁻ reduction performance for Cu and Ni(OH)_x electrodes in a 1 M KOH solution with 0.1 M KNO₂, and proceeded to quantify their intrinsic activity through ECSA normalization. Figure R1-7 unequivocally evinces Cu has a better intrinsic activity for NO₂⁻ reduction than Ni(OH)₂ counterpart. Consequently, the observed performance enhancement of Ni(OH)_x/Cu relative to the Cu electrode does not arise from direct NO₂⁻ reduction by Ni(OH)_x sites. Rather, it stems from the accelerated water dissociation observed on the Ni(OH)_x/Cu electrode, engendering a greater abundance of H_{ad}. This elucidation reinforces a nuanced understanding of the underlying mechanisms governing our experimental results. We have supplemented Figure R1-7

in the revised Supporting Information.

Figure R1-7. I-t curves of Cu (a) and Ni(OH)₂ (b) under different potentials. Potential-dependent NH₃ yield rate and corresponding FE over Cu (b) and Ni(OH)₂ (d). ECSA values of Cu and Ni(OH)₂ (e). ECSA-normalized NH₃ yield rate of Cu and Ni(OH)₂ (f).

Comment 6. *In Figure 2f, sentence: ‘The FEs of NH₃ exceed 90% within a NO₃⁻ concentration range from 5 mM to 100 mM while drop to 80% at 1 mM because the competing HER may become more significant at low NO₃⁻ concentration.’ may not be appropriate since with NO₃⁻ at 50 mM, the FE is also around 80%. This makes the concentration influence on FE of NH₃ questionable. More consideration may be required to explain why at 50 mM NO₃⁻, the FE gets lower compared to other concentrations.*

Response: We acknowledge your valuable comment. We believe that a test error caused the anomalous FE value at 50 mM. This is because when using the indophenol blue method for activity detection, it is often necessary to dilute the electrolyte 50-200

times after the reaction to keep the absorbance in the detectable range of the instrument, which makes the test results prone to errors. In order to more accurately test the effect of different nitrate concentrations on the performance of NH_3 synthesis, three independent tests were reconducted and the standard deviation was taken as the error bar (Figure R1-8). At a nitrate concentration of 50 mM, the mean FE value reached 89.5% at -0.25 V vs. RHE. Figure R1-8 and a new corresponding description of the figure have been added to the revised manuscript.

Figure R1-8. The FE of NH_3 , NH_3 yield rate, and the ratio of the generated NH_3 concentration $[\text{NH}_3]$ to the consumed NO_3^- concentration $[\text{NO}_3^-]$ over $\text{Ni}(\text{OH})_x/\text{Cu}$ at -0.25 V vs. RHE at $[\text{NO}_3^-]$ in the range of 1-100 mM in 1 M KOH.

Comment 7. *In supporting information, Figure S27 what is the NO_3^- concentration? Like 100mM? Besides what is the potential, -0.30 V vs. RHE? If the concentration is 0.1 M and the applied potential is -0.30 V vs. RHE? Is the figure 2b is one of the 15 tests in Figure S27? However, at figure S27 there is no NH_3 yield rate around 3 mmol/h/cm². This needs some explanation. The average NH_3 yield looks like around 2.5 mmol/h/cm². Then back to supplementary figure 23f, it would be more appropriate to provide error bar for the normalized NH_3 yield rate since you have 15 independent tests. With this, if we consider the average normalized NH_3 yield rate and compare it with that on Cu, the difference between their intrinsic catalytic might be much closer than what you presented in supplementary figure 23f.*

I) As a result, I would suggest the author to include the error bar for normalized NH_3 yield rate both for Cu and $\text{Ni}(\text{OH})_x/\text{Cu}$. This should give a better understanding about the intrinsic catalytic activity difference between Cu and $\text{Ni}(\text{OH})_x/\text{Cu}$.

II) I would also suggest replacing figure 2b with supplementary 23f since the latter provides 'real' insights for the catalytic activity difference.

Response: Thanks to the reviewer for these valuable comments. In Figure 27 of our previous manuscript, the electrolyte of 1 M KOH with **0.1 M NO_3^-** was used in the 15 independent tests under the applied potential of **-0.25 V vs. RHE**. In addition, the data of Figure 2b is one of the 15 tests in Figure S27. It is worth noting that in 15 tests we used 15 electrodes instead of testing the same electrode 15 times. In order to mitigate the capillary effect leading to an increased reaction surface area within the Cu foam, we employed silicone rubber for the sealing of the Cu foam. Given the inherent challenge of precisely controlling the degree of silicone rubber infiltration during the sealing process, there may be variations in the actual reaction surface area among different electrodes. Consequently, this discrepancy resulted in variations in activity over the course of 15 tests.

We concur with the reviewer's assertion that the activity comparison with ECSA normalization serves as a more accurate reflection of the performance disparities among different catalysts. To enhance clarity, we have incorporated error bars in the ECSA-normalized NH_3 yield rates for Cu, $\text{Ni(OH)}_x/\text{Cu}$, and Ni(OH)_2 (Figure R1-9 and R1-10), and have integrated the ECSA-normalized activities into the main body of the revised manuscript.

Figure R1-9. NH_3 yield rate and corresponding FE of NH_3 over Ni(OH)_2 under different potentials. The error bars represent the standard deviation from three independent tests.

Figure R1-10. ECSA-normalized NH₃ yield rate of Ni(OH)_x/Cu, Cu, and Ni(OH)₂.

Comment 8. Sentences: “However, the impact of the alkali cation over Cu is very limited due to its weak interaction with hydrated cation. The results illustrate that the Ni(OH)_x species boosts the NH₃ formation on the Cu surface via interaction with hydrated K⁺ and accelerated H₂O dissociation.” is a bit confusing, Please rewrite it like: even though the impact of the alkali cation over Cu is very limited due to its weak interaction with hydrated cation, The results illustrate that the Ni(OH)_x species boosts the NH₃ formation on the Cu surface via interaction with hydrated K⁺

Response: Many thanks to the reviewer for this detailed and valuable comment. As you suggested, we have changed and highlighted the corresponding statements in the revised manuscript. In detail, the sentences “The NH₃ yield rate over Ni(OH)_x/Cu increases from 1.79 to 2.7 mmol h⁻¹ cm⁻² on changing the alkali cation from Li⁺ to K⁺. However, the impact of the alkali cation over Cu is very limited due to its weak interaction with hydrated cation. The results illustrate that the Ni(OH)_x species boosts the NH₃ formation on the Cu surface via interaction with hydrated K⁺ and accelerated H₂O dissociation.” have been revised as “**Even though the impact of the alkali cation over Cu is very limited due to its weak interaction with hydrated cation, the NH₃ yield rate over Ni(OH)_x/Cu increases from 1.79 to 2.7 mmol h⁻¹ cm⁻² on changing the alkali cation from Li⁺ to K⁺, indicating that the Ni(OH)_x species boosts the NH₃ formation on the Cu surface via interaction with hydrated K⁺ and accelerated H₂O dissociation.**”.

Comment 9. If it is possible to calculate, what is the overall energy efficiency on the tandem non-thermal plasma and electrocatalysis setup? For application, it deserves to be considered especially, authors here compared their system to the Li-mediated the N₂ conversion in Figure 5c.

Response: We thank the reviewer's important comment here. It is more representative to estimate energy consumption using data from continuous run rather than intermittent operations. Therefore, we re-conducted the 100-hour experiment and measured various parameters such as plasma power consumption, NO_x^- concentration, current density, cell potential, and NH_3 production for energy consumption calculation. The current density, NH_3 production, and NO_x^- concentration are displayed in Figure R1-11.

Figure R1-11. I-t curves and corresponding NH_3 yield rate of long-term continuous test over $\text{Ni}(\text{OH})_x/\text{Cu}$ at -0.25 V vs. RHE using pure air as the feeding gas (a). Concentration variations of NH_3 (b) and NO_x^- (c) along with the test.

Then we estimated the energy efficiency of pNOR-e NO_x^- RR using the following process.

Calculation of energy consumption for pNOR:

$$\text{Total } \text{NO}_x^- \text{ produced} = \text{NO}_x^- \text{ left in catholyte} + \text{Generated } \text{NH}_3 = 0.9058 \text{ mol} + 0.1257 \text{ mol} = 1.0315 \text{ mol}$$

$$\text{Energy consumption for pNOR} = \text{Total energy consumed by pNOR in 100 hours} / \text{Total } \text{NO}_x^- \text{ produced} = 3.45 \text{ kWh} / 1.0315 \text{ mol} = 3.3446 \text{ kWh mol}^{-1} \text{NO}_x^-$$

Calculation of energy consumption for e NO_x^- RR:

We assume the electrolysis time is 1 hour.

Energy consumption for $eNO_x^-RR = \text{Total current} \times \text{Cell voltage} \times \text{Time} / NH_3$
produced = $0.24 \text{ A} \times 9.2 \text{ V} \times 1 \text{ h} / 1.2574 \text{ mmol} = 1.756 \text{ kWh mol}^{-1} NH_3$

Total energy consumption = Energy consumption for pNOR + Energy consumption
for $eNO_x^-RR = 3.3446 \text{ kWh mol}^{-1} NO_x^- + 1.756 \text{ kWh mol}^{-1} NH_3 = 5.1006 \text{ kWh mol}^{-1}$
= $18.3622 \text{ MJ mol}^{-1}$

Calculation of energy efficiency:

We referred to the work of Sun et al. for the energy efficiency calculation [Energy
Environ. Sci., **2021**, 14, 865]. The detailed calculation process is outlined as follows:

The reaction of the plasma-assisted nitrogen oxidation is described by Equation R1.

$$\Delta H^0 = -64.5 \text{ kJ/mol of } NO_3^-$$

$$\Delta G^0 = 7.3 \text{ kJ/mol of } NO_3^-$$

The reaction in the electrochemical side is described by Equation R2.

$$\Delta H^0 = 360.7 \text{ kJ/mol of } NH_4^+$$

$$\Delta G^0 = 269.1 \text{ kJ/mol of } NH_4^+$$

Therefore, the overall reaction for the pNOR- eNO_x^-RR system is:

$$\Delta H^0 = 296.3 \text{ kJ/mol of } NH_4^+$$

$$\Delta G^0 = 276.4 \text{ kJ/mol of } NH_4^+$$

Table R1-2. Enthalpy (ΔH_f^0) and Gibbs free energy (ΔG_f^0) of the reactants and products.

Compounds	ΔH_f^0 (kJ/mol)	ΔG_f^0 (kJ/mol)
$N_2(g)$	0	0
$O_2(g)$	0	0
$H_2O(l)$	-285.83	-237.178
$H^+(aq)$	0	0
$NO_3^-(aq)$	-207.4	-111.3
$NH_4^+(aq)$	-132.5	-79.37

Hence, the energy efficiency for the pNOR- eNO_x^-RR tandem system is:

$$\eta = \frac{296.3}{18362.2} \times 100\% = 1.61\%$$

Therefore, the energy consumption and energy efficiency through pNOR- eNO_x^-RR
toward NH_3 synthesis is $18.3622 \text{ MJ mol}^{-1}$ and 1.61%, respectively, comparable to

previous work with energy consumption of 15.516 kWh mol⁻¹ and energy efficiency of 1.9% by combining a non-thermal plasma bubble column reactor and Cu nanowire-catalyzed electrochemical conversion [*Energy Environ. Sci.*, **2021**, 14, 865]. Surprisingly, this energy consumption is much lower than a plasma-activated proton conducting solid oxide electrolyzer for NH₃ synthesis, which has an energy consumption of 605 MJ mol⁻¹ [*ACS Energy Lett.* **2021**, 6, 313–319]. We have to admit that this energy consumption of pNOR-eNO_x⁻RR is obviously higher than the electrified Haber-Bosch process (0.70 MJ mol⁻¹) [*Joule* **2018**, 2, 6, 1055-1074] where H₂ is produced through H₂O electrolysis, and also higher than the traditional fossil fuel-based Haber-Bosch process (0.52-0.81 MJ mol⁻¹) [*Energy* **2005**, 30, 13 2487-2504] where H₂ is produced through steam methane reforming. However, it is worth mentioning that the Haber-Bosch process is only cost-efficient at a very large scale. Most Haber-Bosch plants produce 3×10⁵ to 6×10⁵ tons/year, with some even up to 1×10⁶ tons/year [*Angew. Chem. Int. Ed.* **2020**, 59, 23825–23829]. The pNOR-eNO_x⁻RR route is scalable and very well suited for a decentralized small to medium scale NH₃ production, eliminating transport costs for fertilizers. More importantly, the electricity used in this process can be generated from clean/renewable energies now, such as solar and wind. Thus, the use of pNOR-eNO_x⁻RR could provide a sustainable and eco-friendly method for NH₃ synthesis by directly converting air into NH₃ with completely eliminating CO₂ emissions through the use of renewable energy.

We have supplemented Figure R1-11 and the energy efficiency calculation process in the revised manuscript and Supporting Information.

Reviewer #2

This manuscript presented a tandem non-thermal plasma and electrocatalysis (pNOR-eNO_x⁻RR) NH₃ synthesis system over Ni(OH)_x/Cu foam catalyst. And a NH₃ yield rate of 3 mmol h⁻¹ cm⁻² and Faradaic efficiency of 92% at -0.25 V versus reversible hydrogen electrode were obtained. Besides, a 100-hour continuous test was carried out with a steady 1.3 mmol h⁻¹ cm⁻² NH₃ yield rate. The author illustrated the catalytic mechanism of Ni(OH)_x/Cu via in situ spectroscopies and density functional theory. But some of the data analysis and description in this manuscript are inaccurate or paradoxical. Based of all, I cannot recommend this manuscript for the publication in this journal.

Response: We highly appreciate the reviewer's time and efforts in reviewing our manuscript, and providing us with constructive comments and suggestions to further improve the quality of our paper. We emphasized the innovation and scientific breakthrough of this study in the revised manuscript and carefully addressed the comments from all reviewers. Our work now has an enhanced quality, depth, and rigor, thanks to their valuable input. We sincerely hope that the revised manuscript will satisfy your stringent criteria for the publication of *Nature Communications*.

Comment 1. *XRD patterns demonstrated that the presence of Cu₂O in the Ni(OH)_x/Cu before CV prereduction (Supplementary Figure 6), but Cu LMM Auger spectra did not detect Cu⁺ before CV prereduction (Supplementary Figure 7), why?*

Response: We extend our gratitude to the diligent reviewer for their thorough examination and insightful comment. We re-prepared the sample and conducted XPS analysis, the outcomes of which are depicted in Figure R2-1. As shown in the high-resolution Cu 2p spectrum, major peaks of Cu 2p_{3/2} at 932.6 eV and Cu 2p_{1/2} at 952.3 eV confirmed the existence of Cu⁰ or Cu₂O. Meanwhile, the peaks of Cu 2p_{3/2} at 934.3 eV and Cu 2p_{1/2} at 954.2 eV in combination with the satellite peaks at 940.8 eV and 944.3 eV are typical characteristics of CuO. By analysing the Cu LMM spectrum, distinct peaks corresponding to Cu²⁺ and Cu⁺ at kinetic energies of 917.7 eV and 916.8 eV, respectively, are observed. This observation substantiates the coexistence of CuO and Cu₂O in Cu(OH)_x/CuO, aligning with the findings of Raman and XRD results. To rectify the ambiguity caused by our oversight, we have replaced the XPS spectra in the previous manuscript with Figure R2-1.

Figure R2-1. Cu 2p spectra (a), Cu LMM spectra (b), O 1s spectra (c), and Ni 2p spectra (d) of Ni(OH)_x/CuO and Ni(OH)_x/Cu.

Comment 2. *In Supplementary Figure 22, Ni(OH)_x/Cu electrode with 30 min deposition time showed optimum eNO₃⁻ RR performance, why?*

Response: We appreciate the time and effort the reviewer has invested in reviewing my manuscript. The Ni(OH)_x coverage on the surface of Cu nanowire is closely related to the deposition time. We believe that at a deposition time of 30 min, the coverage of Ni(OH)_x on Cu offers optimal eNO₃⁻ RR performance. This coverage not only furnishes sufficient adsorbed hydrogen species for the reduction of NO₃⁻ on the Cu surface but also avoids excessive loss of Cu sites designated for NO₃⁻ reduction. In order to determine the amount of Ni(OH)_x relative to Cu at different deposition times, we conducted ICP-MS tests to obtain the mass ratio of Ni(OH)_x to Cu for the Ni(OH)_x/Cu nanowires ultrasonically exfoliated from the Cu foam skeleton, as well as for the Ni(OH)_x/Cu bulk electrodes (Table R2-1). The value of x in Ni(OH)_x (ca. 1.53) was obtained from the result of EXAFS fitting. As shown in Table R2-1, we observed that the relative content of Ni(OH)_x increases with increasing deposition time. When the deposition time is too short, less Ni(OH)_x covers the Cu surface, resulting in fewer Ni(OH)_x-Cu heterojunction interfaces. This makes the adsorbed hydrogen species generated at the Ni(OH)_x-Cu interface insufficient for NO₃⁻ reduction. On the other hand, when the deposition time is too long, Ni(OH)_x is over-covered on the Cu surface,

resulting in fewer exposed Cu sites. Since the ECSA-normalized NH_3 yield rate showed that Cu is more reactive for eNO_3^- RR than Ni(OH)_x , the over-coverage of Ni(OH)_x was not conducive to eNO_3^- RR. Only when the deposition time is 30 min, there are enough Ni(OH)_x -Cu interfaces on the Cu nanowires to promote the dissociation of water molecules to produce adsorbed hydrogen species, as well as enough NO_3^- reduction sites. In addition, the eNO_3^- RR performance of different $\text{Ni(OH)}_x/\text{Cu}$ electrodes follows the same trend as their HER performance. The HER performance of $\text{Ni(OH)}_x/\text{Cu}$ electrodes decreases when the Ni(OH)_x coverage is too much, indicating that the adsorbed hydrogen generation is more favourable only when the Ni(OH)_x coverage is appropriate. We have supplemented Table R2-1 in the revised Supporting Information.

Table R2-1. ICP-MS data for the $\text{Ni(OH)}_x/\text{Cu}$ nanowires that ultrasonic peeling from the Cu skeleton and $\text{Ni(OH)}_x/\text{Cu}$ bulk electrodes with different deposition times.

Samples	Deposition time / min	Measured mass of Ni / mg	Measured mass of Cu / mg	$\text{Ni(OH)}_x / \text{Cu}$ mass ratio / % ^a
$\text{Ni(OH)}_x/\text{Cu}$ nanowire	15	0.0351	3.99	1.27
	30	0.0902	5.67	2.30
	45	0.1430	6.23	3.31
$\text{Ni(OH)}_x/\text{Cu}$ bulk electrode	15	0.0378	14.00	0.39
	30	0.0474	10.70	0.64
	45	0.0578	11.00	0.76

^aThe x value is estimated to be 1.53 based on the EXAFS analysis.

Comment 3. *In Page 4, Right column, Lines 5-7, the author said that there is a strong synergy between Ni(OH)_x and Cu upon $\text{Ni(OH)}_x/\text{Cu}$ toward NH_3 production, some detail description should be provided to explain this synergy effect.*

Response: Thanks to the reviewer for this constructive suggestion. Owing to its favourable NH_3 selectivity, Cu has been extensively applied in eNO_3^- RR. However, the performance of Cu is highly potential-sensitive. The main constraint is the unfavourable adsorption and dissociation of water on Cu in the alkaline media, leading to a slow provision of protons that struggles to meet the rapid hydrogenation demand of the nitrogenous intermediates. Only at the potentials where HER strongly happens, Cu has a better NH_3 synthesis efficiency. At this point, the occurrence of the HER can reduce the selectivity for NH_3 production. Therefore, achieving rapid water dissociation on the Cu surface at low overpotentials to meet the hydrogenation demands of eNO_3^- RR is pivotal to addressing the aforementioned challenges.

After introducing Ni(OH)_x on the Cu surface, we observed the highest eNO_3^- RR intrinsic activity on $\text{Ni(OH)}_x/\text{Cu}$ compared with Cu and Ni(OH)_2 counterparts. Through

analysing the results of KIE and EPR, we show that the deposited Ni(OH)_x can accelerate the water dissociation kinetics to produce adsorbed hydrogen species on the Cu surface at low overpotential. These enriched adsorbed hydrogen species involved in the hydrogenation process of eNO₃⁻RR contribute to an enhanced NH₃ yield rate and FE of NH₃ on Ni(OH)_x/Cu electrode. DFT calculations indicate that water adsorption and dissociation primarily occur on Ni(OH)_x. By comparing the intrinsic activities of Cu and Ni(OH)₂ for eNO₃⁻RR reactions, it is evident that Cu serves as the main active site for NH₃ synthesis. Thus, the synergy effect between Ni(OH)_x and Cu upon Ni(OH)_x/Cu toward NH₃ synthesis can be described as follows: the deposition of Ni(OH)_x promote the adsorption and dissociation of water to produce adsorbed hydrogen species that then adsorb on nearby Cu sites and involved in hydrogenation of N-containing intermediates.

Comment 4. *The feeding gas of pNOR is not pure air, but the mixture of N₂ and O₂ (Supplementary Figure 38), then the description of "ammonia synthesis from air" is inappropriate.*

Response: We completely understand the reviewer's concern here. In our pNOR and pNOR-eNO_x⁻RR tests, we used a gas mixture of air and O₂, with air serving as the only N₂ source and a partial source of O₂. We carefully adjust the air-to-O₂ ratio in the feeding gas to control the ratio of N₂ and O₂ components in the plasma chamber. The ratios of N₂ to O₂ that are previously displayed in Supplementary Figure 38 are actually conversions that have been calculated from air-to-O₂ ratios. In order to avoid ambiguity, we have changed the ratios of N₂ to O₂ to an air-to-O₂ ratio for greater rigor in our revised Supporting Information (Figure R2-2).

Additionally, to simplify our experimental setup, we ran our pNOR-eNO_x⁻RR tandem system continuously for an additional 100 hours using only pure air as intake without additional O₂ supplementation. According to Figure R2-3, the pNOR-eNO_x⁻RR tandem system maintained stability during long periods of operation, yielding an NH₃ yield rate of ca. 1.25 mmol h⁻¹ cm⁻² at a current density of about 240 mA cm⁻². The results of the pNOR-eNO_x⁻RR test have been included in our revised manuscript.

Figure R2-2. The influence of the ratio of air to O₂ on the NO_x⁻ concentrations.

Figure R2-3. I-t curves and corresponding NH₃ yield rate of long-term continuous test over Ni(OH)_x/Cu at -0.25 V vs. RHE using pure air as the feeding gas (a). Concentration variations of NH₃ (b) and NO_x⁻ (c) along with the operation.

Comment 5. *What is the energy consumption through pNOR-eNO_x⁻RR towards NH₃ synthesis?*

Response: We thank the reviewer's important comment here. It is more representative to estimate energy consumption using data from continuous run rather than intermittent operations. Therefore, we measured various parameters such as plasma power consumption, NO_x⁻ concentration, current density, cell potential, and NH₃ production in the re-conducted 100-hour experiment for energy consumption

calculation. Then we estimated the energy efficiency of pNOR-eNO_x⁻RR using the following process.

Calculation of energy consumption for pNOR:

$$\text{Total NO}_x^- \text{ produced} = \text{NO}_x^- \text{ left in catholyte} + \text{Generated NH}_3 = 0.9058 \text{ mol} + 0.1257 \text{ mol} = 1.0315 \text{ mol}$$

$$\text{Energy consumption for pNOR} = \text{Total energy consumed by pNOR in 100 hours} / \text{Total NO}_x^- \text{ produced} = 3.45 \text{ kWh} / 1.0315 \text{ mol} = 3.3446 \text{ kWh mol}^{-1} \text{ NO}_x^-$$

Calculation of energy consumption for eNO_x⁻RR:

We assume the electrolysis time is 1 hour.

$$\text{Energy consumption for eNO}_x^- \text{RR} = \text{Total current} \times \text{Cell voltage} \times \text{Time} / \text{NH}_3 \text{ produced} = 0.24 \text{ A} \times 9.2 \text{ V} \times 1 \text{ h} / 1.2574 \text{ mmol} = 1.756 \text{ kWh mol}^{-1} \text{ NH}_3$$

$$\text{Total energy consumption} = \text{Energy consumption for pNOR} + \text{Energy consumption for eNO}_x^- \text{RR} = 3.3446 \text{ kWh mol}^{-1} \text{ NO}_x^- + 1.756 \text{ kWh mol}^{-1} \text{ NH}_3 = 5.1006 \text{ kWh mol}^{-1} = 18.3622 \text{ MJ mol}^{-1}$$

Therefore, the energy consumption through pNOR- eNO_x⁻RR toward NH₃ synthesis is 18.3622 MJ mol⁻¹, comparable to a previous work with energy consumption of 15.516 kWh mol⁻¹ by combining a non-thermal plasma bubble column reactor and Cu nanowire-catalyzed electrochemical conversion [*Energy Environ. Sci.*, **2021**, 14, 865] and much higher than a plasma-activated proton conducting solid oxide electrolyzer, which has an energy consumption of 605 MJ mol⁻¹ [*ACS Energy Lett.* **2021**, 6, 313–319]. We have to admit that this energy consumption of pNOR-eNO_x⁻RR is obviously higher than the electrified Haber-Bosch process (0.70 MJ mol⁻¹) [*Joule* **2018**, 2, 6, 1055-1074] where H₂ is produced through H₂O electrolysis, and also higher than the traditional fossil fuel-based Haber-Bosch process (0.52-0.81 MJ mol⁻¹) [*Energy* **2005**, 30, 13 2487-2504] where H₂ is produced through steam methane reforming. However, it is worth mentioning that the Haber-Bosch process is only cost-efficient at a very large scale. Most Haber-Bosch plants produce 3×10⁵ to 6×10⁵ tons/year, with some even up to 1×10⁶ tons/year [*Angew. Chem. Int. Ed.* **2020**, 59, 23825–23829]. The pNOR-eNO_x⁻RR route is scalable and very well suited for a decentralized small to medium scale NH₃ production, eliminating transport costs for fertilizers. More importantly, the electricity used in this process can be generated from clean/renewable energies now, such as solar and wind. Thus, the use of pNOR-eNO_x⁻RR could provide a sustainable and eco-friendly method for NH₃ synthesis by directly converting air into NH₃ with completely eliminating CO₂ emissions through the use of renewable energy.

The energy calculation process has been supplemented in the revised Supporting Information.

Comment 6. *The I-t curves of Cu and Ni(OH)_x/Cu under different applied potentials are suggested to be added in this manuscript.*

Response: We appreciate the reviewer's important suggestion. We have added the I-t curves of Cu and Ni(OH)_x/Cu under different applied potentials shown below to the revised Supporting Information and marked it in yellow.

Figure R2-4. I-t curves under different potentials over Cu (a) and Ni(OH)_x/Cu electrodes with different deposition time of 15 min (b), 30 min (c), and 45 min (d).

Comment 7. *In Supplementary Figure 44, the ¹H NMR spectra peak position of the concentrated NH₃ aqueous product is different from ¹⁴NH₄⁺ reference sample, why?*

Response: Thank you for the rigorous query about the misalignment of signal peaks in the ¹H NMR spectra. We suspect that the discrepancy in ¹H NMR results may stem from an incorrect choice of reference sample (NH₄Cl), and the failure to equalize the pH values of the two samples during the acidification process, leading to inconsistent testing conditions between them. Thus, we retest the ¹H NMR spectra by choosing the commercial NH₃ solution as the reference sample and adjusting the pH of the diluted samples to 2.00. Besides, for a better comparison, we increased the concentration of maleic acid so that its signal peaks could overlap. As shown in Figure R2-5, the peak position of the obtained NH₃ aqueous solution was well aligned with the reference sample. We have replaced the Supplementary Figure 44a with Figure R2-5 in the

revised Supporting Information.

Figure R2-5. ^1H NMR analysis of the synthesized NH_3 (aq).

Comment 8. *Supplementary Figure 39 showed that the NO_2^- is the main species during the spark discharge NTP, while the author investigated electrocatalytic NO_3^- reduction for NH_3 synthesis, why?*

Response: We fully understand the reviewer's concern about the selection of the model electrolyte we use to evaluate catalyst performance. In eNO_3^- RR process, the first deoxygenation step is $^*\text{NO}_3^- + \text{H}_2\text{O} + 2\text{e}^- \rightarrow ^*\text{NO}_2 + 2\text{OH}^-$. Therefore, the eNO_2^- RR can be seen as a subset of the eNO_3^- RR process [*Angew. Chem. Int. Ed.* **2023**, 62, e202218717]. Furthermore, the 2-electron transfer process of NO_3^- reduction to NO_2^- is more rapid than the 6-electron transfer process of NO_2^- reduction to NH_3 over the Cu surface [*Nat. Commun.* **2022**, 13, 1129]. Thus, it is the NO_2^- reduction process that controls the overall kinetic of NO_3^- reduction toward NH_3 , resulting in a domination distribution of NO_2^- in the product of eNO_3^- RR under low overpotentials. In addition, we observed a constant increase in the concentration of NO_3^- over a 100-h pNOR- eNO_x^- RR test (Figure R2-3c). The concentration of NO_3^- is low at the beginning, so the mass transfer limitation allows for a persistent buildup of NO_3^- . As a result, both eNO_3^- RR and eNO_2^- RR will occur simultaneously during continuous operation of the pNOR- eNO_x^- RR tandem system. Therefore, eNO_3^- RR is more representative than eNO_2^- RR.

Our catalysts were designed to promote the dissociation of water molecules to produce more adsorbed hydrogen to facilitate the hydrogenation process of nitrogen-containing intermediates. Considering that eNO_3^- RR and eNO_2^- RR need respectively 9 and 7 protons toward NH_3 synthesis, our conclusions should hold true for eNO_2^- RR as well. To confirm this, we conducted eNO_2^- RR tests at different potentials using Cu and $\text{Ni}(\text{OH})_x/\text{Cu}$ electrodes. As shown in Figure R2-6, both Cu and $\text{Ni}(\text{OH})_x/\text{Cu}$ exhibit

high Faradic efficiency (> 90%) for NH₃ production. However, Ni(OH)_x/Cu has a higher NH₃ yield rate than Cu due to a rapid water dissociation process over the Ni(OH)_x/Cu.

Based on the above analysis, we chose the more representative eNO₃⁻RR reaction to evaluate the catalyst performance.

Figure R2-6. Potential dependent i-t curves and NH₃ yield rate and corresponding FE over Cu (a, b) and Ni(OH)_x/Cu (c, d) electrodes in 1 M KOH solution with 0.1 M KNO₂.

Comment 9. In Page 9, Left column, Lines 43-44, the whole system reached balance between consumption and production of NO_x⁻ after 20 h operation, then what is the concentration of NO_x⁻ after that?

Response: We would like to express our sincere appreciation to the reviewers for their valuable feedback. In order to investigate the correlation between the production of NO_x⁻ in pNOR and its consumption in eNO_x⁻RR, we monitored the concentration variations of NO₂⁻ and NO₃⁻ in the cathode electrolyte during the re-conducted 100-hour test. Figure R2-3c reveals a consistent rise in the concentrations of NO₂⁻ and NO₃⁻ within the cathodic electrolyte. At the end of the test, the concentrations of NO₃⁻ and NO₂⁻ reached 0.43 M and 2.3 M, respectively. Consequently, throughout the entirety of the testing period, the rate of NO_x⁻ generation in the plasma surpasses that of the electrocatalytic process responsible for NO_x⁻ consumption. However, the increase of current density halts at an approximate threshold of 290 mA cm⁻², notably below the

attainable current density during intermittent operations. To elucidate the determinants behind this cessation of current density augmentation, electrochemical impedance spectroscopy (EIS) was conducted during continuous operation. As delineated in Figure R2-7a, a discernible and consistent decline is observed in the charge transfer resistance of the catalyst, correlating with the escalating concentration of NO_x^- in the cathodic electrolyte. In contrast, the solution resistance varies differently. In the initial five hours of operation, a modest decrement in solution resistance is observed. Subsequently, a progressive increase in solution resistance is noted with the extension of operation duration. As shown in Figure R2-7b, the potential on the working electrode ($\Delta\phi_{1,2}=\Delta\phi_{\text{EDL}}$) is determined by the potential between the working electrode and the reference electrode ($\Delta\phi_{1,3}$) and the voltage drop due to the solution resistance ($\Delta\phi_{2,3}=iR_{\text{solution}}$). During potentiostatic testing, the provided $\Delta\phi_{1,3}$ from the electrochemical workstation remains constant. The gradually increasing R_{solution} leads to an increased iR_{solution} drop between the reference and working electrodes, resulting in a gradual reduction of the actual potential applied to the working electrode. We observed an increase in solution resistance of approximately 0.7Ω . Under a current density of ca. 240 mA cm^{-2} , the potential on the working electrode could decrease by about 0.168 V . At this point, the potential on the electrode was approximately -0.082 V vs. RHE. Consequently, during continuous operation, the current density cannot reach the same level as in the intermittent operation, leading to a decrease in NH_3 yield rate.

Figure R2-7. EIS of Ni(OH)_x/Cu obtained at different operation time (a). Schematic representation of potential drop across a simplified three-electrode system: (1) surface of the working electrode; (2) outer boundary of the diffuse layer; (3) junction of the reference electrode. $\Delta\phi_{1,3}$, $\Delta\phi_{1,2}$, and $\Delta\phi_{2,3}$ represent the voltage applied between the working electrode and the reference electrode, the potential drop across the electrical double layer (EDL, $\Delta\phi_{EDL}$), and the potential drop across the bulk electrolyte, respectively (b).

Comment 10. *Detailed test methods for the EPR measurement should be provided.*

Response: Thanks to the reviewer for reminding us of this, which can help to improve the integrity of our manuscript. The detailed test method of EPR measurement shown below has been added in the revised manuscript.

“Electron paramagnetic resonance (EPR) spectra were obtained using 5,5-dimethyl-1-pyrroline-N-oxide (DMPO) as the H_{ad} trapping reagent. Before testing, the electrolyte was saturated with Ar to avoid the oxidation of DMPO. Then, DMPO was added into the cathode electrolyte to reach a concentration of 30 mmol L⁻¹. Each EPR spectra was collected after 10 min electrolysis at -0.25 V vs. RHE under Ar-bubbling.”

Reviewer #3

The authors reported a tandem ammonia synthesis system, which integrated a pNOR device to activate N_2 in the form of NO_x^- followed by effective eNO_x^-RR with $Ni(OH)_x/Cu$ catalyst. The structural and catalytic function of the $Ni(OH)_x/Cu$ catalyst was investigated in detail, and the tandem pNOR- eNO_x^-RR device could run continuously and achieve ammonia synthesis. Some key details regarding the composition of the catalyst, electrochemical tests and the tandem pNOR- eNO_x^-RR device worth more explanations, especially in the following aspects:

Response: We appreciate the reviewer's comments and advice, which helped improve our paper. We revised our manuscript and provided a point-by-point response to their comments. We also conducted additional analysis to support our conclusions and highlighted all revisions in yellow in our revised manuscript.

Comment 1. *My biggest question is that eNO_x^-RR catalysts based on a combination of Ni and Cu centers have been published for many times. For instance, Sargent et al. made a pioneer study on NiCu alloys to enhance the efficiency of eNO_x^-RR for ammonia synthesis (J. Am. Chem. Soc. 2020, 142, 5702-5708), and recently, CuNi alloy nanoparticles on a Cu foil was synthesized by Zhou et al. via laser irradiation to achieve efficient eNO_x^-RR (Energy Environ. Sci., 2023,16, 2991-3001). In this situation, I encourage the authors to state clear the advantage of the $Ni(OH)_x/Cu$ catalyst in this work compared with reported eNO_x^-RR systems with a combination of Ni and Cu catalytic centers.*

Response: We thank the reviewer for the time and effort spent in reviewing our work and providing helpful feedback. Since Simpson and Johnson reported the performance of CuNi alloys in catalyzing the reduction of NO_3^- to synthesize NH_3 in acidic environment [*Electroanalysis* **2004**, 16, 532–538.], there have been numerous studies on CuNi alloys and catalysts composed of these two metals for eNO_3^-RR . We have summarized some reports regarding the use of Cu and Ni as active sites for NO_3^- reduction in Table R3-1. It can be observed that the $Ni(OH)_x/Cu$ electrode in this work exhibits remarkably high NH_3 FE (91.6%) and NH_3 yield rate ($2.7 \text{ mmol h}^{-1} \text{ cm}^{-2}$) at relatively low overpotentials, surpassing most existing reports. Notably, the CuNi NPs/Cu foil demonstrates a high NH_3 yield rate, attributed to the exceptionally high initial NO_3^- concentration of 0.71 M during testing, which is over seven times higher than our experimental concentration of NO_3^- . Additionally, CuNi NPs/Cu foil achieves less than 80% of FE for NH_3 production at overpotentials less than -0.48 V vs. RHE . Similarly, except for $Cu_{50}Ni_{50}$, CFP- Cu_1Ni_1 , and Cu_2O/NiO , the remaining catalysts only achieve high FE of NH_3 at high overpotentials over -0.4 V vs. RHE . In contrast,

Ni(OH)_x/Cu exhibits high FE of NH₃ at lower overpotentials, attributed to the introduction of Ni(OH)_x, which promotes the water dissociation process on the Cu surface at low overpotentials, meeting the demand for continuous NO₃⁻ reduction toward NH₃.

On the other hand, previous works introducing Ni into Cu were mostly aimed at upward shifting the d-band centers of Cu, thereby promoting the adsorption of NO₃⁻, or forming well-dispersed Cu and Ni dual sites (Ni_{1.5}Cu_{1.5}(HITP)₂), utilizing Ni single-atom sites for the specific adsorption of NO₂⁻ and enabling tandem catalysis. Besides the significant influence of intermediate adsorption behavior on NO₃⁻ reduction, the hydrogenation of nitrogen-containing intermediates in alkaline environments also greatly affects the efficiency of NH₃ synthesis. However, this aspect has often been overlooked in previous studies. In order to systematically and comprehensively investigate the impact of the water dissociation process on NO₃⁻ reduction, we intentionally deposited Ni(OH)_x species on the Cu surface instead of Ni and NiO. This is because Ni exhibits slower water dissociation kinetics and favors H adsorption, leading to an easier combination of H into hydrogen gas on the Ni surface. Although NiO has also been reported to promote the water dissociation process, it strongly adsorbs H, which is not conducive to H transfer to the Cu surface [*ACS Energy Lett.* **2019**, 4, 3002–3010]. Compared with Ni and NiO, Ni(OH)_x species have specific advantages. Previous studies have shown that Ni(OH)₂ could be used as a universal promoter of alkaline HER on various metal surfaces. In a bifunctional mechanism formed in Ni(OH)₂ deposited catalysts, the edges of Ni(OH)₂ promote the dissociation of water and the production of hydrogen intermediates that then adsorb on nearby metal sites [*Angew. Chem. Int. Ed.* **2012**, 51, 12495–12498]. Therefore, the constructed Ni(OH)_x/Cu serves as a platform for us to correlate the water dissociation kinetics on the Cu surface with its eNO₃⁻RR performance. Surprisingly, we found that Ni(OH)_x can interact with hydrated alkali metal cations in the solution, promoting the adsorption and dissociation of water molecules to generate adsorbed hydrogen species for the continuous hydrogenation of nitrogen-containing intermediates on the Cu surface. It is worth noting that while there have been related studies on the role of hydrated alkali metal cations in promoting electrocatalytic CO₂ and CO reduction, we discovered their distinct effects on catalyst performance in eNO₃⁻RR. This deepens understanding of the NO₃⁻ reduction process and provides new insights for the development of highly efficient catalysts in the future.

Table R3-1. The comparison of the NH₃ yield rate and FE of Ni(OH)_x/Cu with the reported catalysts with a combination of Ni and Cu catalytic centers for eNO₃⁻RR.

Catalyst	NO ₃ ⁻ concentration	Electrolyte	FE (%)	Current density (mA cm ⁻²)	NH ₃ yield rate (mmol h ⁻¹ cm ⁻²)	Potential (V vs. RHE)	Ref.
Ni(OH) _x /Cu	0.1 M	1 M KOH	91.6	—	2.7	-0.25	This work
Cu ₅₀ Ni ₅₀ /PTFE	0.1 M	1 M KOH	99 ± 1	53.0	—	-0.15	[1]
Cu ₅₀ Ni ₅₀ /Cu foam			~95	90	—	-0.1	
CuNi NPs/Cu foil	0.71 M	1 M NaOH	~97.03	—	5.56	-0.48	[2]
Cu/Cu ₂ O-Ni/Ni(OH) ₂	0.1 M	0.01 M KOH + 0.5 M Na ₂ SO ₄	88.0	—	0.584	-1.0	[3]
Ni ₁ Cu-SAA	200 ppm	0.5 M K ₂ SO ₄	~100	—	0.327	-0.55	[4]
Ni _{1.5} Cu _{1.5} (HITP) ₂	50 mM	0.1 M Na ₂ SO ₄	72.45	—	0.13	-0.9	[5]
CFP-Cu ₁ Ni ₁	0.1 M	0.5 M Na ₂ SO ₄	95.7	—	0.18	-0.22	[6]
Cu/Ni-NC	100 ppm	0.5 M Na ₂ SO ₄	97	—	0.16	-0.7	[7]
Cu ₂ O/NiO	200 ppm	0.5 M Na ₂ SO ₄	95.6	—	0.21	-0.2	[8]

Reference:

- [1] *J. Am. Chem. Soc.* **2020**, 142, 5702-5708
 [2] *Energy Environ. Sci.* **2023**, 16, 2991-3001
 [3] *Angew. Chem. Int. Ed.* **2023**, 62, e202217337
 [4] *Appl. Catal. B Environ.* **2022**, 316, 121683
 [5] *Green Chem.* **2023**, DOI: 10.1039/d3gc02613b
 [6] *Nano Res.* **2023**, 16, 6632–6641
 [7] *Small* **2023**, 19, 2207695
 [8] *Adv. Funct. Mater.* **2023**, 2303803

Comment 2. According to the preparation of the catalyst, “Ni(OH)_x was deposited on the CuO NWA surface by rinsing CuO NWA into a NiCl₂ solution at open circuit potential” (the first paragraph in results and discussions). Here, the term “open circuit potential” is ambiguous, or at least not quantitative and precise enough like “+0.24 V vs. RHE”, particularly given that the open circuit potential of an electrode can be variable, and the chemical state of Cu is highly susceptible to applied potentials (Supplementary Figure 1).

Another issue related with catalyst preparation is that the relative amounts of Ni on Cu seem unclear. Especially, given that different rinsing times were used, the Ni loading on CuO NWA would vary significantly.

Response: Thank you to the reviewer for their valuable comments. As you suggested, we have recorded the open circuit potential during the deposition of Ni(OH)_x (Figure R 3-1). We find that the open circuit potential decreases from 0.49 V vs. RHE to 0.39 V vs. RHE, so we cannot describe it as a specific value.

To provide a more precise variation in the content of Ni(OH)_x compared to Cu in the electrodes with different deposition times, we used the Inductively Coupled Plasma Mass Spectrometer (ICP-MS) to test Ni and Cu in the Ni(OH)_x/Cu electrode. Considering that the Ni(OH)_x/Cu electrode comprises Ni(OH)_x/Cu nanowires uniformly grown on the surface of a Cu foam skeleton, we quantitatively tested the Ni and Cu contents of the Ni(OH)_x/Cu bulk electrode, as well as Ni(OH)_x/Cu nanowires, which were ultrasonically exfoliated from its surface. Using the EXAFS fitting data, we estimated that the x in Ni(OH)_x is roughly equal to 1.53, from which we calculated the relative contents of Ni(OH)_x and Cu at different deposition times. As shown in Table R3-2, the mass ratio of Ni(OH)_x to Cu on Ni(OH)_x/Cu nanowires increased from 1.27% to 2.30% and then to 3.31% as the deposition time increased. However, for Ni(OH)_x/Cu bulk electrodes, the mass ratio of Ni(OH)_x to Cu is much smaller due to the increase of Cu content caused by the Cu foam skeleton. We have added Figure R3-1 and Table R3-2 to the revised Supporting Information.

Figure R3-1. Open circuit potential recorded during Ni(OH)_x deposition in a three-electrode cell.

Table R3-2. ICP-MS data for the Ni(OH)_x/Cu nanowires that ultrasonic peeling from the Cu skeleton and Ni(OH)_x/Cu bulk electrodes with different deposition times.

Samples	Deposition time / min	Measured mass of Ni / mg	Measured mass of Cu / mg	Ni(OH) _x / Cu mass ratio / % ^a
Ni(OH) _x /Cu nanowire	15	0.0351	3.99	1.27
	30	0.0902	5.67	2.30
	45	0.1430	6.23	3.31
Ni(OH) _x /Cu bulk electrode	15	0.0378	14.00	0.39
	30	0.0474	10.70	0.64
	45	0.0578	11.00	0.76

^aThe x value is estimated to be 1.53 based on the EXAFS analysis.

Comment 3. “The linear sweep voltammetry (LSV) polarization curves in Fig. 2a reveal that, in the presence of NO_3^- , eNO_3^- RR exhibits an earlier onset potential when compared to...” It looks obvious, but I think the term “onset potential” warrants a better definition, as different papers evaluate “onset potential” in different ways.

Response: As rightly pointed out by the reviewer, the definition of onset potential varies in different research areas. Although it is a widely accepted measure of electrocatalytic activity, there is no standard definition or measurement procedure agreed upon by the scientific community. To ensure a more rigorous presentation, it is recommended to define the onset potential by stating a threshold current density and reporting the potential at which this threshold is passed [*Curr. Opin. Electroche.* **2023**, 37, 101176]. In this case, we define the onset potential as the potential value corresponding to the current density reaching -5 mA cm^{-2} . However, the polarization curves for eNO_3^- RR in our previous manuscript show current densities larger than -5 mA cm^{-2} at 0.2 V vs. RHE. To address this, we extended the potential scan range positively. The latest polarization curves, shown in Figure R3-2, reveal that the onset potential of eNO_3^- RR (+0.39 V vs. RHE for $\text{Ni(OH)}_x/\text{Cu}$ and Cu) is more positive than that of HER (-0.14 V vs. RHE for $\text{Ni(OH)}_x/\text{Cu}$ and -0.18 V vs. RHE for Cu).

Figure R3-2. LSV curves of $\text{Ni(OH)}_x/\text{Cu}$ and Cu in 1 M KOH with and without 0.1 M NO_3^- under 400 r.p.m. without iR correction.

Comment 4. In Figure 3a and 3b, ΔE at certain potentials were used to visualize the difference of catalytic kinetics in H_2O and D_2O . Here, the question is that ΔE seems highly variable under different current densities, and why the authors chose 40 mA cm^{-2} in HER and 200 mA cm^{-2} in eNO_3^- RR to evaluate ΔE brought by H/D difference? Why a larger ΔE was yielded in eNO_3^- RR than HER?

Response: We would like to express our gratitude to the reviewer for their important comment. To measure the effect of kinetic isotopic effect in LSVs of HER and eNO_3^- RR, we chose to use the middle values of current density in Figures 3a and 3b as a representative quantifier. To better represent the effect of isotope effects on different

electrodes in different reaction environments, we counted the cathode shift values (ΔE) at different current densities. As shown in Figure R3-3, the ΔE of $\text{Ni(OH)}_x/\text{Cu}$ are smaller than those of Cu, with or without the presence of NO_3^- . This suggests that the introduction of Ni(OH)_x enhances the water dissociation process over the Cu surface, which also makes $\text{Ni(OH)}_x/\text{Cu}$ less limited by the hydrogenation steps in eNO_3^- RR at low overpotentials.

Comparing the ΔE of eNO_3^- RR and HER at the same current density is meaningless because their current densities can differ by 1 or 2 orders of magnitude. However, the ΔE of HER at the same electrode are larger than those of eNO_3^- RR in the range of current densities from 0 to -100 mA cm^{-2} . This is due to the fact that the eNO_3^- RR is thermodynamically and kinetically more favourable than HER. It is worth mentioning that, since eNO_3^- RR involves multiple steps of proton-coupled electron transfer, we cannot know in detail the isotope effect on the NH_3 production from the LSV. In Figure 3c, we found that the eNO_3^- RR on the Cu surface is more significantly affected by the isotope effect due to its sluggish kinetics of water dissociation. $\text{Ni(OH)}_x/\text{Cu}$ exhibits a lower KIE value of H/D and a smaller drop of NH_3 FE compared to Cu, further indicating the vital role of Ni(OH)_x in the hydrogenation step of eNO_3^- RR. Figure R3-3 has been added in the revised Supporting Information.

Figure R3-3. Cathode shift values under different current densities over Cu and $\text{Ni(OH)}_x/\text{Cu}$ electrodes in the presence/absence of NO_3^- .

Comment 5. *The pNOR device in this work could run continuously for days at a stable current density of hundreds of mA cm^{-2} . I am very interested how many grams of NH_3 or NH_4Cl was eventually obtained? Based on Figure 5d, one would expect $\sim 100 \text{ mmol cm}^{-2}$ ammonia after the 100-h test, however, only 100 mL 0.2 M HCl was used to absorb it in the gas outlet stream, which could only absorb 20 mmol ammonia. In this situation, will the inadequacy of acid sorbent cause the loss of ammonia obtained?*

Response: We are thankful for the reviewer's thoughtful comments which have

significantly contributed to the improvement of our work. In our previous manuscript, we only assessed the feasibility of NH_3 product collection qualitatively, without converting all NH_3 generated to products. Therefore, we have conducted another 100-h pNOR-e NO_x -RR test, which has given us the opportunity to measure the NH_3 distribution in the catholyte and acid traps as well as to collect all the produced NH_4Cl (s) and NH_3 (aq).

As shown in Figure R3-4c, we placed a secondary acid trap (100 mL of 0.2 M HCl) after the primary acid sorbent (100 mL of 0.2 M HCl) to monitor the NH_3 that may not have been fully absorbed. During the first 20 hours of operation, NH_3 is mainly distributed in the catholyte. After that, NH_3 begins to accumulate in the first acid absorption. In the secondary acid absorption, the concentrations of NH_3 are always at a very low level. **At the end of the operation, only 3.59% of produced NH_3 is distributed in the secondary acid absorption.** Thus, the minimal amount of NH_3 loss will not have a significant impact on the final result of our last manuscript. Additionally, the first acid trap absorbed 39.92 mmol of NH_3 , which is more than 20 mmol. This is due to the fact that NH_3 can react not only with HCl, but also with water. At 25°C and 1.013 bar, the saturation solubility of NH_3 in water is approximately 32 g per 100 mL of water. Therefore, theoretically, 100 mL of 0.2 M HCl is sufficient for our experimental requirements.

In order to collect as much of the produced NH_3 as possible, we first added 0.5 mol of KOH to the catholyte and absorption solution mixture to increase its alkalinity. Then, the mixture was placed in an oil bath and heated to 70 °C. These operations will make it easier to strip out NH_3 . The gas flow rate for air-stripping was 30 sccm. Finally, 200 mL of 1 M HCl absorption solution was used at the end. The experimental setup for the NH_3 collection is displayed in Figure R3-5a. As shown in Figure R3-5b, in the NH_3 (aq) route, 45% of NH_3 in the mixture is condensed as NH_3 (aq) product and 48.53% of it is trapped by the acid solution. In the NH_4Cl (s) route, the acid collection efficiency is about 98%. No NH_3 was detected in the mixture liquid after the air stripping, indicating a complete removal of NH_3 . After rotational evaporation, NH_3 trapped in the acid was converted into solid NH_4Cl (Figure R3-5c). **We eventually obtained 4.6956 g solid NH_4Cl and 63.8 mL of NH_3 aqueous solution with a concentration of 0.77 wt%.** According to the results, 69.8% of the collected NH_3 was converted to NH_4Cl , 23% to NH_3 aqueous solution, and 7.2% was lost in solution transfer, acid absorption, and solid collection.

Figure R3-4 and R3-5 has been added in the revised manuscript and Supporting Information with a corresponding discussion.

Figure R3-4. I-t curves and corresponding NH₃ yield rate of long-term continuous test over Ni(OH)_x/Cu at -0.25 V vs. RHE using pure air as the feeding gas (a). Concentration variations of NH₃ (b) and NO_x⁻ (c) along with the operation.

Figure R3-5. Experimental setup for air stripping and NH₃ collection (a). NH₃ distribution after the air stripping procedure (b). Photographs of collected NH₄Cl (s) after rotational evaporation and NH₃ (aq) after condensation (c). The percentages of NH₃ in the different products relative to the total amount of NH₃ obtained after the long-term test (d).

Comment 6. Whether $\text{Ni(OH)}_x/\text{CuO}$ can be used as electrocatalytic catalyst for nitrate reduction of ammonia. If so, is the performance of $\text{Ni(OH)}_x/\text{CuO}$ better or worse than that of $\text{Ni(OH)}_x/\text{Cu}$?

Response: We extend our sincere gratitude to the esteemed reviewer for their valuable insights. The use of CuO in the reduction of NO_3^- to produce NH_3 has been previously studied, exemplified by Daiyan *et al.*'s synthesis of CuO enriched with oxygen vacancies for NO_3^- reduction [*Energy Environ. Sci.*, **2021**, 14, 3588–3598]. Hence, we posit that $\text{Ni(OH)}_x/\text{CuO}$ can serve as a catalyst for NO_3^- reduction. To assess the eNO_3^- RR performance of $\text{Ni(OH)}_x/\text{CuO}$, we measured its activity and NH_3 FE under different potentials in a 1 M KOH electrolyte containing 0.1 M KNO_3 . As depicted in Figure R3-6, it is discernible that at potentials more positive than -0.2V vs. RHE, $\text{Ni(OH)}_x/\text{CuO}$ exhibits decreased NH_3 yield rate and FE compared to $\text{Ni(OH)}_x/\text{Cu}$, underscoring the enhanced synergistic effect between Ni(OH)_x and Cu over Ni(OH)_x and CuO. However, beyond -0.2V , the performance of $\text{Ni(OH)}_x/\text{CuO}$ aligns with that of $\text{Ni(OH)}_x/\text{Cu}$, signifying the complete reduction of CuO to Cu at -0.2V . This agrees with previous reports, where the Raman signals of CuO and Cu_2O both disappear when the potential is increased to -0.2V vs. RHE [*Adv. Funct. Mater.* **2023**, 2303803].

Given the sensitivity of valence states of Cu in CuO to applied potential, investigating the role of Ni(OH)_x in eNO_3^- RR using $\text{Ni(OH)}_x/\text{CuO}$ as the subject of study would have rendered the exploration challenging. Thus, we pre-reduced $\text{Ni(OH)}_x/\text{CuO}$ to obtain $\text{Ni(OH)}_x/\text{Cu}$, ensuring that the catalyst composition remains unchanged during the reaction.

Figure R3-6. Potential-dependent i - t curves (a) and NH_3 yield rate and FE of NH_3 (b) over $\text{Ni(OH)}_x/\text{CuO}$ electrode.

Comment 7. The authors claim that "The FEs of NH_3 exceed 90% within a NO_3^- concentration range from 5 mM to 100 mM". However, according to Figure 2, the FEs of NH_3 at about 70 mM is similar to that at 1 mM, both of which are about 80%. The explanation given by the authors is that the competition HER is more significant due to

the lower concentration of NO₃⁻ (1mM). So, what causes the FEs drop at 70 mM?

Response: We acknowledge your valuable comment. We believe that a test error caused the anomalous FE value at 50 mM. This is because when using the indophenol blue method for activity detection, it is often necessary to dilute the electrolyte 50-200 times after the reaction to keep the absorbance in the detectable range of the instrument, which makes the test results prone to errors. In order to more accurately test the effect of different NO₃⁻ concentrations on the performance of NH₃ synthesis, three independent tests were conducted and the standard deviation was taken as the error bar (Figure R3-7). At a NO₃⁻ concentration of 50 mM, the mean FE value reached 89.5%. A new description of the figure has been added to the revised manuscript.

Figure R3-7. The FE of NH₃, NH₃ yield rate, and the ratio of the generated NH₃ concentration [NH₃] to the consumed NO₃⁻ concentration [NO₃⁻] over Ni(OH)_x/Cu at -0.25 V vs. RHE at [NO₃⁻] in the range of 1-100 mM in 1 M KOH.

Comment 8. *According to Figure S26b, no matter what NO₃⁻ concentration is, the current on the Ni(OH)_x/Cu electrode is not stable during half an hour of electrolysis, and the current density fluctuates by about 100 mA/cm². It seems that the Ni(OH)_x/Cu electrode is not very stable.*

Response: We express our gratitude to the reviewers for their invaluable feedback. We don't believe that electrode instability is responsible for the fluctuations in current density. Current fluctuations seem to be a common phenomenon in NO₃⁻ reduction reaction [*Nat. Catal.* **2023**, 6, 402–414] due to the current can be affected by the decrease of NO₃⁻ concentration as the reaction proceeds, the production of numerous intermediates during electrolysis, and changes in the reaction environment such as pH and temperature [*Angew. Chem. Int. Ed.* **2022**, 61, e202202556]. As we employed a 1 M KOH solution, the pH fluctuations during the electrolysis process are not as conspicuous as they would be in a neutral electrolyte. Thus, we posit that the variation in current density is primarily governed by alterations in the concentration of NO₃⁻ within the electrolyte and fluctuations in the temperature of the electrolyte. We

monitored the temperature variations during the electrolysis process under different NO_3^- concentrations, as depicted in Figure R3-8. At lower NO_3^- concentrations (1-50 mM), the temperature rises in the electrolyte due to Joule heating generated during the electrolysis is not markedly significant. Consequently, at these conditions, the variation in current is predominantly determined by the reduction in NO_3^- concentration. However, at a NO_3^- concentration of 100 mM, the elevated current leads to a rapid increase in the electrolyte temperature. At this point, the temperature rise exerts a more substantial promoting effect on the reaction compared to the inhibitory effect of the reduced NO_3^- concentration on the current density [*J. Am. Chem. Soc.* **2020**, 142, 7036-7046].

Figure R3-8. Temperature variation in the electrolyte with different NO_3^- concentrations.

To further validate the stability of our catalyst, 10 cyclic tests were performed on the $\text{Ni}(\text{OH})_x/\text{Cu}$ electrode. As shown in Figure R3-9, FEs of NH_3 were stable at ~91% in all the 10 tests and the NH_3 yield rates were maintained at $\sim 2.6 \text{ mmol h}^{-1} \text{ cm}^{-2}$. This result, together with the results of long-term operations in the H-type flow cell as well as in the pNOR-e NO_3^- RR tandem system, demonstrates the good stability of the $\text{Ni}(\text{OH})_x/\text{Cu}$ electrode.

Figure R3-9. I-t curves of cycling eNO₃⁻RR tests over Ni(OH)_x/Cu at -0.25 V vs. RHE in 1 M KOH solution with 0.1 M KNO₃ (a). The corresponding NH₃ yield rate and FE of NH₃ (b).

Comment 9. According to Figure 5d, during the 100-hour test, the ammonia yield and current density in the first 20 hours are not stable. The author should give the corresponding explanation.

Response: We quietly appreciate the valuable advice. In the continuous experiment, we employed a 1 M KOH solution as the electrolyte for absorbing the NO and NO₂ generated by the plasma. Initially, upon the initiation of the pNOR-eNO_x⁻RR tandem system, as there was no NO_x⁻ in the electrolyte, only the HER occurred on the electrode surface. As pNOR continuously generated NO and NO₂, the concentration of NO_x⁻ in the electrolyte progressively increased, resulting in a continuous elevation of current density and NH₃ yield rate. However, as the current reached ~290 mA cm⁻², it ceased to increase further, although the concentration of NO_x⁻ was still on the rise (Figure 3-4c). To elucidate the determinants behind this cessation of current density augmentation, electrochemical impedance spectroscopy (EIS) was conducted during continuous operation. As delineated in Figure R3-10a, a discernible and consistent decline is observed in the charge transfer resistance of the catalyst, correlating with the escalating concentration of NO_x⁻ in the cathodic electrolyte. In contrast, the solution resistance varies differently. In the initial five hours of operation, a modest decrement in solution resistance is observed. Subsequently, a progressive increase in solution resistance is noted with the extension of operation duration. As shown in Figure R3-10b, the potential on the working electrode ($\Delta\phi_{1,2}=\Delta\phi_{EDL}$) is determined by the potential between the working electrode and the reference electrode ($\Delta\phi_{1,3}$) and the voltage drop due to the solution resistance ($\Delta\phi_{2,3}=iR_{\text{solution}}$). During potentiostatic testing, the

provided $\Delta\phi_{1,3}$ from the electrochemical workstation remains constant. The gradually increasing R_{solution} leads to an increased iR_{solution} drop between the reference and working electrodes, resulting in a gradual reduction of the actual potential applied to the working electrode. Therefore, during the reaction process, the environment at the electrode surface is constantly changing, leading to fluctuations in both current density and NH_3 yield.

Figure R3-10. EIS of $\text{Ni}(\text{OH})_x/\text{Cu}$ at different operation time (a). Schematic representation of potential drop across a simplified three-electrode system: (1) surface of the working electrode; (2) outer boundary of the diffuse layer; (3) junction of the reference electrode. $\Delta\phi_{1,3}$, $\Delta\phi_{1,2}$, and $\Delta\phi_{2,3}$ represent the voltage applied between the working electrode and the reference electrode, the potential drop across the electrical double layer (EDL, $\Delta\phi_{\text{EDL}}$), and the potential drop across the bulk electrolyte, respectively (b).

Comment 10. *Figure 5d and S43, compared with $\text{Ni}(\text{OH})_x/\text{Cu}$ electrode, the current density of the $p\text{NOR}-e\text{NO}_x^-$ RR system is more stable when the Cu electrode is used for long-term continuous testing.*

Response: Thank you very much for the reviewer's engagement and valuable comment. In the previous versions of the manuscript, we conducted sampling tests on

the Ni(OH)_x/Cu electrode during long-term testing. After each sampling, an equal volume of fresh 1 M KOH was added to the cathodic circulating solution to ensure constant electrolyte volume. As for the Cu electrode, we solely recorded the current density without sampling and analyzing the NH₃ content in the system. Thus, the system remained undisturbed. Upon prolonged operation of Ni(OH)_x/Cu, fluctuations in current density were observed after each addition of fresh electrolyte. To standardize the testing conditions between the two, we re-evaluated the long-term stability of Ni(OH)_x/Cu and Cu electrodes in the pNOR-eNO₃⁻RR system. In this new testing, air was employed as the source of aeration, and no equal volume of fresh electrolyte was added after intermittent sampling to avoid disturbing the system. As displayed in Figure R3-4a and R3-11, a similar trend of current density variation can be observed over Ni(OH)_x/Cu and Cu electrodes. I-t curve of Cu for continuous run in the previous Supporting Information has been replaced with Figure R3-11.

Figure R3-11. I-t curves and corresponding NH₃ yield rate over Cu electrode in continuous run of the pNOR-eNO₃⁻RR tandem system.

Comment 11. *High-resolution Ni 2p XPS spectra of Ni(OH)_x/CuO and Ni(OH)_x/Cu needs to be fitted.*

Response: We greatly appreciate the reviewer's suggestions. In the revised manuscript, we have re-examined the XPS spectra of Ni(OH)_x/CuO and Ni(OH)_x/Cu (in response to Reviewer #2's comments) and subsequently performed fitting for all the spectra. As shown in Figure R3-12d, in the high-resolution Ni 2p XPS spectra of Ni(OH)_x/CuO and Ni(OH)_x/Cu, two prominent peaks at bind energy of 855.9 and 873.4 eV can be assigned to Ni 2p_{3/2} and Ni 2p_{1/2} of Ni²⁺ spin-orbit doublets and those at 861.5 and 879.2 eV are ascribed to two accompanying satellites. XPS spectra in the previous

Supporting Information has been replaced with Figure R3-12.

Figure R3-12. Cu 2p spectra (a), Cu LMM spectra (b), O 1s spectra (c), and Ni 2p spectra (d) of Ni(OH)_x/CuO and Ni(OH)_x/Cu.

Comment 12. Why did the NH₃ yield rate decrease from 3.0 to 1.3 mmol h⁻¹ cm⁻² when using the pNOR-eNO_x⁻RR system instead of the solutions with different spark discharge time as electrolyte.

Response: Thank you very much for the reviewer's feedback. It is important to note that the environment in which the catalyst operates during intermittent testing differs from that during continuous long-term operation. As shown in Figure R3-4b and R3-4c, over the 100-hour test, the concentrations of NH₃ and NO_x⁻ in the cathodic electrolyte gradually increase with time. Although the concentration of reactants continuously increases, the escalating solution resistance after 5 hours of operation prevents the current from reaching as high as it does during intermittent operation. We observed an increase in solution resistance of approximately 0.7 Ω in Figure R3-10. Under a current density of 240 mA cm⁻², the potential on the working electrode could decrease by about 0.168 V (100% *iR* compensation) or 0.134 V (80% *iR* compensation). At this point, the potential on the electrode was approximately -0.116 ~ -0.082 V vs. RHE. Consequently, during continuous operation, the current density cannot reach the same level as in the intermittent operation, leading to a decrease in NH₃ yield rate.

REVIEWER COMMENTS

Reviewer #1 (Remarks to the Author):

Thank all authors for the detailed reply on our comments. However, there are some critical questions or suggestions for this manuscript from my aspect, as follows. Besides, I cannot recommend this manuscript for the publication in this journal.

Major comments:

(1) For the first question

i) Is Ni(OH)_x really on top of Cu contributes to the NO₃⁻ reduction, what about NiO or even NiCu alloy (oxides) contribute to the increase of NH₃ formation? How does the catalytic active site look like really needs more attention.

Authors argued that 'The eNO₃-RR performance over the Cu surface is closely related to the supply of adsorbed hydrogen species, ...'. what is the support for this? Is it more concerned about the conversion from NO₃⁻ towards NO₂⁻?

(2) For the second question:

ii) As for cation effect, it is very interesting. Independently if more detailed investigation is included, it can be full another story. That would be great if more insightful explanation, like DFT is provided, like this has been done on CO₂/CO reduction on transition metals, of course, in order to do this, detailed active sites is demanded.

Thank authors' effort for adding more simulations, but 1) why do u fix all the cations on the same distance above metal surfaces? For different cations due itself size, the distance to metal surface is different, right? Besides, in the literature [Chem. Sci., 2022, 13,7634–7643], for example, the Na⁺ is suggested remaining near the surface around 3Å. Besides, this cations distribution should be system dependent and potential dependent [J. Phys. Chem. C 2020, 124, 37, 20055–20065]. When it comes to the Ni(OH)_x/Cu system, the cations distribution might be totally different. For now the present simulation is not enough. Besides, the construction of the system Ni(OH)_x/Cu, especially, Ni(OH)_x part is not convincing. Additionally, why authors calculate the adsorption of H₂O to explain the effect from cations on NH₃ formation?

(3) For the question: "In the study, for the impact from cations (Li⁺, Na⁺, K⁺, Cs⁺), authors claimed 'The M+(H₂O)_n (where n = 22, 13, 7, and 6 for Li⁺, Na⁺, K⁺, and Cs⁺, respectively) with smaller n and radii is believed to have a stronger interaction with anionic species and thus a greater ability to activate H₂O'. From this, Cs⁺ and K⁺ should have similar effect on the NO₃⁻ reduction. however, it seems for Cs⁺, it doesn't follow the trend and actually deactivates the NH₃ formation, what is the possible explanation for this?" Authors mentioned that: 'the concentration of alkali metal cations is very high (1 M CsOH plus 0.1 M CsNO₃), potentially leading to direct adsorption of Cs⁺ on the catalyst surface ...', however, the potential for the cations' adsorption or not is more important factor. Then as for Cs poisoning and blocking the active site, you can't just guess but to prove. For now, DFT simulation doesn't contribute too this key experimental observations (cations effect on NH₃ formation).

Reviewer #2 (Remarks to the Author):

The authors have well revised the manuscript according to the comments, and I recommend the

publication of this manuscript in Nature Communications.

Reviewer #3 (Remarks to the Author):

In the revised manuscript, the authors have made significant efforts to elucidate the structure and composition of catalysts and evaluate the energy efficiency of the tandem pNOR-NO₃-RR scheme. I have the following questions that worth more explanation.

1. According to ICP, there is only a small, auxiliary amount of Ni with large, major portions of Cu in Ni(OH)_x/Cu. However, it seems Ni(OH)_x forms a separate, exterior covering layer outside Cu based on Figure 1d. Does the difference of deposition time finally affect the coverage and thickness of Ni(OH)_x layer?
2. In calculating the energy efficiency of the tandem pNOR-NO₃-RR process, the ΔH and ΔG of the overall reaction from N₂, H₂ and H₂O to NH₄⁺ and O₂ was evaluated. Here, is it more rational to adopt NH₃ instead of NH₄⁺ as the product, given that the electrolysis was done under basic conditions with excessive NaOH/KOH, while ammonia is more inclined to exist as NH₃ instead of NH₄⁺ at a pH above 9-10?
3. In tandem pNOR-NO₃-RR, the revised manuscript discussed how the solution conductivity and current density varied after long hours. Here, I wonder whether the following two factors are in effect: a) as the OH⁻ reacts with nitric oxides generated in pNOR, the generated NO_x⁻ ions would have lower ion mobilities compared with OH⁻, which in turn increase the solution resistance? b) as the air after pNOR is purged into the NaOH/KOH sorbent, the consumption of OH⁻ cause a major pH drop of the electrolyte. This pH drop is increasingly significant if untreated air is used, because the small amounts of CO₂, although only exist in hundreds of ppm, are cumulatively captured by OH⁻ and forms carbonates: As far as I have observed recently, when 1.0 M NaOH was used to treat the mixture gas in pNOR using ambient air, the solution pH dropped to about 11 at room temperature when the NO_x⁻ concentration reached about 0.5 M, which would be above 13 (50% OH⁻ consumption) if we only take the reaction between OH⁻ and NO_x into consideration. Since the acidity of HCO₃⁻ is much weaker than HNO₂, the existence of carbonates does not affect the sorbent's ability to capture NO_x⁻, eventually obtained ~1.0 M NO_x⁻ when the pH dropped to ~8. However, the major pH drops due to CO₃²⁻ in long-hour pNOR would be a non-negligible factor for electrochemical properties.
4. In the last sentence of the section on "Catalyst synthesis and characterization", it should be "Ni(OH)_x/CuO and Ni(OH)_x/Cu electrodes" instead of "Ni(OH)_x/Cu and Ni(OH)_x/Cu electrodes". Please check the manuscript carefully.
5. Except for FIG. 2a, the control group in FIG. 2 consists of three groups: Cu, Ni(OH)₂, and Ni(OH)_x/Cu. Consistency is recommended.
6. According to Figure S24, the test time affects FE (NH₃). So, the Faraday efficiency and yield in Figure 2 were tested after how long the reaction took?
7. The authors mentioned "As shown in Fig. 3e, when using 15NO₃⁻ as the nitrogen source, only doublet peaks of 15NH₄⁺ are detected in the 1H nuclear magnetic resonance (NMR) spectra of the electrolyte without seeing any triple coupling peaks of 14NH₄⁺". The correct reference for the figure should be "Fig.3e" instead of "Fig.2e".
8. Although the surface morphology and structure of the electrode material remain unchanged after a long time of testing, is there any material leaching from the electrode into the electrolyte?

9. Figure S45 indicates that the concentration of NO_x^- in the reaction solution exceeds 100 mM after 40 minutes, thus necessitating an evaluation of the e NO_3 -RR activity of $\text{Ni}(\text{OH})_x/\text{Cu}$ in electrolytes with higher NO_3^- concentrations.

Response to reviewers' comments:

We are grateful for the critical feedback and constructive suggestions provided by the reviewers. Their input is invaluable in improving the quality of our work. We have carefully revised our manuscript in response to their comments and highlighted the changes in yellow. Below, we provide point-to-point responses to the reviewers' comments. The original comments are presented in blue italics, while our responses are in black.

Reviewer #1

Thank all authors for the detailed reply on our comments. However, there are some critical questions or suggestions for this manuscript from my aspect, as follows. Besides, I cannot recommend this manuscript for the publication in this journal.

Response: We highly appreciate the reviewer's time and efforts in reviewing our manuscript, and providing us with constructive comments and suggestions to further improve the quality of our paper. We have carefully addressed all the received comments. Thanks to their valuable input, our work now has an enhanced quality, depth, and rigor. We sincerely hope that the revised manuscript will satisfy your stringent criteria for the publication in *Nature Communications*.

Major comments:

(1) For the first question

i) Is Ni(OH)_x really on top of Cu contributes to the NO₃⁻ reduction, what about NiO or even NiCu alloy (oxides) contribute to the increase of NH₃ formation? How does the catalytic active site look like really needs more attention.

Authors argued that 'The eNO₃⁻RR performance over the Cu surface is closely related to the supply of adsorbed hydrogen species, ...'. what is the support for this? Is it more concerned about the conversion from NO₃⁻ towards NO₂⁻?

Respond: Thanks to the reviewer for this valuable comment. We have supplemented EPR spectra over Cu and Ni(OH)_x/Cu under different applied potentials and found that the amount of active adsorbed hydrogen species (H_{ad}) has a positive correlation with the NH₃ generation rate in eNO₃⁻RR.

As shown in R1-1, eNO₃⁻RR toward NH₃ production involves 8 electrons and 9 protons. The protons for N-containing intermediates hydrogenation are generated from the water dissociation process (the Volmer step) in the strong alkaline environment (R1-

2, M in the equation stands for any metal acting as a catalyst). However, the kinetics of water dissociation process over Cu is sluggish, leading to an insufficient supply of H_{ad} . $Ni(OH)_x$ species were intentionally deposited on the Cu surface to accelerate the water dissociation process. As shown in Figure R1-1a, b, substituting H_2O with D_2O leads to an obvious cathode shift in the HER and eNO_3^-RR LSVs on Cu since the dissociation kinetics of D_2O is more sluggish than H_2O . While this isotopic effect is insignificant over $Ni(OH)_x/Cu$ due to its fast water dissociation kinetics. Furthermore, the smaller KIE value and higher FE of NH_3 over $Ni(OH)_x/Cu$ than those over Cu in D_2O indicates that the eNO_3^-RR is less limited by H_2O dissociation with the presence of surficial $Ni(OH)_x$ (Figure R1-1c). Additionally, the H_{ad} desorption peak of $Ni(OH)_x/Cu$ at the range of $-0.2 \sim -0.1$ V is more significant than that of Cu, suggesting the existence of more H_{ad} on the surface of $Ni(OH)_x/Cu$. Correspondingly, $Ni(OH)_x/Cu$ shows a higher NH_3 yield rate and FE than those of Cu. These results shows that the $Ni(OH)_x$ deposited on Cu surface can accelerate the water dissociation process and produce more H_{ad} than bare Cu.

Figure R1-1. Comparison of the LSV polarization curves over Cu and $Ni(OH)_x/Cu$ in electrolytes using H_2O or D_2O as the solvent without (a) and with (b) adding 0.1 M NO_3^- . KIE of H/D for NH_3 synthesis over Cu and $Ni(OH)_x/Cu$ and the corresponding FE of NH_3 in the presence of 0.1 M NO_3^- (c). CV curves in Ar-saturated 1 M KOH with a scan rate of $100\ mV\ s^{-1}$ (d).

In order to further correlate the amounts of generated H_{ad} over Cu and $Ni(OH)_x/Cu$ with their eNO_3^-RR performance, we conduct EPR measurements using DMPO as a H_{ad} trapping reagent under different applied potentials. As the applied potential on the Cu and $Ni(OH)_x/Cu$ electrodes gradually increases, the signal of DMPO-H correspondingly intensifies, indicating the augmented generation of H_{ad} on the electrode surfaces (Figure R1-2). Notably, $Ni(OH)_x/Cu$ exhibits a stronger DMPO-H signal than Cu at the same potential, signifying an enhanced ability of $Ni(OH)_x/Cu$ to drive the H_2O dissociation and generate H_{ad} .

The conversion from NO_3^- to NO_2^- or NH_3 on Cu is highly dependent on the amount of H_{ad} . In the initial submission, we have tested FEs for NO_2^- and NH_3 production on Cu and $Ni(OH)_x/Cu$ under different potentials (Figure R1-3). The FEs of NO_2^- over Cu reach over 50% when the overpotentials are less than -0.2 V, under which fewer H_{ad} on the Cu surface cannot meet the H_{ad} demand for hydrogenation of NO_3^- toward NH_3 , leading to accumulation of NO_2^- . As the amount of H_{ad} on Cu surface increases under a more negative potential, the FEs of NO_2^- decreases. Notably, as for the $Ni(OH)_x/Cu$, the FEs of NO_2^- are significantly lower than those of Cu under the same overpotential due to more H_{ad} can be generated over the $Ni(OH)_x/Cu$ surface.

Figure R1-2. DMPO-involved EPR spectra of Cu (a) and $Ni(OH)_x/Cu$ (b) at different applied potentials.

Figure R1-3. Potential-dependent FEs of NH_3 and NO_2^- in eNO_3^-RR tests over the Cu (a) and the $Ni(OH)_x/Cu$ (b).

(2) For the second question:

ii) As for cation effect, it is very interesting. Independently if more detailed investigation is included, it can be full another story. That would be great if more insightful explanation, like DFT is provided, like this has been done on CO₂/CO reduction on transition metals, of course, in order to do this, detailed active sites is demanded.

Thank authors' effort for adding more simulations, but 1) why do u fix all the cations on the same distance above metal surfaces? For different cations due itself size, the distance to metal surface is different, right? Besides, in the literature [Chem. Sci., 2022, 13, 7634–7643], for example, the Na⁺ is suggested remaining near the surface around 3 Å. Besides, this cations distribution should be system dependent and potential dependent [J. Phys. Chem. C 2020, 124, 37, 20055–20065]. When it comes to the Ni(OH)_x/Cu system, the cations distribution might be totally different. For now the present simulation is not enough. Besides, the construction of the system Ni(OH)_x/Cu, especially, Ni(OH)_x part is not convincing. Additionally, why authors calculate the adsorption of H₂O to explain the effect from cations on NH₃ formation?

Response: Thanks to the reviewer for this valuable comment. We agree with the reviewer that the distance between the cation and the catalyst surface is determined by the type of cation, system environment and overpotential applied on the electrode. To obtain the molecular insights of the structural behaviors of cations in the vicinity of Cu and Ni(OH)_x/Cu, classic molecular dynamics (MD) simulations were performed. MD simulation boxes with dimensions of around 40 × 40 × 100 Å³ were created, in which the solid substrate is the Cu (111) or the Ni(OH)_x/Cu. Meanwhile, 1.1 M Li⁺, 1.1 M Na⁺, 1.1 M K⁺, or 1.1 M Cs⁺ aqueous solution was established to simulate the electrolyte environment. More parameter setting for MD simulations are described at the end of this response.

MD simulation results show that the solvation environment of cations at the electrified interface induced by noncovalent interactions is cation-dependent. As shown in Figure R1-4, The z-axial cation number density profiles from MD simulations reveal increasing peak intensity in the order of Li⁺ < Na⁺ < K⁺ < Cs⁺, indicating the greater willingness of large, weakly solvated cations (like Cs⁺) to approach the electrode surface. The highest peak centers for cations on Ni(OH)_x/Cu (Li⁺: 6.4 Å, Na⁺: 6.9 Å, K⁺: 4.9 Å, and Cs⁺: 2.9 Å) are closer to the surface than those on Cu (Li⁺: 6.8 Å, Na⁺: 7.2 Å, K⁺: 5.5 Å, and Cs⁺: 3 Å). In addition, the number densities corresponding to the highest peak for Li⁺, Na⁺, K⁺, and Cs⁺ on Ni(OH)_x/Cu (1.2, 1.3, 1.7, and 2.8, respectively) are larger than those of on Cu (1.0, 1.1, 1.5, and 2.5, respectively). These results suggest that Ni(OH)_x species on Cu can attract cations closer to the electrified

interface. Therefore, the following DFT calculations were carried out based on the DM analysis results, in which the highest peak centers in Figure R1-4 were considered to be the distance between the cation hydrates and the electrocatalyst surface (Cu or Ni(OH)_x/Cu).

Figure R1-4. Density profiles of Li⁺, Na⁺, K⁺, and Cs⁺ over the Cu (a) and the Ni(OH)_x/Cu (b).

The cation–water clusters can be varied in different systems. To shed more light on such cation–water clusters, we calculated the average radial distribution functions (RDF) of the cation and the surrounding water molecule (the O atom in the water molecule). The MD analysis results are shown in Figure R1-5. On the Cu surface, there are clear peaks in the cation–O RDF within 2.7 Å, 3.2 Å, 3.7 Å, and 4.1 Å cut-off for Li⁺, Na⁺, K⁺, and Cs⁺, respectively. These values become 2.7 Å, 3.1 Å, 3.6 Å, and 3.9 Å on the Ni(OH)_x/Cu, respectively. It is worth noting that these results are consistent with the previous reference using *ab initio* molecular dynamics (AIMD) simulations [*J. Mater. Chem. A* **2020**, 8, 24428–24437], which reported cut-off values of 2.5 Å, 3.0 Å, 3.6 Å, and 4.0 Å for Li⁺, Na⁺, K⁺, and Cs⁺ on (111) plane of Cu electrode, respectively. The pronounced first peak and deep minimum indicate a highly structured first solvation shell [*Mol. Phys.* **2014**, 112, 1448-1456]. The integrated quantities of these first peaks are used to calculate the average coordination numbers of water molecule around cations in the first solvation shell. Integration up to the first minimum of the cation–O RDF gives the coordination numbers of 4.04, 5.49, 6.86, and 9.17 for Li⁺, Na⁺, K⁺, and Cs⁺, respectively, when using Cu as the substrate. While these coordination numbers become 4.06, 5.48, 6.57, and 7.87 over the Ni(OH)_x/Cu, respectively.

Figure R1-5. Radial distribution function (RDF) profiles of Li⁺-, Na⁺-, K⁺- and Cs⁺-O (in H₂O) when the solid substrates are Cu (a) and Ni(OH)_x/Cu (b), as well as their coordination number as a function of radial distance.

In regards to the second question, we added more comparison and screening of Ni(OH)_x structural models. We constructed and optimized three Cu(111) models loaded with different numbers of Ni(OH)₂ molecular units. As shown in Figure R1-6 and Table R1-1, we observed that the coordination numbers for Ni-O, Ni-Cu, and Ni-Ni are 3, 2, and 1 for Ni₂(OH)₄/Cu, 3.7, 1, and 2 for Ni₃(OH)₆/Cu, and 3.5, 0, and 2.5 for Ni₄(OH)₈/Cu, respectively. Compared with Ni(OH)₂ crystal with definite crystal structure, amorphous Ni(OH)_x has more active chemical properties and excellent H₂O adsorption and dissociation site (Ni atom) on the surface of Cu catalyst. However, it is difficult to accurately obtain the chemical structure of Ni(OH)_x. Therefore, by referring to many reported literatures [*Angew. Chem. Int. Ed.* **2022**, 61, e202207512; *Angew. Chem. Int. Ed.* **2023**, 62, e202301957], the coordination environment around Ni atoms in Ni(OH)_x was obtained by EXAFS and XPS analysis, and then established the Ni₃(OH)₆/Cu model as a reasonable DFT calculation model for studying local catalytic reactions. This model (Ni₃(OH)₆/Cu) can reflect the coordination environment of Ni and reveal the reaction of reactive molecules (NO₃⁻ or H₂O) on Ni(OH)_x/Cu.

Figure R1-6. Atomic configurations of $\text{Ni(OH)}_x/\text{Cu}$ with two (a, b), three (c, d), and four (e, f) molecular units of Ni(OH)_2 on the $\text{Cu}(111)$ surface.

Table R1-1. Coordination number (CN) of Ni-O, Ni-Cu, and Ni-Ni in experimentally obtained $\text{Ni(OH)}_x/\text{Cu}$ and different models.

Sample	CN of Ni-O	CN of Ni-Cu	CN of Ni-Ni
$\text{Ni(OH)}_x/\text{Cu}$	4.6 ± 0.3	2.2 ± 0.3	4.8 ± 0.7
$\text{Ni}_2(\text{OH})_4/\text{Cu}$	3	2	1
$\text{Ni}_3(\text{OH})_6/\text{Cu}$	3.7	1	2
$\text{Ni}_4(\text{OH})_8/\text{Cu}$	3.5	0	2.5

Then, based on the MD simulation results, we performed DFT calculations to reveal the effects of different cations on the H_2O adsorption and dissociation processes. The distances between the cation hydrates and the first Cu layer were determined by the cation number density profiles shown in Figure R1-4. We also adjusted the water coordination numbers around cations based on the integration of cation-O RDF in Figure R1-5. Specifically, we set the water molecule numbers as 4, 5, and 7 for Li^+ , Na^+ , and K^+ cations, respectively. It is worth noting that the large and weakly solvated Cs^+ cations would accumulate above the surface of Cu or $\text{Ni(OH)}_x/\text{Cu}$ at a distance of about 3 Å, blocking the active sites and reducing the performance of eNO_3^-/RR . Therefore, Cs^+ ions will be excluded when considering the influence of cation effect on the H_2O adsorption and dissociation processes. The models constructed for the calculations and the atomic configurations of water molecules adsorption and dissociation on Cu and $\text{Ni(OH)}_x/\text{Cu}$ in the presence of different cation hydrates are shown in Figure R1-7 and R1-8, and the corresponding Gibbs free energies obtained are shown in Figure R1-9.

On the Cu surface, the Gibbs free energies for water adsorption ($* \rightarrow * \text{H}_2\text{O}$) in the presence of $\text{Li}^+(\text{H}_2\text{O})_4$, $\text{Na}^+(\text{H}_2\text{O})_5$, and $\text{K}^+(\text{H}_2\text{O})_7$ were -0.058 eV, -0.14 eV, and

−0.282 eV, respectively, whereas the subsequent water dissociation process on Cu ($*\text{H}_2\text{O} \rightarrow *\text{OH}-*\text{H}$) delivered Gibbs free energy uphill of 0.205 eV, 0.436 eV, and 0.668 eV. These results demonstrate that water adsorption on the Cu surface became more favorable with the change of the cation from Li^+ to K^+ , while the water dissociation process became more unfavorable. Thus, these two opposite influences eventually make the cation effect on the generation of H_{ad} on the Cu surface very limited.

On the $\text{Ni}(\text{OH})_x/\text{Cu}$ surface, the Gibbs free energies of water adsorption ($* \rightarrow *\text{H}_2\text{O}$) were −0.197 eV, −0.245 eV, and −0.359 eV in the presence of $\text{Li}^+(\text{H}_2\text{O})_4$, $\text{Na}^+(\text{H}_2\text{O})_5$, and $\text{K}^+(\text{H}_2\text{O})_7$, respectively. The subsequent water dissociation process on $\text{Ni}(\text{OH})_x/\text{Cu}$ ($*\text{H}_2\text{O} \rightarrow *\text{OH}-*\text{H}$) delivered Gibbs free energy uphill of 0.124 eV, 0.115 eV, and 0.104 eV. Compared with Cu, $\text{Ni}(\text{OH})_x/\text{Cu}$ shows more Gibbs free energy downhill of the water adsorption process and less Gibbs free energy uphill of the water dissociation process in the presence of the same cation hydrate, indicating that the water adsorption and dissociation on $\text{Ni}(\text{OH})_x/\text{Cu}$ are more favorable. Notably, the Gibbs free energy for water adsorption and the energy barrier for water dissociation on $\text{Ni}(\text{OH})_x/\text{Cu}$ both decreases in the order of $\text{Li}^+ > \text{Na}^+ > \text{K}^+$, indicating that the water adsorption and dissociation processes become more favorable with increasing cation size. This cation effect is well aligned with the experimentally observed cation-dependent HER and eNO_3^- RR activities.

Figure R1-7. Atomic configurations of water adsorption and dissociation on Cu in the presence of $\text{Li}^+(\text{H}_2\text{O})_4$ (a-c), $\text{Na}^+(\text{H}_2\text{O})_5$ (d-f), and $\text{K}^+(\text{H}_2\text{O})_7$ (g-i).

Figure R1-8. Atomic configurations of water adsorption and dissociation on $\text{Ni(OH)}_x/\text{Cu}$ in the presence of $\text{Li}^+(\text{H}_2\text{O})_4$ (a-c), $\text{Na}^+(\text{H}_2\text{O})_5$ (d-f), and $\text{K}^+(\text{H}_2\text{O})_7$ (g-i).

Figure R1-9. Gibbs free energy diagram of water adsorption and dissociation processes on Cu and $\text{Ni(OH)}_x/\text{Cu}$ in the presence of different cation–water clusters.

Details of the MD simulations:

Periodic boundary conditions (PBCs) are imposed in the two orthogonal (x and y) directions to mimic infinite planar Cu crystalline substrate, while a wall-boundary condition is applied in the out-of-plane (z) direction of substrate. The forcefield

parameters of as-investigated systems are taken from literature [*JACS Au* **2021**, 1, 1674–1687]. For the non-bonded atomic interactions in the system, the 12-6 Lennard-Jones potential with cutoff distance of 10.0 Å is applied to describe the van der Waals (vdW) forces between atoms, while the standard Coulomb potential is utilized to mimic the electrostatic interactions that is evaluated by the particle–particle particle–mesh (PPPM) algorithm. To satisfy an imposed voltage of –0.25 V vs. RHE across the systems along the z-direction, the charges of each Cu atom are computed at each timestep using the constant potential fix in the Large-scale Atomic/Molecular Massively Parallel Simulator (LAMMPS) [*J. Comput. Phys.* **1995**, 117, 1-19]. Prior to MD simulations, energy minimizations are firstly performed to relax the configuration of as-investigated systems with energy and force tolerances of 0.0001 Kcal/mol and 0.0001 Kcal/(mol·Å), respectively. Then, MD simulations with 1,000,000 timesteps are carried out to further relax the systems at temperature of 300 K under canonical (NVT) ensemble, in which the temperature is maintained by the Nose-hoover thermostat. Finally, production MD simulations with 1,000,000 timesteps are performed to capture the structural behaviors of ions and water in the vicinity of surface of Cu-based substrates. During the whole MD simulations, the Cu-based substrates is frozen. The dynamics of atoms in the solution are on the basis of the classical Newton’s motion, in which the velocity-Verlet algorithm with a timestep of 1.0 fs is applied to integrate the classic Newton’s equation. All the MD simulations are implemented using the LAMMPS package.

(3) For the question: “In the study, for the impact from cations (Li^+ , Na^+ , K^+ , Cs^+), authors claimed ‘The $\text{M}^+(\text{H}_2\text{O})_n$ (where $n = 22, 13, 7,$ and 6 for Li^+ , Na^+ , K^+ , and Cs^+ , respectively) with smaller n and radii is believed to have a stronger interaction with anionic species and thus a greater ability to activate H_2O ’. From this, Cs^+ and K^+ should have similar effect on the NO_3^- reduction. however, it seems for Cs^+ , it doesn’t follow the trend and actually deactivates the NH_3 formation, what is the possible explanation for this?”

Authors mentioned that: ‘the concentration of alkali metal cations is very high (1 M CsOH plus 0.1 M CsNO₃), potentially leading to direct adsorption of Cs^+ on the catalyst surface ...’, however, the potential for the cations’ adsorption or not is more important factor. Then as for Cs poisoning and blocking the active site, you can’t just guess but to prove. For now, DFT simulation doesn’t contribute to this key experimental observations (cations effect on NH_3 formation).

Response: Thanks to the reviewer for this valuable comment. As shown in Figure R1-4, the number density for surface cations from MD simulations revealed increasing

peak intensity in the order of $\text{Li}^+ < \text{Na}^+ < \text{K}^+ < \text{Cs}^+$, corresponding to more Cs^+ in the vicinity of the electrified interface than K^+ , Na^+ , and Li^+ . More importantly, Cs^+ is mainly distributed at a distance of about 3 Å from the surface, and also appears at a distance less than 3 Å, indicating that Cs^+ can be adsorbed on the surface of Cu and $\text{Ni}(\text{OH})_x/\text{Cu}$ to block the active sites. The snapshots of the MD simulations shown in Figure R1-10 also provide a direct vision of the close distribution of Cs^+ to the catalyst surface. Thus, the active site blocking caused by Cs^+ makes it unfavorable to NH_3 production.

Figure R1-10. Snapshots of spatial distribution of Li^+ , Na^+ , K^+ , and Cs^+ on Cu (a-d) and $\text{Ni}(\text{OH})_x/\text{Cu}$ (e-h).

Reviewer #2

The authors have well revised the manuscript according to the comments, and I recommend the publication of this manuscript in Nature Communications.

Response: We sincerely appreciate the reviewer for the time and efforts spent in evaluating our manuscript.

Reviewer #3

In the revised manuscript, the authors have made significant efforts to elucidate the structure and composition of catalysts and evaluate the energy efficiency of the tandem pNOR-NO₃⁻RR scheme. I have the following questions that worth more explanation.

Response: We sincerely appreciate the reviewer for the time and efforts spent in evaluating our manuscript. We have provided point-to-point responses to the questions raised by the reviewers below.

1. *According to ICP, there is only a small, auxiliary amount of Ni with large, major portions of Cu in Ni(OH)_x/Cu. However, it seems Ni(OH)_x forms a separate, exterior covering layer outside Cu based on Figure 1d. Does the difference of deposition time finally affect the coverage and thickness of Ni(OH)_x layer?*

Response: Thank you for the constructive comment from the reviewer. We believe that the primary influence of varying deposition time lies in the coverage of Ni(OH)_x on Cu. Various facets of Cu single crystals exhibit distinct hydroxide electrosorption (OH_{ad}) peaks at different potentials in voltammograms, allowing for the use of OH_{ad} to probe the surface structures of polycrystalline Cu electrodes. Voltammograms for OH_{ad} analysis were recorded in the electrolysis cell using Ar purged 1 M KOH as the electrolyte. As shown in Figure R3-1, the voltammograms measured for the Cu nanowires exhibit a series of reversible peaks in the potential region of 0.35–0.50 V, which can be assigned to the features of low-index facets of face-centered cubic (fcc) Cu (i.e., ~0.38 V for (100), ~0.43 V for (110), and ~0.49 V for (111)) [*ACS Catal.* **2017**, *7*, 4467–4472]. As the deposition time increases, the OH_{ad} peaks gradually diminish, indicating that an increasing proportion of the Cu surface is being covered by Ni(OH)_x.

Figure R3-1. Voltammograms of OH_{ad} peaks collected in an Ar-purged 1 M KOH electrolyte with a scan rate of 50 mV s^{-1} .

2. *In calculating the energy efficiency of the tandem pNOR- NO_3^- RR process, the ΔH and ΔG of the overall reaction from N_2 , H^+ and H_2O to NH_4^+ and O_2 was evaluated. Here, is it more rational to adopt NH_3 instead of NH_4^+ as the product, given that the electrolysis was done under basic conditions with excessive NaOH/KOH, while ammonia is more inclined to exist as NH_3 instead of NH_4^+ at a pH above 9-10?*

Response: Appreciation is extended to the reviewer for their valuable suggestion. In response to the feedback received, we have replaced NH_4^+ (aq) with NH_3 (g) in the following reaction equations and recalculated the energy efficiency of the pNOR- eNO_3^- RR tandem system. The specific procedure is outlined below.

Calculation of energy efficiency:

The reaction of the plasma-assisted nitrogen oxidation is described by Equation R3-1.

The reaction in the electrochemical side is described by Equation R3-2.

$$\Delta H^0 = 503.56 \text{ kJ/mol of } \text{NH}_3$$

$$\Delta G^0 = 411.956 \text{ kJ/mol of } \text{NH}_3$$

Therefore, the overall reaction for the pNOR-eNO_x⁻RR system is:

$$\Delta H^0 = 382.845 \text{ kJ/mol of } \text{NH}_3$$

$$\Delta G^0 = 339.267 \text{ kJ/mol of } \text{NH}_3$$

Table R3-1. Enthalpy (ΔH_f^0) and Gibbs free energy (ΔG_f^0) of the reactants and products.

Compounds	ΔH_f^0 (kJ/mol)	ΔG_f^0 (kJ/mol)
N ₂ (g)	0	0
O ₂ (g)	0	0
H ₂ O (l)	-285.83	-237.178
OH ⁻ (aq)	-229.6	-157.2
NO ₃ ⁻ (aq)	-207.4	-111.3
NH ₃ (g)	-45.9	-16.5

Hence, the energy efficiency for the pNOR-eNO_x⁻RR tandem system is:

$$\eta = \frac{382.845}{18362.2} \times 100\% = 2.08\%$$

3. In tandem pNOR-NO₃⁻RR, the revised manuscript discussed how the solution conductivity and current density varied after long hours. Here, I wonder whether the following two factors are in effect: a) as the OH⁻ reacts with nitric oxides generated in pNOR, the generated NO_x⁻ ions would have lower ion mobilities compared with OH⁻, which in turn increase the solution resistance? b) as the air after pNOR is purged into the NaOH/KOH sorbent, the consumption of OH⁻ cause a major pH drop of the electrolyte. This pH drop is increasingly significant if untreated air is used, because the small amounts of CO₂, although only exist in hundreds of ppm, are cumulatively captured by OH⁻ and forms carbonates: As far as I have observed recently, when 1.0 M NaOH was used to treat the mixture gas in pNOR using ambient air, the solution pH dropped to about 11 at room temperature when the NO_x⁻ concentration reached about 0.5 M, which would be above 13 (50% OH⁻ consumption) if we only take the reaction between OH⁻ and NO_x into consideration. Since the acidity of HCO₃⁻ is much weaker than HNO₂, the existence of carbonates does not affect the sorbent's ability to capture NO_x⁻, eventually obtained ~1.0 M NO_x⁻ when the pH dropped to ~8. However, the major pH drops due to CO₃²⁻ in long-hour pNOR would be a non-negligible factor for electrochemical properties.

Response: We express our sincere gratitude for the reviewer's comments. We monitored the pH variation of the catholyte during the continuous operation of the pNOR-eNO₃⁻RR tandem system, as illustrated in Figure R3-2. Despite a sustained increase in the concentration of NO_x⁻ in the electrolyte over time, the pH of the catholyte exhibited no discernible changes throughout the entire reaction process. This phenomenon can be attributed, on the one hand, to the reduction of NO₃⁻ and NO₂⁻ to NH₃, generating 9 and 7 OH⁻ ions, respectively (R3-4 and R3-5). On the other hand, NH₃ produced in the catholyte can undergo hydrolysis in alkaline media, contributing to the generation of OH⁻ ions (R3-6).

Furthermore, we investigated the pH variation in the NO_x absorption solution solely during the operation of pNOR. We observed a continuous decrease in the pH value over time. Concurrently, we recorded the electrochemical impedance spectroscopy (EIS) and measured the solution resistance of the absorption solution. As depicted in Figure R3-3, we noted an increase in solution resistance as OH⁻ ions reacted with NO_x to form NO_x⁻. Since the mole conductivities of NO₂⁻ (71.7 S cm² mol⁻¹) and NO₃⁻ (71.44 S cm² mol⁻¹) are smaller than that of OH⁻ (198.0 S cm² mol⁻¹), it can be inferred that the accumulation of NO_x⁻ with low mole conductivities contributes to the observed increase in solution resistance during the operation of the pNOR-eNO₃⁻RR tandem system.

Figure R3-2. pH variations of the catholyte in pNOR-eNO_x⁻RR tandem system and absorption solution of pNOR without eNO_x⁻RR.

Figure R3-3. EIS obtained in the absorption solution of pNOR without eNO_x^-RR .

4. *In the last sentence of the section on "Catalyst synthesis and characterization", it should be "Ni(OH)_x/CuO and Ni(OH)_x/Cu electrodes" instead of "Ni(OH)_x/Cu and Ni(OH)_x/Cu electrodes". Please check the manuscript carefully.*

Response: Thanks for your kind comment. We have replaced the statement of “Ni(OH)_x/Cu and Ni(OH)_x/Cu electrodes” with “Ni(OH)_x/CuO and Ni(OH)_x/Cu electrodes” in the revised manuscript.

5. *Except for FIG. 2a, the control group in FIG. 2 consists of three groups: Cu, Ni(OH)₂, and Ni(OH)_x/Cu. Consistency is recommended.*

Response: Thanks to the reviewer for this valuable comment. The polarization curves of Ni(OH)₂ with or without the presence of NO_3^- have been supplemented. Figure R3-4 has been added in the revised manuscript.

Figure R3-4. The polarization curves of Cu, Ni(OH)₂, and Ni(OH)_x/Cu with or without the presence of NO_3^- .

6. According to Figure S24, the test time affects FE (NH₃). So, the Faraday efficiency and yield in Figure 2 were tested after how long the reaction took?

Response: All the batch experiments were tested for 30 min.

7. The authors mentioned “As shown in Fig. 3e, when using ¹⁵NO₃⁻ as the nitrogen source, only doublet peaks of ¹⁵NH₄⁺ are detected in the ¹H nuclear magnetic resonance (NMR) spectra of the electrolyte without seeing any triple coupling peaks of ¹⁴NH₄⁺”. The correct reference for the figure should be “Fig.3e” instead of “Fig.2e”.

Response: Thanks for your kind comment. We have corrected our statement in the revised manuscript.

8. Although the surface morphology and structure of the electrode material remain unchanged after a long time of testing, is there any material leaching from the electrode into the electrolyte?

Response: Thanks for this valuable comment. As shown in Table R3-2, we only observed a ppb-level of Ni²⁺ and Cu²⁺ concentrations in the catholyte after the 100-h test, indicating that the material leaching is negligible.

Table R3-2. Concentrations of Ni²⁺ and Cu²⁺ in the catholyte after the 100-h of pNOR-eNO_x⁻RR operation.

Reaction time (h)	Ni ²⁺ concentration (ppb)	Cu ²⁺ concentration (ppb)
100	2.49	1.88

9. Figure S45 indicates that the concentration of NO_x⁻ in the reaction solution exceeds 100 mM after 40 minutes, thus necessitating an evaluation of the eNO₃⁻RR activity of Ni(OH)_x/Cu in electrolytes with higher NO₃⁻ concentrations.

Response: Thanks for this valuable comment. We have conducted eNO₃⁻RR performance evaluation in electrolytes with higher NO₃⁻ concentrations of 1000 and 2000 M. As shown in Figure R3-5, when the concentration of NO₃⁻ becomes exceptionally high, the deoxygenation reaction of NO₃⁻ to NO₂⁻ intensifies, resulting in a notable decrease in the FE and selectivity of NH₃ production.

Figure R3-5. LSV curves of Ni(OH)_x/Cu in 1 M KOH containing different concentrations of NO₃⁻ at a scan rate of 5 mV s⁻¹ (a). I-t curves of Ni(OH)_x/Cu under different concentrations of NO₃⁻ at -0.25 V (vs. RHE) (b). The FE of NH₃, NH₃ yield rate, and the ratio of the generated NH₃ concentration [NH₃] to the consumed NO₃⁻ concentration [NO₃⁻] over Ni(OH)_x/Cu at -0.25 V vs. RHE at [NO₃⁻] in the range of 1-2000 mM in 1 M KOH (c).

REVIEWER COMMENTS

Reviewer #1 (Remarks to the Author):

Reviewer Comment to

Manuscript Number: NCOMMS-23-27437

Title: Efficient ammonia synthesis from the air using tandem non-thermal plasma and electrocatalysis at ambient conditions

Summary:

Thank all authors for the detailed reply on our comments, It adds a lot of work and effort and I really appreciate that. However, there are some critical questions or suggestions for this manuscript from my aspect, as follows.

Major comments:

(1) Thank authors for further providing information possibly related to 'The eNO₃-RR performance over the Cu surface is closely related to the supply of adsorbed hydrogen species, ..'. No doubt here, the presence of Ni(OH)_x provides more adsorbed H on the surface. Is there also probability for Ni(OH)_x promotes the N-O cleavage in NO₃⁻/NO₂⁻ which Cu is not good at it? For NO₂⁻/NO reduction Cu is a very good catalyst. This indicates Cu is not that bad to have adsorbed H on the surface for electro-reductions. Is it more concerned about the conversion from NO₃⁻ towards NO₂⁻? For this, I would suggest authors do the NO₂⁻/NO reduction on Ni(OH)_x/Cu, Cu and Ni(OH)_x to compare their efficiency for NH₃ formation.

(2) For the energy diagram in Fig. 4a, what is your reference? What are your corrections for calculating Gibbs free energy? Why do you get positive adsorption energy of NO₃⁻ on Cu? From previous publication: (figure 5c in Angew. Chem. Int. Ed.2020,59, 5350 – 5354; figure 3a in Angew. Chem. 2021, 133, 22137 – 22143), the adsorption energy of NO₃⁻ on Cu demonstrates a negative value. In addition, the protonation of *NO forming *NOH seems more stable (ACS Catal.2021, 11, 14417–14427; figure 5c in Angew. Chem. Int. Ed.2020,59, 5350 – 5354; figure 3 in SI of Angew. Chem. 2021, 133, 22137 – 22143). All these difference make it demanding more detailed computational details. I appreciate authors provide the figure 60 in SI, but more information is needed, for example the values for Delta G(protonation), Delta G(vap).

(3) In figure 2(a) and (d), the current density for NO₃⁻ reduction on Ni(OH)₂ is almost zero while in Figure 2(b) and (c) the FE and yield is not. What is reason for this?

(4) In the manuscript, authors only provide the FE/yield of NH₃ formation upto -0.3 V vs. RHE, what about upto -0.6V or even -0.8V, are the FE on Cu and Ni(OH)_x getting close to similar? And also what about the yields?

Reviewer #3 (Remarks to the Author):

In the revised manuscript, the authors have made efforts to answer the reviewers' questions and added relevant content to the revised manuscript. Now I think the manuscript can be published after minor

revision noted below.

1. When authors evaluated catalysts' eNO_x-RR performance, either electrochemical surface area or geometrical area was used, this discrepancy made readers confused.
2. S fig 32 didn't present intrinsic activity of Ni(OH)_x/Cu for NO₂-RR.
3. In S table 3, why was apparent activity used to compare different types of catalysts?
4. As authors mentioned "we hypothesize that the improved eNO₃-RR performance on Ni(OH)_x/Cu is due to its enhanced water activation". However, the eNO₂-RR performance on Ni(OH)_x/Cu just improved slightly compared with Cu.

Response to reviewers' comments:

We are grateful for the insightful feedback and constructive comments provided by the reviewers. Their thorough review and thoughtful suggestions will undoubtedly enhance the clarity and rigor of the research presented. We have carefully revised our manuscript in response to their comments and highlighted the changes in yellow. Below, we provide point-to-point responses to the reviewers' comments. The original comments are presented in blue italics, while our responses are in black.

Reviewer #1

Thank all authors for the detailed reply on our comments. It adds a lot of work and effort and I really appreciate that. However, there are some critical questions or suggestions for this manuscript from my aspect, as follows.

Response: Thank you for your acknowledgment of our detailed reply to your comments. We genuinely appreciate your recognition of the additional work and effort we put into addressing your feedback. We understand that you have some critical questions or suggestions regarding the manuscript, and we are eager to address them thoroughly. Your insights are invaluable to us, and we are committed to ensuring that the manuscript meets the highest standards of quality and rigor.

Major comments:

(1) Thank authors for further providing information possibly related to 'The eNO_3^-RR performance over the Cu surface is closely related to the supply of adsorbed hydrogen species, ...'. No doubt here, the presence of $Ni(OH)_x$ provides more adsorbed H on the surface. Is there also probability for $Ni(OH)_x$ promotes the N-O cleavage in NO_3^-/NO_2^- which Cu is not good at it? For NO_2^-/NO reduction Cu is a very good catalyst. This indicates Cu is not that bad to have adsorbed H on the surface for electro-reductions. Is it more concerned about the conversion from NO_3^- towards NO_2^- ? For this, I would suggest authors do the NO_2^-/NO reduction on $Ni(OH)_x/Cu$, Cu and $Ni(OH)_x$ to compare their efficiency for NH_3 formation.

Respond: Thanks to the reviewer for this valuable comment. Electrochemical NO_2^- reduction (eNO_2^-RR) performance on Cu, $Ni(OH)_x/Cu$, and $Ni(OH)_2$ were evaluated. As shown in Figure R1-1, $Ni(OH)_2$ shows the worst apparent and intrinsic activity for NO_2^- reduction. Additionally, $Ni(OH)_2$ also exhibit poor intrinsic activity in eNO_3^-RR , indicating that $Ni(OH)_x$ species have no promotion effect on the N-O cleavage.

Furthermore, according to the calculated Gibbs free energies (ΔG) of $^*\text{NO}_3$, $^*\text{NO}_2$, and $^*\text{NO}$ on Cu and $\text{Ni}(\text{OH})_x/\text{Cu}$ (Table R1-1), the first deoxygenation step ($^*\text{NO}_3 \rightarrow ^*\text{NO}_2$) and the second deoxygenation step ($^*\text{NO}_2 \rightarrow ^*\text{NO}$) on Cu have ΔG value of -2.06 and -1.26, respectively, which are more negative than those on $\text{Ni}(\text{OH})_x/\text{Cu}$ (-2.00 for $^*\text{NO}_3 \rightarrow ^*\text{NO}_2$ and -0.68 for $^*\text{NO}_2 \rightarrow ^*\text{NO}$), indicating the deoxygenation processes (N-O cleavage) from $^*\text{NO}_3$ to $^*\text{NO}$ on Cu is more exothermic than those on $\text{Ni}(\text{OH})_x/\text{Cu}$. Therefore, the presence of $\text{Ni}(\text{OH})_x$ on Cu enhances eNO_x^- RR performance by accelerating water activation instead of promoting N-O cleavage.

Figure R1-1. Potential-dependent eNO_2^- RR performance over Cu (a), $\text{Ni}(\text{OH})_x/\text{Cu}$ (b), and $\text{Ni}(\text{OH})_2$ (c). ECSA-normalized NH_3 yield rates of different catalysts (d).

Table R1-1. Calculated Gibbs free energies (ΔG) of adsorbed species of $^*\text{NO}_3$, $^*\text{NO}_2$, and $^*\text{NO}$ on Cu and $\text{Ni}(\text{OH})_x/\text{Cu}$ surface.

	Cu	$\text{Ni}(\text{OH})_x/\text{Cu}$
$\text{NO}_3^-(\text{l}) + ^*$ (reference)	0	0
$^*\text{NO}_3$	0.48	-0.27
$^*\text{NO}_2$	-2.06	-2.00
$^*\text{NO}$	-1.26	-0.68

(2) For the energy diagram in Fig. 4a, what is your reference? What are your corrections for calculating Gibbs free energy? Why do you get positive adsorption energy of NO_3^- on Cu? From previous publication: (figure 5c in Angew. Chem. Int.

*Ed.2020,59, 5350 – 5354; figure 3a in Angew. Chem. 2021, 133, 22137 – 22143), the adsorption energy of NO₃⁻ on Cu demonstrates a negative value. In addition, the protonation of *NO forming *NOH seems more stable (ACS Catal.2021, 11, 14417–14427; figure 5c in Angew. Chem. Int. Ed.2020,59, 5350 – 5354; figure 3 in SI of Angew. Chem. 2021, 133, 22137 – 22143). All these difference make it demanding more detailed computational details. I appreciate authors provide the figure 60 in SI, but more information is needed, for example the values for Delta G (protonation), Delta G (vap).*

Response: Thanks to the reviewer for these constructive comments.

(For the first question) In the calculation of the Gibbs free energy diagram of eNO₃⁻RR, we mainly referred to the publications of *Phys. Chem. Chem. Phys.* **2013**, 15, 3196-3202, *J. Am. Chem. Soc.* **2020**, 142, 5702-5708, and *ACS Catal.* **2021**, 11, 14417-14427 for nitrate adsorption, deoxygenation, and hydrogenation calculations.

(For the second question) In detail, all Gibbs free energy values were referenced to the computational hydrogen electrode (CHE) model using the proton-coupled electron transfer (PCET) approach [*Phys. Rev. Lett.* **2007**, 99, 126101; *J. Phys. Chem. B* **2004**, 108, 17886-17892]. The calculations were performed under conditions of pH = 0 and U (applied potential) = 0 V vs. the standard hydrogen electrode. The chemical potential of the H⁺/e⁻ pair was considered as half of the H₂ gas molecule. The ΔG value was obtained as

$$\Delta G = \Delta E_{DFT} + \Delta(ZPE - TS)$$

where ΔE is the reaction energy difference between the product and the reactant occurring on catalysts. ΔZPE and ΔS are the changes in zero point energies and entropy at 298.15 K, which were calculated via considering only the vibrational frequencies. The Gibbs free energy of H* was calculated by

$$\Delta G_{H^*} = \Delta E_{H^*} + \Delta ZPE - T\Delta S$$

where ΔS was obtained by

$$\Delta S = S(H^*) - \frac{1}{2}S(H_2) \approx -\frac{1}{2}S(H_2)$$

This equation is obtained due to the negligible vibrational entropy of H*. At 300 K and 1 atm, TS(H₂) = 0.41 eV; thus, TΔS = -0.205 eV [*Adv. Energy Mater.* **2019**, 9, 1803369].

Figure R1-2 The thermodynamic cycle used to calculate the adsorption Gibbs free energy of NO_3^- in the aqueous phase.

The details for calculation of NO_3^- adsorption are described as follows.

The desired reaction for NO_3^- adsorption is

The thermodynamic cycle used to arrive at the required energetics is shown in Figure R1-2. The overall free energy of nitrate adsorption is calculated as

$$\Delta G_{\text{ads}}(\text{NO}_3^-) = G_{\text{NO}_3^*} + [G_{\text{H}^+} + G_{e^-}] - [G_{\text{NO}_3^-(\text{l})} + G_{\text{H}^+}] - G_*$$

where $G_{\text{NO}_3^*}$ is the energy of NO_3^* adsorbed on the catalytic surface, $G_{\text{NO}_3^-(\text{l})}$ is the free energy of aqueous NO_3^- . G_* is the energy of the respective surface, G_{e^-} is the free energy of elementary charge and G_{H^+} is the free energy of the proton. Here, we have neglected the rotational, translational, and vibrational free energies associated with adsorbed species, i.e., $G_{\text{NO}_3^*} = E_{\text{NO}_3^*}$, the value calculated from DFT. The zero-point energy correction for all species were not included.

Thus, the equation above can be converted into

$$\Delta G_{\text{ads}}(\text{NO}_3^-) = G_{\text{NO}_3^*} + \frac{1}{2} G_{\text{H}_2} - [G_{\text{HNO}_3(\text{g})} - \Delta G_{\text{vap}} - \Delta G_{\text{protonation}}] - G_*$$

where G_{H_2} and $G_{\text{HNO}_3(\text{g})}$ are the corresponding Gibbs free energies of H_2 and HNO_3 molecules in the gas phase at 300 K and 1 atm. The entropic (ΔS) and enthalpic (ΔH) contributions to the free energy of the gaseous species were obtained from JANAF thermodynamic tables [Allison, T. C. NIST-JANAF Thermochemical Tables - SRD 13. *National Institute of Standards and Technology, 2013*]. ΔG_{vap} is the free energy of vaporization of $\text{HNO}_3(\text{l})$ and is calculated from the Gibbs free energy difference between standard formation of HNO_3 in liquid and gas phases and has a value of 0.075 eV. $\Delta G_{\text{protonation}}$ is the free energy of the association of NO_3^- in solution with a proton, and is calculated to be 0.317 eV. These Gibbs free energies were all obtained from the

CRC handbook of chemistry and physics [Lide, D. R. CRC handbook of chemistry and physics. CRC press, **2004**]. Ultimately, the overall Gibbs free energy for NO₃⁻ adsorption from the solution phase ($\Delta G_{\text{ads}}(\text{NO}_3^-)$) was calculated as:

$$\begin{aligned}\Delta G_{\text{ads}}(\text{NO}_3^-) &= G_{\text{NO}_3^*} + \frac{1}{2}G_{\text{H}_2} - G_{\text{HNO}_3(\text{g})} - G_* + 0.392 \text{ eV} \\ &= E_{\text{NO}_3^*} + \frac{1}{2}E_{\text{H}_2} - E_{\text{HNO}_3(\text{g})} - E_* + 0.75 \text{ eV}\end{aligned}$$

Therefore, we applied 0.75 eV correction to compensating the DFT calculation.

(For the third question) The Gibbs free energy will be different when choosing HNO₃(g) or NO₃⁻(l) as the reference due to the corrections mentioned above. In the publication of *Angew. Chem. Int. Ed.* **2020**, 59, 5350–5354, the adsorption Gibbs free energy of nitrate was calculated with respect to NO₃⁻(g) instead of NO₃⁻(l). Thus, ΔG_{vap} and $\Delta G_{\text{protonation}}$ was not considered. In the publication of *Angew. Chem.* **2021**, 133, 22137–22143, the authors reported the adsorption energy of NO₃⁻ ($\Delta E_{\text{NO}_3^-}$) instead of the Gibbs free energy ($\Delta G_{\text{NO}_3^-}$). The Gibbs free energy changes of NO₃⁻ adsorption on Cu(111) in our work are align with the previous reports such as *J. Am. Chem. Soc.* **2020**, 142, 5702-5708 ($\Delta G_{\text{NO}_3^-} = 0.40 \text{ eV}$), *Energy Environ. Sci.* **2021**, 14, 4989-4997 ($\Delta G_{\text{NO}_3^-} = 0.46 \text{ eV}$), *Nat. Commun.* **2022**, 13, 7899 ($\Delta G_{\text{NO}_3^-} = 0.61 \text{ eV}$), *Nat. Commun.* **2022**, 13, 2338 ($\Delta G_{\text{NO}_3^-} = 0.624 \text{ eV}$), *Angew. Chem. Int. Ed.* **2022**, 61, e202202556 ($\Delta G_{\text{NO}_3^-} = 0.66 \text{ eV}$), *Energy Environ. Sci.* **2023**, 16, 2991-3001 ($\Delta G_{\text{NO}_3^-} = 0.3 \text{ eV}$). These fully justify the methodology and results of our calculations. The detailed computational methods have been supplemented in the revised Supplementary Information.

(For the fourth question) In performing the Gibbs free energy calculations for eNO₃⁻RR, we screened the adsorption configurations of each reaction intermediate. It has been reported in the literature that *NO hydrogenation can be on either O or N atoms [*Inorg. Chem. Front.* **2020**, 7, 4507-4516; *ACS Energy Lett.* **2023**, 8, 3, 1281–1288; *Nat. Commun.* **2024**, 15, 88]. In our results, the energy of *NHO is lower than that of *NOH. Therefore, we consider *NHO as a possible intermediate for eNO₃⁻RR.

(3) In figure 2(a) and (d), the current density for NO₃⁻ reduction on Ni(OH)₂ is almost zero while in Figure 2(b) and (c) the FE and yield is not. What is reason for this?

Response: In Figs. 2b and c, the display of LSV and NH₃ partial currents for Ni(OH)₂ looks near zero because the range shown in vertical coordinates is too large. Figure R1-3 shows the LSV curves with/without NO₃⁻ of Ni(OH)₂ and its potential-dependent NH₃ partial current density. The reaction current on the Ni(OH)₂ electrode in the presence of NO₃⁻ is larger than that in the absence of NO₃⁻, indicating that eNO₃⁻RR can proceed on Ni(OH)₂.

Figure R1-3 have been added in the revised Supplementary Information.

Figure R1-3. (a) LSV curves of Ni(OH)₂ in the presence or absence of NO₃⁻. (b) NH₃ partial current density over Ni(OH)₂ in eNO₃⁻RR.

(4) In the manuscript, authors only provide the FE/yield of NH₃ formation up to -0.3 V vs. RHE, what about up to -0.6 V or even -0.8 V, are the FE on Cu and Ni(OH)_x getting close to similar? And also what about the yields?

Response: Thanks to the reviewer for this valuable comment. Supplemented eNO₃⁻RR tests at potentials from -0.4 to -0.8 V vs. RHE over Cu and Ni(OH)_x/Cu were conducted. As shown in Figure R1-4, Ni(OH)_x/Cu electrode still delivers higher NH₃ yield rates than those of Cu at more negative potential. The FEs of Cu can reach over 90% when the overpotential exceeds -0.5 V. Due to more intensive HER, the FE of NH₃ over Ni(OH)_x/Cu at -0.8 V drops to 86.4%.

The Figure R1-4 has been added in the revised Supplementary Information.

Figure R1-4. Potential-dependent NH₃ yield rate (a) and corresponding FE of NH₃.

Reviewer #3

In the revised manuscript, the authors have made efforts to answer the reviewers' questions and added relevant content to the revised manuscript. Now I think the manuscript can be published after minor revision noted below.

Response: We sincerely appreciate the reviewer for the time and efforts spent in evaluating our manuscript. We have provided point-to-point responses to the questions raised by the reviewers below.

1. When authors evaluated catalysts' $e\text{NO}_x^-$ RR performance, either electrochemical surface area or geometrical area was used, this discrepancy made readers confused.

Response: Thanks to the reviewer for this detailed comment. Both electrochemical surface area and geometrical area-normalized activities are important in evaluating the performance of electrocatalysts. The electrochemical surface area-normalized activity needs to be illustrated along with the geometrical area-normalized activity to exclude the contribution of the electrochemical surface area and reveal the intrinsic activity. In the revised manuscript, the electrochemical surface area-normalized activity was qualitatively compared and geometrical activity was quantitatively illustrated.

2. S fig 32 didn't present intrinsic activity of $\text{Ni}(\text{OH})_x/\text{Cu}$ for $e\text{NO}_2^-$ RR.

Response: Two additional independent tests over Cu, $\text{Ni}(\text{OH})_x/\text{Cu}$, and $\text{Ni}(\text{OH})_2$ were supplemented in order to adding error bar. The obtained intrinsic activities over Cu, $\text{Ni}(\text{OH})_x/\text{Cu}$, and $\text{Ni}(\text{OH})_2$ were displayed in Figure R2-1.

Figure R2-1. ECSA-normalized NH_3 yield rate of Cu, $\text{Ni}(\text{OH})_x/\text{Cu}$, and $\text{Ni}(\text{OH})_2$ for $e\text{NO}_2^-$ RR.

3. *In S table 3, why was apparent activity used to compare different types of catalysts?*

Response: Compared with the traditional Haber-Bosch method, electrochemical NH₃ synthesis alternatives mainly suffer from low NH₃ yield rate and Faradic efficiency. In this context, the apparent NH₃ yield is a vital index in evaluating the practical application potential of these technologies. Thus, the Supplementary Table 3 mainly focuses on comparing the NH₃ production rates of **different electrocatalytic NH₃ synthesis routes** based on clean energy sources.

4. *As authors mentioned “we hypothesize that the improved eNO₃⁻RR performance on Ni(OH)_x/Cu is due to its enhanced water activation”. However, the eNO₂⁻RR performance on Ni(OH)_x/Cu just improved slightly compared with Cu.*

Response: Thanks to the reviewer for this valuable comment. Different from the eNO₃⁻RR, eNO₂⁻RR over Cu in the alkaline environment does not encounter the issue of by-product accumulation within the potential range we examined, leading to high Faradic efficiencies toward NH₃ production. Thus, both Cu and Ni(OH)_x/Cu exhibited excellent NH₃ selectivity in eNO₂⁻RR. However, as shown in Table R2-1, the apparent NH₃ yield rates of Ni(OH)_x/Cu are over 1.6 times higher than those of Cu. Meanwhile, as shown in Figure R2-1, ECSA-normalized NH₃ yield rates over Ni(OH)_x/Cu are higher than those over Cu. These results indicate that the enhanced water activation also facilitate eNO₂⁻RR over Ni(OH)_x/Cu.

Table R2-1. Potential-dependent apparent NH₃ yield rates over Cu and Ni(OH)_x/Cu for eNO₂⁻RR.

Potential (V vs. RHE)	Cu	Ni(OH)_x/Cu	Increase multiple
0	0.55 ± 0.07	0.90 ± 0.04	1.64
-0.05	0.67 ± 0.05	1.18 ± 0.20	1.76
-0.1	0.88 ± 0.11	1.51 ± 0.20	1.72
-0.15	1.05 ± 0.05	1.76 ± 0.10	1.68
-0.2	1.29 ± 0.09	2.10 ± 0.17	1.63
-0.25	1.44 ± 0.06	2.54 ± 0.10	1.67
-0.3	1.64 ± 0.09	2.97 ± 0.09	1.81

REVIEWERS' COMMENTS

Reviewer #1 (Remarks to the Author):

In the revised manuscript, the authors have made efforts to answer the reviewers' questions and added relevant content to the revised manuscript. Now I think the manuscript can be published